# TRANSFORMERS CAN LEARN TEMPORAL DIFFERENCE METHODS FOR IN-CONTEXT REINFORCEMENT LEARNING

**Jiuqi Wang**[*]
University of Virginia
jiuqi@email.virginia.edu

**Ethan Blaser**[*]
University of Virginia
blaser@email.virginia.edu

**Hadi Daneshmand**[†]
University of Virginia
dhadi@virginia.edu

**Shangtong Zhang**
University of Virginia
shangtong@virginia.edu

## ABSTRACT

Traditionally, reinforcement learning (RL) agents learn to solve new tasks by updating their neural network parameters through interactions with the task environment. However, recent works demonstrate that some RL agents, after certain pretraining procedures, can learn to solve unseen new tasks without parameter updates, a phenomenon known as in-context reinforcement learning (ICRL). The empirical success of ICRL is widely attributed to the hypothesis that the forward pass of the pretrained agent neural network implements an RL algorithm. In this paper, we support this hypothesis by showing, both empirically and theoretically, that when a transformer is trained for policy evaluation tasks, it can discover and learn to implement temporal difference learning in its forward pass.

## 1 INTRODUCTION

In reinforcement learning (RL, Sutton and Barto (2018)), an agent typically learns to solve new tasks by updating its neural network parameters based on interactions with the task environment. For example, the DQN agent (Mnih et al., 2015) incrementally updates the parameters of its $Q$-network while playing the Atari games (Bellemare et al., 2013). However, recent works (e.g., Duan et al. (2016); Wang et al. (2016); Laskin et al. (2022)) demonstrate that RL can also occur without any parameter updates. These works demonstrate that an RL agent with *fixed pretrained parameters* can take as input its observation history in the new task (referred to as context) and output good actions for that task. Specifically, let $\tau_t \doteq (S_0, A_0, R_1, \ldots, S_{t-1}, A_{t-1}, R_t)$ be a sequence of state-action-reward triples that an agent obtains until time $t$ in some new task. This $\tau_t$ is referred to as the *context*. The agent then outputs an action $A_t$ based on the context $\tau_t$ and the current state $S_t$ without updating its parameters. Notably, the context can span multiple episodes. As the context length increases, action quality improves, suggesting that this improvement is not due to memorized policies encoded in the fixed parameters. Instead, it indicates that a reinforcement learning process occurs during the forward pass as the agent processes the context, a phenomenon termed in-context reinforcement learning (ICRL), where RL occurs at inference time within the forward pass. See Moeini et al. (2025) for a comprehensive survey of ICRL.

Previous works (e.g., Lin et al. (2023)) understand the ICRL phenomenon from the supervised pretraining perspective, where during the pretraining stage, the RL agent is explicitly tasked to imitate the behavior of some existing RL algorithms. It is therefore not surprising that the pretrained agent neural network implements the corresponding RL algorithm in its forward pass. This paper instead provides the first theoretical analysis of the emergence of ICRL from reinforcement pretraining, where in the pretraining stage, the RL agent is only asked to complete some task, but there is no

---

[*]Equal contribution. The order is determined by tossing a fair coin.
[†]Work performed while affiliated with MIT LIDS/Boston University.

constraint on how it should complete it. ICRL emerges in the sense that the agent neural network itself discovers and implements a certain RL algorithm in the forward pass.

Although most existing ICRL studies focus on control tasks (i.e., outputting actions given a state and context), to better understand ICRL, in this work, we investigate ICRL for policy evaluation, as it is widely known in the RL community that understanding policy evaluation is often the first step toward understanding control (Sutton and Barto, 2018). Specifically, suppose an agent with fixed pretrained parameters follows some fixed policy $\pi$ in a new task. We explore how the agent can estimate the value function $v_\pi(s)$ for a given state $s$ based on its context $\tau_t$[1] without parameter updates. We call this *in-context policy evaluation* and believe that understanding it will pave the way for a comprehensive understanding of ICRL. We demonstrate that ICRL can emerge even with simplified neural network architectures, e.g., transformer (Vaswani et al., 2017) with linear attention[2].

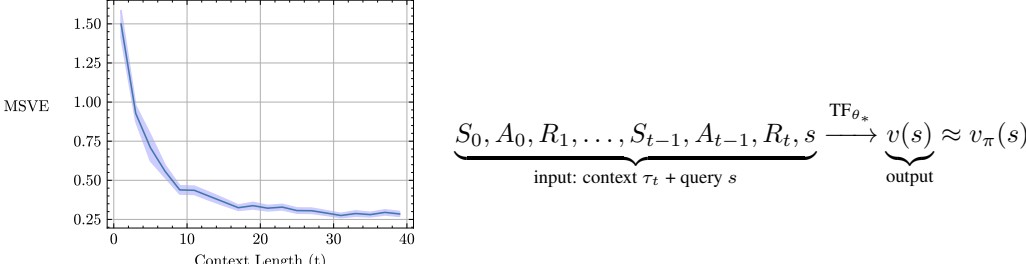

Figure 1: A transformer capable of in-context policy evaluation. This 15-layer transformer $\text{TF}_{\theta_*}$ takes the context $\tau_t$ and a state of interest $s$ as input and outputs $\text{TF}_{\theta_*}(\tau_t, s)$ as the estimation of the state value $v_\pi(s)$. The $y$-axis is the mean square value error (MSVE) $\sum_s d_\pi(s)(\text{TF}_{\theta_*}(\tau_t, s) - v_\pi(s))^2$, with $d_\pi(s)$ being the stationary state distribution. The curves are averaged over 300 randomly generated policy evaluation tasks, with shaded regions being standard errors. The tasks vary in state space, transition function, reward function, and policy. Yet a single $\theta_*$ is used for all tasks. See Appendix B for more details.

Figure 1 provides a concrete example of a transformer capable of in-context policy evaluation. To our knowledge, this is the first empirical demonstration of in-context policy evaluation. Let $\text{TF}_{\theta_*}$ denote the transformer used in Figure 1 with parameters $\theta_*$. Figure 1 demonstrates that the value approximation error of this transformer drops when the context length $t$ increases even though $\theta_*$ remains fixed. Notably, this improvement cannot be attributed to $\theta_*$ hard-coding the true value function. The approximation error in Figure 1 is averaged over a wide range of tasks and policies, each with distinct value functions, while only a single $\theta_*$ is used. The only plausible explanation seems to be that the transformer $\text{TF}_{\theta_*}$ is able to perform some policy evaluation algorithm in the forward pass to process the context and thus predict the value of $s$. This immediately raises two key questions:

(Q1) What is the specific policy evaluation algorithm that $\text{TF}_{\theta_*}$ is implementing?
(Q2) What kind of pretraining can generate such a powerful transformer?

This work aims to answer these questions to better understand in-context RL for policy evaluation. To this end, this work makes three contributions.

First, we confirm the existence of such a $\theta_*$ by construction. We prove that this $\theta_*$ enables in-context policy evaluation because the layer-by-layer forward pass of $\text{TF}_{\theta_*}$ is precisely equivalent to the iteration-by-iteration updates of a batch version of temporal difference learning (TD, Sutton (1988)). To summarize, a short answer to (Q1) is "TD." **This is the first time that a pretrained agent neural network for ICRL is fully white-boxed**. Furthermore, we also prove by construction that

---

[1]We, of course, also need to provide the discount factor to the agent. We ignore it for now to simplify the presentations.

[2]Linear attention is a widely used transformer variant for simplifying both computation and analysis (Katharopoulos et al., 2020; Wang et al., 2020; Schlag et al., 2021; Choromanski et al., 2020; Mahankali et al., 2023; Ahn et al., 2023; Von Oswald et al., 2023a;b; Wu et al., 2023; Ahn et al., 2024; Gatmiry et al., 2024; Zhang et al., 2024; Zheng et al., 2024; Sander et al., 2024).

transformers are capable of implementing many other policy evaluation algorithms, including TD($\lambda$) (Sutton, 1988), residual gradient (Baird, 1995), and average reward TD (Tsitsiklis and Roy, 1999).

Second, we empirically demonstrate that this $\theta_*$ naturally emerges after we regard TF$_\theta$ as a standard nonlinear function approximator and train it using nonlinear TD on multiple randomly generated policy evaluation tasks (similar to training a single DQN agent on multiple Atari games). This empirical finding is surprising because the pretraining only drives TF$_\theta$ to output good value estimates. There is no explicit mechanism that forces the transformer's weights to implement TD in its forward pass (cf. that the forward pass of DQN's $Q$-network can be anything as long as it outputs good action value approximations). Despite having the capacity to implement other algorithms like residual gradient, the pretraining process consistently leads the transformer weights to converge to those that implement TD. This observation parallels the historical development of the RL community itself, where TD became the favored method for policy evaluation after extensive trial-and-error with alternative approaches. Thus, a short answer to Question (Q2) is also "TD." Naturally, this leads to our third and final question.

(Q3) Why does TD pretraining give rise to in-context TD?

Our third contribution addresses this question by proving that the parameters $\theta_*$ that implement TD in the forward pass lie in an invariant set of the TD pretraining algorithm. It is, of course, not a complete answer. Similar to Wu et al. (2023); Zhang et al. (2024), we only prove the single-layer case, and we do not prove that the parameters will surely converge to this invariant set. However, we argue that our invariant set analysis and the techniques developed to prove it are a significant step toward future work that can fully characterize how in-context reinforcement learning emerges from reinforcement pretraining.

## 2    RELATED WORKS

Our first question (Q1) is closely related to the expressivity of neural networks (Siegelmann and Sontag, 1992; Graves et al., 2014; Jastrzębski et al., 2017; Hochreiter et al., 2001; Lu et al., 2017). Per the universal approximation theorem (Hornik et al., 1989; Cybenko, 1989; Leshno et al., 1993; Bengio et al., 2017), sufficiently wide neural networks can approximate any function arbitrarily well. However, this theorem focuses only on input-output behavior, meaning that given the same input, the network will produce similar outputs as the target function. It does not say anything about how the forward pass is able to produce the desired outputs, nor how the number of layers affects the approximation error. In the supervised learning community, there are a few works that are able to white-box the forward pass of neural networks to some extent (Frosst and Hinton, 2017; Alvarez Melis and Jaakkola, 2018; Chan et al., 2022; Yu et al., 2023; Von Oswald et al., 2023a; Ahn et al., 2024). But in the RL community, this work is, to our knowledge, the first to white-box how the forward pass of a pretrained transformer can implement RL algorithms. Notably, Lin et al. (2023) also construct some weights of transformer such that the forward pass of the transformer can implement some RL algorithm. However, their constructed transformer is overly complicated and there is no evidence that their weight construction can emerge through any kind of pretraining in practice.

Our second question (Q2) is closely related to the pretraining in ICRL, which can be divided into supervised pretraining and reinforcement pretraining. In supervised pretraining, the agent is explicitly tasked with imitating the behavior of some existing RL algorithms demonstrated in an offline dataset (Xu et al., 2022; Laskin et al., 2022; Raparthy et al., 2023; Sinii et al., 2023; Zisman et al., 2023; Shi et al., 2024; Dai et al., 2024; Huang et al., 2024a;b; Kirsch et al., 2023; Wang et al., 2024; Lee et al., 2024). It is thus less surprising (Krishnamurthy et al., 2024) that the pretrained agent network does implement some RL algorithm in the forward pass. Supervised pretraining can be understood through the lens of behavior cloning, see Lin et al. (2023) for a theoretical analysis of supervised pretraining. In reinforcement pretraining, the agent is only tasked with maximizing the return, and there is no constraint on how the agent network should achieve this in the forward pass (Duan et al., 2016; Wang et al., 2016; Kirsch et al., 2022; Bauer et al., 2023; Grigsby et al., 2023; Lu et al., 2023; Park et al., 2024; Xu et al., 2024; Grigsby et al., 2024; Cook et al., 2024). Reinforcement pretraining also is closely related to algorithm discovery in meta RL. The difference is that the reinforcement pretraining in ICRL discovers and implements novel RL algorithm in the forward pass without parameter updates, while a large body of prior works of algorithm discovery in meta RL (Kirsch et al., 2019; Oh et al.,

2020; Lu et al., 2022) require parameter updates when executing the discovered algorithm. See Beck et al. (2023) for a comprehensive survey of meta RL and see Moeini et al. (2025) for a comprehensive survey of different pretraining methods for ICRL. The reinforcement pretraining method we use is a very simple version of multi-task RL and is very standard in the meta RL community. We do not claim any novelty in our pretraining method. Instead, the novelty lies in the empirical and theoretical analysis of this simple yet standard pretraining method.

Our third question (Q3) is closely related to the dynamics of RL algorithms (Borkar and Meyn, 2000; Bhandari et al., 2018; Cai et al., 2019; Qian et al., 2024; Liu et al., 2025), which is an active research area. In the context of ICRL, the dynamics of supervised pretraining is previously studied in Lin et al. (2023) following the behavior cloning framework. This work is to our knowledge the first theoretical analysis of reinforcement pretraining for ICRL.

ICRL is broadly related to the general in-context learning (ICL) community in machine learning (Garg et al., 2022; Müller et al., 2022; Akyürek et al., 2023; Von Oswald et al., 2023a; Zhao et al., 2023; Allen-Zhu and Li, 2023; Mahankali et al., 2023; Ahn et al., 2024; Zhang et al., 2024). While ICL is widely studied in the context of large language models (LLMs) (Brown et al., 2020), ICRL and LLM-based ICL represent distinct areas of research. ICRL typically needs RL-based pretraining while LLM's pretraining is usually unsupervised. Additionally, ICRL focuses on RL capabilities during inference, while LLM-based ICL typically examines supervised learning behavior during inference. RL and supervised learning are fundamentally different problems, and similarly, ICRL and in-context supervised learning (ICSL) require different approaches. For example, Ahn et al. (2024) prove that ICSL can be viewed as gradient descent in the forward pass. While our work draws inspiration from Ahn et al. (2024), the scenario in ICRL is more complex. TD, which we analyze in this paper, is not equivalent to gradient descent. Our proof that transformers can implement TD in the forward pass is, therefore, more intricate, especially when extending it to TD($\lambda$) and average reward TD. Moreover, Ahn et al. (2024) consider a gradient descent-based pretraining paradigm where the transformer is trained to minimize an in-context regression loss. As a result, they analyze the critical points of the regression loss to understand their pretraining. By contrast, we consider TD-based pretraining, which is not gradient descent. To address this, we introduce a novel invariant set perspective to analyze the behavior of transformers under TD-based pretraining.

## 3 BACKGROUND

**Transformers and Linear Self-Attention.** All vectors are column vectors. We denote the identity matrix in $\mathbb{R}^n$ by $I_n$ and an $m \times n$ all-zero matrix by $0_{m \times n}$. We use $Z^\top$ to denote the transpose of $Z$ and use both $\langle x, y \rangle$ and $x^\top y$ to denote the inner product. Given a prompt $Z \in \mathbb{R}^{d \times n}$, standard single-head self-attention (Vaswani et al., 2017) processes the prompt by $\text{Attn}_{W_k, W_q, W_v}(Z) \doteq W_v Z \text{softmax}\left(Z^\top W_k^\top W_q Z\right)$, where $W_v \in \mathbb{R}^{d \times d}$, $W_k \in \mathbb{R}^{m \times d}$, and $W_q \in \mathbb{R}^{m \times d}$ represent the value, key and query weight matrices. The softmax function is applied to each row. Linear attention is a widely used architecture in transformers (Mahankali et al., 2023; Ahn et al., 2023; Von Oswald et al., 2023a;b; Wu et al., 2023; Ahn et al., 2024; Gatmiry et al., 2024; Zhang et al., 2024; Zheng et al., 2024; Sander et al., 2024), where the softmax function is replaced by an identity function. Given a prompt $Z \in \mathbb{R}^{(2d+1) \times (n+1)}$, linear self-attention is defined as

$$\text{LinAttn}(Z; P, Q) \doteq PZM(Z^\top QZ), \tag{1}$$

where $P \in \mathbb{R}^{(2d+1) \times (2d+1)}$ and $Q \in \mathbb{R}^{(2d+1) \times (2d+1)}$ are parameters and $M \in \mathbb{R}^{(n+1) \times (n+1)}$ is a *fixed* mask of the input matrix $Z$, defined as

$$M \doteq \begin{bmatrix} I_n & 0_{n \times 1} \\ 0_{1 \times n} & 0 \end{bmatrix}. \tag{2}$$

Note that we can view $P$ and $Q$ as reparameterizations of the original weight matrices for simplifying presentation. The mask $M$ is introduced for in-context learning (Von Oswald et al., 2023a) to designate the last column of $Z$ as the query and the first $n$ columns as the context. We use this fixed mask in most of this work. However, the linear self-attention mechanism can be altered using a different mask $M'$, when necessary, by defining $\text{LinAttn}(Z; P, Q, M') = PZM'(Z^\top QZ)$. In an $L$-layer transformer with parameters $\{(P_l, Q_l)\}_{l=0,\dots,L-1}$, the input $Z_0$ evolves layer by layer as

$$Z_{l+1} \doteq Z_l + \frac{1}{n}\text{LinAttn}_{P_l, Q_l}(Z_l) = Z_l + \frac{1}{n}P_l Z_l M(Z_l^\top Q_l Z_l). \tag{3}$$

Here, $\frac{1}{n}$ is a normalization factor simplifying presentation. We follow the convention in Von Oswald et al. (2023a); Ahn et al. (2024) and use

$$\text{TF}_L(Z_0; \{P_l, Q_l\}_{l=0,1,\ldots L-1}) \doteq -Z_L[2d+1, n+1] \tag{4}$$

to denote the output of the $L$-layer transformer, given an input $Z_0$. Note that $Z_l[2d+1, n+1]$ is the bottom-right element of $Z_l$. Equation (4) establishes the notation convention that we adopt to define the output of a $L$-layer transformer. Specifically, linear attention produces a matrix, but for policy evaluation, we require a scalar output. Following prior works, we define the bottom-right element of the output matrix as this scalar.

**Reinforcement Learning.** We consider an infinite horizon Markov Decision Process (MDP, Puterman (2014)) with a finite state space $\mathcal{S}$, a finite action space $\mathcal{A}$, a reward function $r_{\text{MDP}} : \mathcal{S} \times \mathcal{A} \to \mathbb{R}$, a transition function $p_{\text{MDP}} : \mathcal{S} \times \mathcal{S} \times \mathcal{A} \to [0, 1]$, a discount factor $\gamma \in [0, 1)$, and an initial distribution $p_0 : \mathcal{S} \to [0, 1]$. An initial state $S_0$ is sampled from $p_0$. At a time $t$, an agent at a state $S_t$ takes an action $A_t \sim \pi(\cdot|S_t)$, where $\pi : \mathcal{A} \times \mathcal{S} \to [0, 1]$ is the policy being followed by the agent, receives a reward $R_{t+1} \doteq r_{\text{MDP}}(S_t, A_t)$, and transitions to a successor state $S_{t+1} \sim p_{\text{MDP}}(\cdot|S_t, A_t)$. If the policy $\pi$ is fixed, the MDP can be simplified to a Markov Reward Process (MRP) where transitions and rewards are determined solely by the current state: $S_{t+1} \sim p(\cdot|S_t)$ with $R_{t+1} \doteq r(S_t)$. Here, $p(s'|s) \doteq \sum_a \pi(a|s) p_{\text{MDP}}(s'|s, a)$ and $r(s) \doteq \sum_a \pi(a|s) r_{\text{MDP}}(s, a)$. In this work, we consider the policy evaluation problem where the policy $\pi$ is fixed. So, it suffices to consider only an MRP represented by the tuple $(p_0, p, r)$, and trajectories $(S_0, R_1, S_1, R_2, \ldots)$ sampled from it. The value function of this MRP is defined as $v(s) \doteq \mathbb{E}\left[\sum_{i=t+1}^{\infty} \gamma^{i-t-1} R_i | S_t = s\right]$. Estimating the value function $v$ is one of the fundamental tasks in RL. To this end, one can consider a linear architecture. Let $\phi : \mathcal{S} \to \mathbb{R}^d$ be the feature function. The goal is then to find a weight vector $w \in \mathbb{R}^d$ such that for each $s$, the estimated value $\hat{v}(s; w) \doteq w^\top \phi(s)$ approximates $v(s)$. TD is a prevalent method for learning this weight vector, which updates $w$ iteratively as

$$\begin{aligned}
w_{t+1} &= w_t + \alpha_t \left(R_{t+1} + \gamma \hat{v}(S_{t+1}; w_t) - \hat{v}(S_t; w_t)\right) \nabla \hat{v}(S_t; w_t) \\
&= w_t + \alpha_t \left(R_{t+1} + \gamma w_t^\top \phi(S_{t+1}) - w_t^\top \phi(S_t)\right) \phi(S_t),
\end{aligned} \tag{5}$$

where $\{\alpha_t\}$ is a sequence of learning rates. Notably, TD is not a gradient descent algorithm. It is instead considered as a *semi-gradient* algorithm because the gradient is only taken with respect to $\hat{v}(S_t; w_t)$ and does not include the dependence on $\hat{v}(S_{t+1}; w_t)$ (Sutton and Barto, 2018). Including this dependency modifies the update to

$$w_{t+1} = w_t + \alpha_t \left(R_{t+1} + \gamma w_t^\top \phi(S_{t+1}) - w_t^\top \phi(S_t)\right) \left(\phi(S_t) - \gamma \phi(S_{t+1})\right), \tag{6}$$

known as the (naïve version of) residual gradient method (Baird, 1995).[3] The update in (5) is also called TD(0) — a special case of the TD($\lambda$) algorithm (Sutton, 1988). TD($\lambda$) employs an eligibility trace that accumulates the gradients as $e_{-1} \doteq 0$, $e_t \doteq \gamma \lambda e_{t-1} + \phi(S_t)$ and updates $w$ iteratively as $w_{t+1} = w_t + \alpha_t (R_{t+1} + \gamma w_t^\top \phi(S_{t+1}) - w_t^\top \phi(S_t)) e_t$. The hyperparameter $\lambda$ controls the decay rate of the trace. If $\lambda = 0$, we recover (5). On the other end with $\lambda = 1$, it is known that TD($\lambda$) recovers Monte Carlo (Sutton, 1988). Another important setting in RL is the average-reward setting (Puterman, 2014; Sutton and Barto, 2018), focusing on the rate of receiving rewards, without using a discount factor $\gamma$. The average reward $\bar{r}$ is defined as $\bar{r} \doteq \lim_{T \to \infty} \frac{1}{T} \sum_{t=1}^{T} \mathbb{E}[R_t]$. Similar to the value function in the discounted setting, a differential value function $\bar{v}(s)$ is defined for the average-reward setting as $\bar{v}(s) \doteq \mathbb{E}\left[\sum_{i=t+1}^{\infty}(R_i - \bar{r})|S_t = s\right]$. One can similarly estimate $\bar{v}(s)$ using a linear architecture with a vector $w$ as $w^\top \phi(s)$. Average-reward TD (Tsitsiklis and Roy, 1999) updates $w$ iteratively as $w_{t+1} = w_t + \alpha_t\left(R_{t+1} - \bar{r}_{t+1} + w_t^\top \phi(S_{t+1}) - w_t^\top \phi(S_t)\right)\phi(S_t)$, where $\bar{r}_t \doteq \frac{1}{t} \sum_{i=1}^{t} R_i$ is the empirical average of the received reward.

## 4 TRANSFORMERS CAN IMPLEMENT IN-CONTEXT TD(0)

In this section, we reveal the parameters of the transformer used to generate Figure 1 and answer (Q1). Namely, we construct that transformer below and prove that it implements TD(0) in its forward

---

[3]This is a naïve version because the update does not account for the double sampling issue. We refer the reader to Chapter 11 of Sutton and Barto (2018) for detailed discussion.

pass. Given a trajectory $(S_0, R_1, S_1, R_2, S_3, R_4, \ldots, S_n)$ sampled from an MRP, using as shorthand $\phi_i \doteq \phi(S_i)$, we define for $l = 0, 1, \ldots, L - 1$

$$
Z_0 = \begin{bmatrix} \phi_0 & \cdots & \phi_{n-1} & \phi_n \\ \gamma\phi_1 & \cdots & \gamma\phi_n & 0 \\ R_1 & \cdots & R_n & 0 \end{bmatrix}, P_l^{\mathrm{TD}} \doteq \begin{bmatrix} 0_{2d \times 2d} & 0_{2d \times 1} \\ 0_{1 \times 2d} & 1 \end{bmatrix}, Q_l^{\mathrm{TD}} \doteq \begin{bmatrix} -C_l^\top & C_l^\top & 0_{d \times 1} \\ 0_{d \times d} & 0_{d \times d} & 0_{d \times 1} \\ 0_{1 \times d} & 0_{1 \times d} & 0 \end{bmatrix}. \quad (7)
$$

Here, $Z_0 \in \mathbb{R}^{(2d+1) \times (n+1)}$ is the prompt matrix, $C_l \in \mathbb{R}^{d \times d}$ is an arbitrary matrix, and $\left\{ (P_l^{\mathrm{TD}}, Q_l^{\mathrm{TD}}) \right\}_{l=0,1,\ldots,L-1}$ are the parameters of the $L$-layer transformer. We then have

**Theorem 1** (Forward pass as TD(0)). *Consider the L-layer linear transformer following* (3), *using the mask* (2), *parameterized by* $\left\{ P_l^{TD}, Q_l^{TD} \right\}_{l=0,\ldots,L-1}$ *in* (7). *Let* $y_l^{(n+1)}$ *be the bottom right element of the l-th layer's output, i.e.,* $y_l^{(n+1)} \doteq Z_l[2d+1, n+1]$. *Then, it holds that* $y_l^{(n+1)} = -\langle \phi_n, w_l \rangle$, *where* $\{w_l\}$ *is defined as* $w_0 = 0$ *and*

$$
w_{l+1} = w_l + \frac{1}{n} C_l \sum_{j=0}^{n-1} \left( R_{j+1} + \gamma w_l^\top \phi_{j+1} - w_l^\top \phi_j \right) \phi_j. \quad (8)
$$

The proof is in Appendix A.1 and with numerical verification in Appendix H as a sanity check. Notably, Theorem 1 holds for any $C_l$. In particular, if $C_l = \alpha_l I$ (this is used in the transformer to generate Figure 1), then the update (8) becomes a batch version of TD(0) in (5). For a general $C_l$, the update (8) can be regarded as preconditioned batch TD(0) (Yao and Liu, 2008). Theorem 1 precisely demonstrates that transformers are expressive enough to implement iterations of TD in its forward pass. We call this *in-context TD*. It should be noted that although the construction of $Z_0$ in (7) uses $\phi_n$ as the query state for conceptual clarity, any arbitrary state $s \in \mathcal{S}$ can serve as the query state and Theorem 1 still holds. In other words, by replacing $\phi_n$ with $\phi(s)$, the transformer will then estimate $v(s)$. Notably, if the transformer has only one layer, i.e., $L = 1$, there are other parameter configurations that can also implement in-context TD(0).

**Corollary 1.** *Consider the 1-layer linear transformer following* (3), *using the mask* (2). *Consider the following parameters*

$$
P_0^{TD} \doteq \begin{bmatrix} 0_{2d \times 2d} & 0_{2d \times 1} \\ 0_{1 \times 2d} & 1 \end{bmatrix}, Q_0^{TD} \doteq \begin{bmatrix} -C_l^\top & 0_{d \times d} & 0_{d \times 1} \\ 0_{d \times d} & 0_{d \times d} & 0_{d \times 1} \\ 0_{1 \times d} & 0_{1 \times d} & 0 \end{bmatrix} \quad (9)
$$

*Then, it holds that* $y_1^{(n+1)} = -\langle \phi_n, w_1 \rangle$, *where* $w_1$ *is defined as*

$$
w_1 = w_0 + \frac{1}{n} C_l \sum_{j=0}^{n-1} \left( R_{j+1} + \gamma w_0^\top \phi_{j+1} - w_0^\top \phi_j \right) \phi_j \quad \text{with } w_0 = 0.
$$

The proof is in Appendix A.2. An observant reader may notice that this corollary holds primarily because $w_0 = 0$, making it a unique result for $L = 1$. Nevertheless, this special case helps understand a few empirical and theoretical results below.

## 5 TRANSFORMERS DO IMPLEMENT IN-CONTEXT TD(0)

In this section, we reveal our pretraining method that generates the powerful transformer used in Figure 1, answering (Q2).[4] We also theoretically analyze this pretraining method, answering (Q3).

**Multi-Task Temporal Difference Learning.** In existing ICRL works for control, the transformer takes the observation history as input and outputs actions. A behavior cloning loss is used during pretraining to ensure that the transformer outputs actions similar to those in the pretraining data. In contrast, our work seeks to understand ICRL through the lens of policy evaluation, where the goal is for the transformer to output value estimates rather than actions. To ground the value estimation, we use the most straightforward method in RL: the TD loss. This yields a pretraining algorithm (Algorithm 1) where the transformer is trained using nonlinear TD on multiple tasks. We call it multi-task TD.

Recall that a policy evaluation task is essentially a tuple $(p_0, p, r, \phi)$. In Algorithm 1, we assume that there is a task distribution $d_{\mathrm{task}}$ over those tuples. Recall that $\mathrm{TF}_L(Z_0; \theta)$ and $\mathrm{TF}_L(Z_0'; \theta)$ are

---

[4]The implementation is available at https://github.com/Sequential-Intelligence-Lab/InContextTD

---

**Algorithm 1:** Multi-Task Temporal Difference Learning

---

1: **Input:** context length $n$, MRP sample length $\tau$, number of training tasks $k$, learning rate $\alpha$, discount factor $\gamma$, transformer parameters $\theta \doteq \{P_l, Q_l\}_{l=0,1,\dots L-1}$

2: **for** $i \leftarrow 1$ **to** $k$ **do**

3:     Sample $(p_0, p, r, \phi)$ from $d_{\text{task}}$

4:     Sample $(S_0, R_1, S_1, R_2, \dots, S_\tau, R_{\tau+1}, S_{\tau+1})$ from the MRP $(p_0, p, r)$

5:     **for** $t = 0, \dots, \tau - n - 1$ **do**

6:         $Z_0 \leftarrow \begin{bmatrix} \phi_t & \cdots & \phi_{t+n-1} & \phi_{t+n+1} \\ \gamma\phi_{t+1} & \cdots & \gamma\phi_{t+n} & 0 \\ R_{t+1} & \cdots & R_{t+n} & 0 \end{bmatrix}, Z_0' \leftarrow \begin{bmatrix} \phi_{t+1} & \cdots & \phi_{t+n} & \phi_{t+n+2} \\ \gamma\phi_{t+2} & \cdots & \gamma\phi_{t+n+1} & 0 \\ R_{t+2} & \cdots & R_{t+n+1} & 0 \end{bmatrix}$

7:         $\theta \leftarrow \theta + \alpha(R_{t+n+2} + \gamma\text{TF}_L(Z_0'; \theta) - \text{TF}_L(Z_0; \theta))\nabla_\theta\text{TF}_L(Z_0; \theta)$   `// TD`

8:     **end for**

9: **end for**

---

intended to estimate $v(S_{t+n+1})$ and $v(S_{t+n+2})$ respectively. So, Algorithm 1 essentially applies TD using $(S_{t+n+1}, R_{t+n+2}, S_{t+n+2})$ to train the transformer. Ideally, when a new prompt $Z_{\text{test}}$ is constructed using a trajectory from a new (possibly out-of-distribution) evaluation task $(p_0, p, r, \phi)_{\text{test}}$, the predicted value $\text{TF}_L(Z_{\text{test}}; \theta)$ with $\theta$ from Algorithm 1 should be close to the value of the query state in $Z_{\text{test}}$. This problem is a multi-task meta-learning problem, a well-explored area with many existing methodologies (Beck et al., 2023). However, the unique and significant aspect of our work is the demonstration that in-context TD emerges in the learned transformer, providing a novel *explanation* for how the model solves the problem.

**Empirical Analysis.** We first empirically study Algorithm 1. To this end, we construct $d_{\text{task}}$ based on Boyan's chain (Boyan, 1999), a canonical environment for diagnosing RL algorithms. We keep the structure of Boyan's chain but randomly generate initial distributions $p_0$, transition probabilities $p$, reward functions $r$, and the feature function $\phi$. Details of this random generation process are provided in Algorithm 2 with Figure 3 visualizing Boyan's chain, both in Appendix C.

For the linear transformer specified in (3), we first consider the autoregressive case following (Akyürek et al., 2023; Von Oswald et al., 2023a), where all the transformer layers share the same parameters, i.e., $P_l \equiv P_0$ and $Q_l \equiv Q_0$ for $l = 0, 1, \dots, L-1$. We consider a three-layer transformer ($L = 3$). Importantly, all elements of $P_0$ and $Q_0$ are equally trainable — we did not force any element of $P_0$ or $Q_0$ to be 0. We then run Algorithm 1 with Boyan's chain-based evaluation tasks (i.e., $d_{\text{task}}$) to train this autoregressive transformer. The dimension of the feature is $d = 4$ (i.e., $\phi(s) \in \mathbb{R}^4$). Other hyperparameters of Algorithm 1 are specified in Appendix D.1.

Figure 2a visualizes the final learned $P_0$ and $Q_0$ by Algorithm 1 after 4000 MRPs (i.e., $k = 4000$), which closely match our specifications $P^{\text{TD}}$ and $Q^{\text{TD}}$ in (7) with $C_l = I_d$. In Figure 2b, we visualize the element-wise learning progress of $P_0$ and $Q_0$. We observe that the bottom right element of $P_0$ increases (the $P_0[-1, -1]$ curve), while the average absolute value of all other elements remain close to zero (the "Avg Abs Others" curve), closely aligning with $P^{\text{TD}}$ up to some scaling factor. Furthermore, the trace of the upper left $d \times d$ block of $Q_0$ approaches $-d$ (the $\text{tr}(Q_0[:d,:d])$ curve), and the trace of the upper right block (excluding the last column) approaches $d$ (the $\text{tr}(Q_0[:d, d:2d])$ curve). Meanwhile, the average absolute value of all the other elements in $Q_0$ remain near zero, aligning with $Q^{\text{TD}}$ using $C_l = I_d$ up to some scaling factor.

More empirical analysis is provided in the Appendix. In particular, besides showing the parameter-wise convergence in Figure 2, we also use other metrics including value difference, implicit weight similarity, and sensitivity similarity, inspired by Von Oswald et al. (2023a); Akyürek et al. (2023), to examine the learned transformer. We also study **normal transformers without parameter sharing** (Appendix D.3), as well as **different choices of hyperparameters** in Algorithm 1. Furthermore, we empirically investigate the original **softmax-based transformers** (Appendix E). Finally, we also conducted experiments where we constructed $d_{\text{task}}$ based on the **Cartpole** environment (Brockman et al., 2016) (Appendix F). The overall conclusion is the same — in-context TD emerges in the transformers learned by Algorithm 1. Notably, Theorem 1 and Corollary 1 suggest that for $L = 1$, there are two distinct ways to implement in-context TD (i.e., (7) v.s. (9)). Our empirical results in Appendix D.2 show that Algorithm 1 ends up with (9) in Corollary 1 for $L = 1$, aligning well with

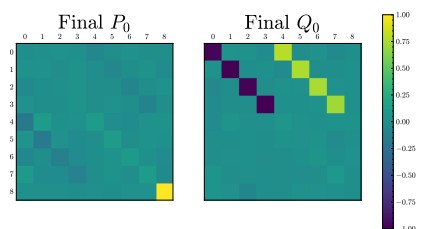
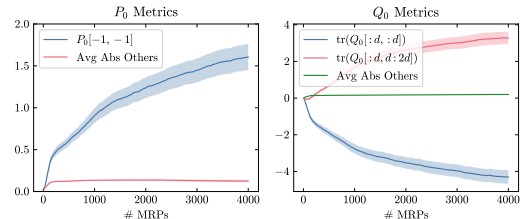

(a) Learned $P_0$ and $Q_0$ after 4000 MRPs

(b) Element-wise learning progress of $P_0$ and $Q_0$

Figure 2: Visualization of the learned transformers and the learning progress. Both (a) and (b) are averaged across 30 seeds and the shaded regions in (b) denotes the standard errors. Since $P_0$ and $Q_0$ are in the same product in (1), the algorithm can rescale both or flip the sign of both, but still end up with exactly the same transformer. Therefore, to make sure the visualization are informative, we rescale $P_0$ and $Q_0$ properly first before visualization. See Appendix D.1.1 for details.

**Theorem 2.** For $L = 2, 3, 4$, Algorithm 1 always ends up with (7) in Theorem 1, as shown in Figure 4 in Appendix D.2. We also empirically observed that for in-context TD to emerge, the task distribution $d_{\text{task}}$ has to be "difficult" enough. For example, if $(p_0, p)$ or $\phi$ are always fixed, we did not observe the emergence of in-context TD.

**Theoretical Analysis.** The problem that Algorithm 1 aims to solve is highly non-convex and non-linear (the linear transformer is still a nonlinear function). We analyze a simplified version of Algorithm 1 and leave the treatment to the full version for future work. In particular, we study the single-layer case with $L = 1$, and let $\theta \doteq (P_0, Q_0)$ be the parameters of the single-layer transformer. We consider expected updates, i.e.,

$$\theta_{k+1} = \theta_k + \alpha_k \Delta(\theta_k) \text{ with } \Delta(\theta) \doteq \mathbb{E}\left[(R + \gamma \text{TF}_1(Z_0', \theta) - \text{TF}_1(Z_0, \theta))\nabla \text{TF}_1(Z_0, \theta)\right]. \quad (10)$$

Here, the expectation integrates both the randomness in sampling $(p_0, p, r, \phi)$ from $d_{\text{task}}$ and the randomness in constructing $(R, Z_0, Z_0')$ thereafter. We sample $(S_0, R_1, S_1, \ldots, S_{n+1}, R_{n+2}, S_{n+2})$ following $(p_0, p, r)$ and construct using shorthand $\phi_i \doteq \phi(S_i)$

$$Z_0 \doteq \begin{bmatrix} \phi_0 & \ldots & \phi_{n-1} & \phi_{n+1} \\ \gamma\phi_1 & \ldots & \gamma\phi_n & 0 \\ R_1 & \ldots & R_n & 0 \end{bmatrix}, Z_0' \doteq \begin{bmatrix} \phi_1 & \ldots & \phi_n & \phi_{n+2} \\ \gamma\phi_2 & \ldots & \gamma\phi_{n+1} & 0 \\ R_2 & \ldots & R_{n+1} & 0 \end{bmatrix}, R \doteq R_{n+2}. \quad (11)$$

The structure of $Z_0$ and $Z_0'$ is similar to those in Algorithm 1. The main difference is that we do not use the sliding window. We recall that $(p_0, p, r, \phi)$ are random variables with joint distribution $d_{\text{task}}$. Here, $\phi$ is essentially a random matrix taking value in $\mathbb{R}^{d \times |\mathcal{S}|}$, represented as $\phi = [\phi(s)]_{s \in \mathcal{S}}$. We use $\triangleq$ to denote "equal in distribution" and make the following assumptions.

**Assumption 5.1.** *The random matrix $\phi$ is independent of $(p_0, p, r)$.*

**Assumption 5.2.** $\Pi\phi \triangleq \phi, \Lambda\phi \triangleq \phi$, *where $\Pi$ is any $d$-dimensional permutation matrix and $\Lambda$ is any diagonal matrix in $\mathbb{R}^d$ where each diagonal element of $\Lambda$ can only be $-1$ or $1$.*

Those assumptions are easy to satisfy. For example, as long as the elements of the random matrix $\phi$ are i.i.d. from a symmetric distribution centered at zero, e.g., a uniform distribution on $[-1, 1]$, then both assumptions hold. We say a set $\Theta$ is an invariant set of (10) if for any $k, \theta_k \in \Theta \implies \theta_{k+1} \in \Theta$. Define

$$\theta_*(\eta, c, c') \doteq \left( P_0 = \begin{bmatrix} 0_{2d \times 2d} & 0_{2d \times 1} \\ 0_{1 \times 2d} & \eta \end{bmatrix}, Q_0 = \begin{bmatrix} cI_d & 0_{d \times d} & 0_{d \times 1} \\ c'I_d & 0_{d \times d} & 0_{d \times 1} \\ 0_{1 \times d} & 0_{1 \times d} & 0 \end{bmatrix} \right).$$

**Theorem 2.** *Let Assumptions 5.1 and 5.2 hold. For the construction (11) of $(R, Z_0, Z_0')$, the set $\Theta_* \doteq \{\theta_*(\eta, c, c') | \eta, c, c' \in \mathbb{R}\}$ is an invariant set of (10).*

The proof is in Appendix A.3. Theorem 2 demonstrates that once $\theta_k$ enters $\Theta_*$ at some $k$, it can never leave, i.e., $\Theta_*$ is a candidate set that the update (10) can *possibly* converge to. Consider a subset $\Theta_*' \subset \Theta_*$ with a stricter constraint $c' = 0$, i.e., $\Theta_*' \doteq \{\theta_*(\eta, c, 0) | \eta, c \in \mathbb{R}\}$. Corollary 1 then confirms that all parameters in $\Theta_*'$ implement in-context TD. That being said, whether (10) is guaranteed to converge to $\Theta_*$, or further to $\Theta_*'$, is left for future work.

# 6 TRANSFORMERS CAN IMPLEMENT MORE RL ALGORITHMS

In this section, we prove that transformers are expressive enough to implement three additional well-known RL algorithms in the forward pass. We warm up with the (naive version of) residual gradient (RG). We then move to the more difficult TD($\lambda$). This section culminates with average-reward TD, which requires multi-head linear attention and memory within the prompt. We do note that whether those three RL algorithms will emerge after training is left for future work.

**Residual Gradient.** The construction of RG is an easy extension of Theorem 1. We define

$$P_l^{\text{RG}} = P_l^{\text{TD}}, Q_l^{\text{RG}} \doteq \begin{bmatrix} -C_l^\top & C_l^\top & 0_{d \times 1} \\ C_l^\top & -C_l^\top & 0_{d \times 1} \\ 0_{1 \times d} & 0_{1 \times d} & 0 \end{bmatrix} \in \mathbb{R}^{(2d+1) \times (2d+1)}. \tag{12}$$

**Corollary 2** (Forward pass as Residual Gradient). *Consider the L-layer linear transformer following* (3), *using the mask* (2), *parameterized by* $\left\{ P_l^{RG}, Q_l^{RG} \right\}_{l=0,\dots,L-1}$ *in* (12). *Define* $y_l^{(n+1)} \doteq Z_l[2d + 1, n + 1]$. *Then, it holds that* $y_l^{(n+1)} = -\langle \phi_n, w_l \rangle$, *where* $\{w_l\}$ *is defined as* $w_0 = 0$ *and*

$$w_{l+1} = w_l + \tfrac{1}{n} C_l \sum_{j=0}^{n-1} \left( R_{j+1} + \gamma w_l^\top \phi_{j+1} - w_l^\top \phi_j \right) (\phi_j - \gamma \phi_{j+1}). \tag{13}$$

The proof is in A.4 with numerical verification in Appendix H as a sanity check. Again, if $C_l \doteq \alpha_l I_d$, then (13) can be regarded as a batch version of (6). For a general $C_l$, it is then preconditioned batch RG. Notably, Figure 2 empirically demonstrates that Algorithm 1 eventually ends up with in-context TD instead of in-context RG. This observation aligns with the conventional wisdom in the RL community that TD is usually superior to the naïve RG (see, e.g., Zhang et al. (2020) and references therein).

**TD($\lambda$).** Incorporating eligibility traces is an important extension of TD(0). We now demonstrate that by using a different mask, transformers are able to implement in-context TD($\lambda$). We define

$$M^{\text{TD}(\lambda)} \doteq \begin{bmatrix} 1 & 0 & 0 & 0 & \cdots & 0 & 0 \\ \lambda & 1 & 0 & 0 & \cdots & 0 & 0 \\ \vdots & \vdots & \vdots & \vdots & \ddots & \vdots & \vdots \\ \lambda^{n-1} & \lambda^{n-2} & \lambda^{n-3} & \lambda^{n-4} & \cdots & 1 & 0 \\ 0 & 0 & 0 & 0 & \cdots & 0 & 0 \end{bmatrix} \in R^{(n+1) \times (n+1)}. \tag{14}$$

Notably, if $\lambda = 0$, the above mask for TD($\lambda$) recovers the mask for TD(0) in (2).

**Corollary 3** (Forward pass as TD($\lambda$)). *Consider the L-layer linear transformer parameterized by* $\left\{ P_l^{TD}, Q_l^{TD} \right\}_{l=0,\dots,L-1}$ *as specified in* (7) *with the input mask used in* (3) *being* $M^{TD(\lambda)}$ *in* (14). *Define* $y_l^{(n+1)} \doteq Z_l[2d + 1, n + 1]$. *Then, it holds that* $y_l^{(n+1)} = -\langle \phi_n, w_l \rangle$ *where* $\{w_l\}$ *is defined with* $w_0 = 0, e_0 = 0$, $e_j = \lambda e_{j-1} + \phi_j$, *and*

$$w_{k+1} = w_k + \tfrac{1}{n} C_k \sum_{i=0}^{n-1} \left( r_{i+1} + \gamma w_k^\top \phi_{i+1} - w_k^\top \phi_i \right) e_i.$$

The proof is in A.5 with numerical verification in Appendix H as a sanity check.

**Average-Reward TD.** We now demonstrate that transformers are expressive enough to implement in-context average-reward TD. Different from TD(0), average-reward TD (Tsitsiklis and Roy, 1999) exhibits additional challenges in that it updates two estimates (i.e., $w_t$ and $\bar{r}_t$) in parallel. To account for this challenge, we use two additional mechanisms beyond the basic single-head linear transformer. Namely, we allow additional "memory" in the prompt and consider two-head linear transformers. Given a trajectory $(S_0, R_1, S_1, R_2, S_3, R_4, \dots, S_n)$ sampled from an MRP, we construct the prompt matrix $Z_0$ as

$$Z_0 = \begin{bmatrix} \phi_0 & \cdots & \phi_{n-1} & \phi_n \\ \phi_1 & \cdots & \phi_n & 0 \\ R_1 & \cdots & R_n & 0 \\ 0 & \cdots & 0 & 0 \end{bmatrix} \in \mathbb{R}^{(2d+2) \times (n+1)}.$$

Notably, the last row of zeros is the "memory", which is used by the transformer to store some intermediate quantities during the inference time. We then define the transformer parameters and masks as

$$P_l^{\overline{\text{TD}},(1)} \doteq \begin{bmatrix} 0_{2d\times 2d} & 0_{2d\times 1} & 0_{2d\times 1} \\ 0_{1\times 2d} & 1 & 0 \\ 0_{1\times 2d} & 0 & 0 \end{bmatrix}, P_l^{\overline{\text{TD}},(2)} \doteq \begin{bmatrix} 0_{2d\times 2d} & 0_{2d\times 1} & 0_{2d\times 1} \\ 0_{1\times 2d} & 0 & 0 \\ 0_{1\times 2d} & 0 & 1 \end{bmatrix}, \tag{15}$$

$$Q_l^{\overline{\text{TD}}} \doteq \begin{bmatrix} -C_l^\top & C_l^\top & 0_{d\times 2} \\ 0_{d\times d} & 0_{d\times d} & 0_{d\times 2} \\ 0_{2\times d} & 0_{2\times d} & 0_{2\times 2} \end{bmatrix}, W_l \doteq \begin{bmatrix} 0_{2d\times 2d} & 0_{2d\times 1} & 0_{2d\times (2d+2)} & 0_{2d\times 1} \\ 0_{1\times 2d} & 1 & 0_{1\times (2d+2)} & 1 \end{bmatrix}, \tag{16}$$

$$M^{\overline{\text{TD}},(2)} \doteq \begin{bmatrix} I_n & 0_{n\times 1} \\ 0_{1\times n} & 0 \end{bmatrix}, M^{\overline{\text{TD}},(1)} \doteq \left(I_{n+1} - U_{n+1}\text{diag}\left(\begin{bmatrix} 1 & \frac{1}{2} & \cdots & \frac{1}{n+1} \end{bmatrix}\right)\right)M^{\overline{\text{TD}},(2)}, \tag{17}$$

where $C_l \in \mathbb{R}^{d\times d}$ is again an arbitrary matrix, $U_{n+1}$ is the $(n+1)\times(n+1)$ upper triangle matrix where all the nonzero elements are 1, and $\text{diag}(x)$ constructs a diagonal matrix, with the diagonal entry being $x$. Here, $\left\{P_l^{\overline{\text{TD}},(1)}, Q_l^{\overline{\text{TD}}}\right\}_{l=0,\ldots,L-1}$ are the parameters of the first attention heads, with the input mask being $M^{\overline{\text{TD}},(1)}$. $\left\{P_l^{\overline{\text{TD}},(2)}, Q_l^{\overline{\text{TD}}}\right\}_{l=0,\ldots,L-1}$ are the parameters of the second attention heads, with the input mask being $M^{\overline{\text{TD}},(2)}$. The two heads coincide on some parameters. $W_l$ is the affine transformation that combines the embeddings from the two attention heads. Define the two-head linear-attention as $\text{TwoHead}(Z; P, Q, M, P', Q', M', W) \doteq W\begin{bmatrix} \text{LinAttn}(Z; P, Q, M) \\ \text{LinAttn}(Z; P', Q', M') \end{bmatrix}$. The $L$-layer transformer we are interested in is then given by

$$Z_{l+1} \doteq Z_l + \frac{1}{n}\text{TwoHead}(Z_l; P_l^{\overline{\text{TD}},(1)}, Q_l^{\overline{\text{TD}}}, M^{\overline{\text{TD}},(1)}, P_l^{\overline{\text{TD}},(2)}, Q_l^{\overline{\text{TD}}}, M^{\overline{\text{TD}},(2)}, W_l). \tag{18}$$

**Theorem 3** (Forward pass as average-reward TD). *Consider the $L$-layer transformer in* (18). *Let $h_l^{(n+1)}$ be the bottom-right element of the $l$-th layer output, i.e., $h_l^{(n+1)} \doteq Z_l[2d+2, n+1]$. Then, it holds that $h_l^{(n+1)} = -\langle\phi_n, w_l\rangle$ where $\{w_l\}$ is defined as $w_0 = 0$,*

$$w_{l+1} = w_l + \frac{1}{n}C_l\sum_{j=1}^n \left(R_j - \bar{r}_j + w_l^\top\phi_j - w_l^\top\phi_{j-1}\right)\phi_{j-1}$$

*for $l = 0, \ldots, L-1$, where $\bar{r}_j \doteq \frac{1}{j}\sum_{k=1}^j R_k$.*

The proof is in A.6 with numerical verification in Appendix H as a sanity check.

## 7 CONCLUSION

This work makes the first step towards white-boxing the mechanism of ICRL under reinforcement pretraining, focusing specifically on policy evaluation. We provide constructive proof that transformers can implement multiple temporal difference algorithms in the forward pass for in-context policy evaluation. Additionally, we theoretically and empirically show that the parameters enabling in-context policy evaluation emerge naturally through multi-task TD pretraining. We find it compelling that a randomly initialized transformer, only trained for simple policy evaluation tasks, can learn to discover and implement TD, a provably capable RL algorithm for policy evaluation.

Admittedly, this work does have a few limitations. First, we focus solely on policy evaluation. Second, to facilitate theoretical analysis, we make a few assumptions (e.g., Assumptions 5.1 & 5.2) and simplifications (e.g., using linear attention instead of softmax attention). Yet those assumptions and simplifications may not hold in many practical scenarios. Third, our pretraining method (Algorithm 1) requires the ability to randomly generate policy evaluation tasks, which may not be available in many cases, such as offline training. Fourth, despite the fact that we evaluate Algorithm 1 in both Boyan's chain and CartPole, it is not evaluated on large-scale environments such as the Atari games (Bellemare et al., 2013) or DeepMindLab (Beattie et al., 2016). We believe that addressing those limitations would be fruitful directions for future works.

ACKNOWLEDGEMENTS

This work is supported in part by the US National Science Foundation (NSF) under grants III-2128019 and SLES-2331904. EB acknowledges support from the NSF Graduate Research Fellowship (NSF-GRFP) under award 1842490. HD acknowledges support from the NSF TRIPODS program under award DMS-2022448.

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

TABLE OF CONTENTS

## A Proofs

### A.1 Proof of Theorem 1

*Proof.* We recall from (3) that the embedding evolves according to

$$Z_{l+1} = Z_l + \frac{1}{n} P_l Z_l M(Z_l^\top Q_l Z_l).$$

We first express $Z_l$ using elements of $Z_0$. To this end, it is convenient to give elements of $Z_l$ different names, in particular, we refer to the elements in $Z_l$ as $\left\{ (x_l^{(i)}, y_l^{(i)}) \right\}_{i=1,\dots,n+1}$ in the following way

$$Z_l = \begin{bmatrix} x_l^{(1)} & \cdots & x_l^{(n)} & x_l^{(n+1)} \\ y_l^{(1)} & \cdots & y_l^{(n)} & y_l^{(n+1)} \end{bmatrix},$$

where we recall that $Z_l \in \mathbb{R}^{(2d+1)\times(n+1)}, x_l^{(i)} \in \mathbb{R}^{2d}, y_l^{(i)} \in \mathbb{R}$. Sometimes it is more convenient to refer to the first half and second half of $x_l^{(i)}$ separately, by, e.g., $\nu_l^{(i)} \in \mathbb{R}^d, \xi_l^{(i)} \in \mathbb{R}^d$, i.e., $x_l^{(i)} = \begin{bmatrix} \nu_l^{(i)} \\ \xi_l^{(i)} \end{bmatrix}$. Then we have

$$Z_l = \begin{bmatrix} \nu_l^{(1)} & \cdots & \nu_l^{(n)} & \nu_l^{(n+1)} \\ \xi_l^{(1)} & \cdots & \xi_l^{(n)} & \xi_l^{(n+1)} \\ y_l^{(1)} & \cdots & y_l^{(n)} & y_l^{(n+1)} \end{bmatrix}.$$

We utilize the shorthands

$$X_l = \begin{bmatrix} x_l^{(1)} & \cdots & x_l^{(n)} \end{bmatrix} \in \mathbb{R}^{2d\times n},$$

$$Y_l = \begin{bmatrix} y_l^{(1)} & \cdots & y_l^{(n)} \end{bmatrix} \in \mathbb{R}^{1\times n}.$$

Then we have

$$Z_l = \begin{bmatrix} X_l & x_l^{(n+1)} \\ Y_l & y_l^{(n+1)} \end{bmatrix}.$$

For the input $Z_0$, we assume $\xi_0^{(n+1)} = 0, y_0^{(n+1)} = 0$ but all other entries of $Z_0$ are arbitrary. We recall our definition of $M$ in (2) and $\left\{ P_l^{\mathrm{TD}}, Q_l^{\mathrm{TD}} \right\}_{l=0,\dots,L-1}$ in (7). In particular, we can express $Q_l^{\mathrm{TD}}$ in a more compact way as

$$M_1 \doteq \begin{bmatrix} -I_d & I_d \\ 0_{d\times d} & 0_{d\times d} \end{bmatrix} \in \mathbb{R}^{2d\times 2d},$$

$$B_l \doteq \begin{bmatrix} C_l^\top & 0_{d\times d} \\ 0_{d\times d} & 0_{d\times d} \end{bmatrix} \in \mathbb{R}^{2d\times 2d},$$

$$A_l \doteq B_l M_1 = \begin{bmatrix} -C_l^\top & C_l^\top \\ 0_{d\times d} & 0_{d\times d} \end{bmatrix} \in \mathbb{R}^{2d\times 2d},$$

$$Q_l^{\mathrm{TD}} \doteq \begin{bmatrix} A_l & 0_{2d\times 1} \\ 0_{1\times 2d} & 0 \end{bmatrix} \in \mathbb{R}^{(2d+1)\times(2d+1)}.$$

We now proceed with the following claims.

**Claim 1.** $X_l \equiv X_0, x_l^{(n+1)} \equiv x_0^{(n+1)}, \forall l$.

Recall that $P_l^{\mathrm{TD}} \doteq \begin{bmatrix} 0_{2d\times 2d} & 0_{2d\times 1} \\ 0_{1\times 2d} & 1 \end{bmatrix} \in \mathbb{R}^{(2d+1)\times(2d+1)}$. Let

$$W_l \doteq Z_l M \left( Z_l^\top Q_l^{\mathrm{TD}} Z_l \right) \in \mathbb{R}^{(2d+1)\times(n+1)}.$$

The embedding evolution can then be expressed as

$$Z_{l+1} = Z_l + \frac{1}{n} P_l^{\mathrm{TD}} W_l.$$

By simple matrix arithmetic, we get

$$P_l^{\mathrm{TD}} W_l = \begin{bmatrix} 0_{2d \times (n+1)} \\ W_l(2d+1) \end{bmatrix},$$

where $W_l(2d+1)$ denotes the $(2d+1)$-th row of $W_l$. Therefore, we have $X_{l+1} = X_l, x_{l+1}^{(n+1)} = x_l^{(n+1)}$. By induction, we get $X_l \equiv X_0$ and $x_l^{(n+1)} \equiv x_0^{(n+1)}$ for all $l = [0, \ldots, L-1]$.

In light of this, we drop all the subscripts of $X_l$, as well as subscripts of $x_l^{(i)}$ for $i = 1, \ldots, n+1$.

**Claim 2.**

$$Y_{l+1} = Y_l + \frac{1}{n} Y_l X^\top A_l X$$

$$y_{l+1}^{(n+1)} = y_l^{(n+1)} + \frac{1}{n} Y_l X^\top A_l x^{(n+1)}.$$

The easier way to show why this claim holds is to factor the embedding evolution into the product of $P_l^{\mathrm{TD}} Z_l M$ and $Z_l^\top Q_l^{\mathrm{TD}} Z_l$. Firstly, we have

$$P_l^{\mathrm{TD}} Z_l = \begin{bmatrix} 0_{2d \times n} & 0_{2d \times 1} \\ Y_l & y_l^{(n+1)} \end{bmatrix}.$$

Applying the mask, we get

$$P_l^{\mathrm{TD}} Z_l M = \begin{bmatrix} 0_{2d \times n} & 0_{2d \times 1} \\ Y_l & 0 \end{bmatrix}.$$

Then, we analyze $Z_l^\top Q_l^{\mathrm{TD}} Z_l$. Applying the block matrix notations, we get

$$\begin{aligned}
Z_l^\top Q_l^{\mathrm{TD}} Z_l &= \begin{bmatrix} X^\top & Y_l^\top \\ x^{(n+1)^\top} & y_l^{(n+1)} \end{bmatrix} \begin{bmatrix} A_l & 0_{2d \times 1} \\ 0_{1 \times 2d} & 0 \end{bmatrix} \begin{bmatrix} X & x^{(n+1)} \\ Y_l & y_l^{(n+1)} \end{bmatrix} \\
&= \begin{bmatrix} X^\top A_l & 0_{n \times 1} \\ x^{(n+1)^\top} A_l & 0 \end{bmatrix} \begin{bmatrix} X & x^{(n+1)} \\ Y_l & y_l^{(n+1)} \end{bmatrix} \\
&= \begin{bmatrix} X^\top A_l X & X^\top A_l x^{(n+1)} \\ x^{(n+1)^\top} A_l X & x^{(n+1)^\top} A_l x^{(n+1)} \end{bmatrix}.
\end{aligned}$$

Combining the two, we get

$$\begin{aligned}
P_l^{\mathrm{TD}} Z_l M \left( Z_l^\top Q_l^{\mathrm{TD}} Z_l \right) &= \begin{bmatrix} 0_{2d \times n} & 0_{2d \times 1} \\ Y_l & 0 \end{bmatrix} \begin{bmatrix} X^\top A_l X & X^\top A_l x^{(n+1)} \\ x^{(n+1)^\top} A_l X & x^{(n+1)^\top} A_l x^{(n+1)} \end{bmatrix} \\
&= \begin{bmatrix} 0_{2d \times n} & 0_{2d \times 1} \\ Y_l X^\top A_l X & Y_l X^\top A_l x^{(n+1)} \end{bmatrix}.
\end{aligned}$$

Hence, according to our update rule in (3), we get

$$Y_{l+1} = Y_l + \frac{1}{n} Y_l X^\top A_l X$$

$$y_{l+1}^{(n+1)} = y_l^{(n+1)} + \frac{1}{n} Y_l X^\top A_l x^{(n+1)}.$$

**Claim 3.**

$$y_{l+1}^{(i)} = y_0^{(i)} + \left\langle M_1 x^{(i)}, \frac{1}{n} \sum_{j=0}^{l} B_j^\top M_2 X Y_j^\top \right\rangle,$$

for $i = 1, \ldots, n+1$, where $M_2 = \begin{bmatrix} I_d & 0_{d \times d} \\ 0_{d \times d} & 0_{d \times d} \end{bmatrix}$.

Following Claim 2, we can unroll $Y_{l+1}$ as

$$Y_{l+1} = Y_l + \frac{1}{n} Y_l X^\top A_l X$$

$$Y_l = Y_{l-1} + \frac{1}{n} Y_{l-1} X^\top A_{l-1} X$$

$$\vdots$$

$$Y_1 = Y_0 + \frac{1}{n} Y_0 X^\top A_0 X.$$

We can then compactly express $Y_{l+1}$ as

$$Y_{l+1} = Y_0 + \frac{1}{n} \sum_{j=0}^{l} Y_j X^\top A_j X.$$

Recall that we define $A_j = B_j M_1$. Then, we can rewrite $Y_{l+1}$ as

$$Y_{l+1} = Y_0 + \frac{1}{n} \sum_{j=0}^{l} Y_j X^\top M_2 B_j M_1 X.$$

The introduction of $M_2$ here does not break the equivalence because $B_j = M_2 B_j$. However, it will help make our proof steps easier to comprehend later.

With the identical procedure, we can easily rewrite $y_{l+1}^{(n+1)}$ as

$$y_{l+1}^{(n+1)} = y_0^{(n+1)} + \frac{1}{n} \sum_{j=0}^{l} Y_j X^\top M_2 B_j M_1 x^{(n+1)}.$$

In light of this, we define $\psi_0 \doteq 0$ and for $l = 0, \ldots$

$$\psi_{l+1} \doteq \frac{1}{n} \sum_{j=0}^{l} B_j^\top M_2 X Y_j^\top \in \mathbb{R}^{2d}. \tag{19}$$

Then we can write

$$y_{l+1}^{(i)} = y_0^{(i)} + \left\langle M_1 x^{(i)}, \psi_{l+1} \right\rangle, \tag{20}$$

for $i = 1, \ldots, n+1$, which is the claim we made. In particular, since we assume $y_0^{(n+1)} = 0$, we have

$$y_{l+1}^{(n+1)} = \left\langle M_1 x^{(n+1)}, \psi_{l+1} \right\rangle.$$

**Claim 4.** The bottom $d$ elements of $\psi_l$ are always 0, i.e., there exists a sequence $\{w_l \in \mathbb{R}^d\}$ such that we can express $\psi_l$ as

$$\psi_l = \begin{bmatrix} w_l \\ 0_{d \times 1} \end{bmatrix}. \tag{21}$$

for all $l = 0, 1, \ldots, L$.

We prove the claim by induction. The base case holds trivially since $\psi_0 \doteq 0$. Suppose that for some $l$, (21) holds. It can be easily verified from the definition of $\psi_{l+1}$ in (19) that

$$\psi_{l+1} = \psi_l + \frac{1}{n} B_l^\top M_2 X Y_l^\top. \tag{22}$$

If we let

$$N_l = \frac{1}{n} M_2 X Y_l^\top \in \mathbb{R}^{2d \times 1},$$

the evolution of $\psi_{l+1}$ can then be compactly expressed as,

$$\psi_{l+1} = \psi_l + B_l^\top N_l.$$

By matrix arithmetic, we have

$$B_l^\top N_l = \begin{bmatrix} C_l^\top & 0_{d \times d} \\ 0_{d \times d} & 0_{d \times d} \end{bmatrix}^\top \begin{bmatrix} N_l(1:d) \\ N_l(d:2d) \end{bmatrix}$$

$$= \begin{bmatrix} C_l N_l(1:d) \\ 0_{d \times 1} \end{bmatrix}$$

where $N_l(1:d) \in \mathbb{R}^d$ and $N_l(d:2d) \in \mathbb{R}^d$ represent the first $d$ and second $d$ elements of $N_l$ respectively. Substituting in our inductive hypothesis into (22), we have:

$$\psi_{l+1} = \begin{bmatrix} w_l \\ 0_{d \times 1} \end{bmatrix} + \begin{bmatrix} C_l N_l(1:d) \\ 0_{d \times 1} \end{bmatrix},$$

$$= \begin{bmatrix} w_l + C_l N_l(1:d) \\ 0_{d \times 1} \end{bmatrix}$$

if we let $w_{l+1} = w_l + C_l N_l(1:d)$, we can see that the property holds for $\psi_{l+1}$, thereby verifying Claim 4.

Given all the claims above, we can then compute that

$$\left\langle \psi_{l+1}, M_1 x^{(n+1)} \right\rangle$$

$$= \left\langle \psi_l, M_1 x^{(n+1)} \right\rangle + \frac{1}{n} \left\langle B_l^\top M_2 X Y_l^\top, M_1 x^{(n+1)} \right\rangle \qquad \text{(By (22))}$$

$$= \left\langle \psi_l, M_1 x^{(n+1)} \right\rangle + \frac{1}{n} \sum_{i=1}^n \left\langle B_l^\top M_2 x^{(i)} y_l^{(i)}, M_1 x^{(n+1)} \right\rangle$$

$$= \left\langle \psi_l, M_1 x^{(n+1)} \right\rangle + \frac{1}{n} \sum_{i=1}^n \left\langle B_l^\top M_2 x^{(i)} \left( \left\langle \psi_l, M_1 x^{(i)} \right\rangle + y_0^{(i)} \right), M_1 x^{(n+1)} \right\rangle \qquad \text{(By (20))}$$

$$= \left\langle \psi_l, M_1 x^{(n+1)} \right\rangle + \frac{1}{n} \sum_{i=1}^n \left\langle B_l^\top \begin{bmatrix} \nu^{(i)} \\ 0_{d \times 1} \end{bmatrix} \left( \left\langle \psi_l, \begin{bmatrix} -\nu^{(i)} + \xi^{(i)} \\ 0_{d \times 1} \end{bmatrix} \right\rangle + y_0^{(i)} \right), M_1 x^{(n+1)} \right\rangle$$

$$= \left\langle \psi_l, M_1 x^{(n+1)} \right\rangle + \frac{1}{n} \sum_{i=1}^n \left\langle \begin{bmatrix} C_l \nu^{(i)} \\ 0_{d \times 1} \end{bmatrix} \left( y_0^{(i)} + w_l^\top \xi^{(i)} - w_l^\top \nu^{(i)} \right), M_1 x^{(n+1)} \right\rangle \qquad \text{(By Claim 4)}$$

$$= \left\langle \psi_l, M_1 x^{(n+1)} \right\rangle + \frac{1}{n} \sum_{i=1}^n \left\langle \begin{bmatrix} C_l \nu^{(i)} \left( y_0^{(i)} + w_l^\top \xi^{(i)} - w_l^\top \nu^{(i)} \right) \\ 0_{d \times 1} \end{bmatrix}, M_1 x^{(n+1)} \right\rangle$$

This means

$$\left\langle w_{l+1}, \nu^{(n+1)} \right\rangle = \left\langle w_l, \nu^{(n+1)} \right\rangle + \frac{1}{n} \sum_{i=1}^n \left\langle C_l \nu^{(i)} \left( y_0^{(i)} + w_l^\top \xi^{(i)} - w_l^\top \nu^{(i)} \right), \nu^{(n+1)} \right\rangle.$$

Since the choice of the query $\nu^{(n+1)}$ is arbitrary, we get

$$w_{l+1} = w_l + \frac{1}{n} \sum_{i=1}^n C_l \left( y_0^{(i)} + w_l^\top \xi^{(i)} - w_l^\top \nu^{(i)} \right) \nu^{(i)}.$$

In particular, when we construct $Z_0$ such that $\nu^{(i)} = \phi_{i-1}$, $\xi^{(i)} = \gamma \phi_i$ and $y_0^{(i)} = R_i$, we get

$$w_{l+1} = w_l + \frac{1}{n} \sum_{i=1}^n C_l \left( R_i + \gamma w_l^\top \phi_i - w_l^\top \phi_{i-1} \right) \phi_{i-1}$$

which is the update rule for pre-conditioned TD learning. We also have

$$y_l^{(n+1)} = \left\langle \psi_l, M_1 x^{(n+1)} \right\rangle = -\left\langle w_l, \phi^{(n+1)} \right\rangle.$$

This concludes our proof. $\qquad\qquad\square$

## A.2    Proof of Corollary 1

*Proof.* The proof presented here closely mirrors the methodology and notation established in Theorem 1. Since we are only considering a 1-layer transformer in this Corollary, we can recall the embedding evolution from (3) and write

$$Z_1 = Z_0 + \frac{1}{n} P_0 Z_0 M(Z_0^\top Q_0 Z_0).$$

We once again refer to the elements in $Z_l$ as $\left\{ (x_l^{(i)}, y_l^{(i)}) \right\}_{i=1,\dots,n+1}$ in the following way

$$Z_l = \begin{bmatrix} x_l^{(1)} & \dots & x_l^{(n)} & x_l^{(n+1)} \\ y_l^{(1)} & \dots & y_l^{(n)} & y_l^{(n+1)} \end{bmatrix},$$

where we recall that $Z_l \in \mathbb{R}^{(2d+1)\times(n+1)}, x_l^{(i)} \in \mathbb{R}^{2d}, y_l^{(i)} \in \mathbb{R}$. We utilize, $\nu_l^{(i)} \in \mathbb{R}^d, \xi_l^{(i)} \in \mathbb{R}^d$, to refer to the first half and second half of $x_l^{(i)}$ i.e., $x_l^{(i)} = \begin{bmatrix} \nu_l^{(i)} \\ \xi_l^{(i)} \end{bmatrix}$. Then we have

$$Z_l = \begin{bmatrix} \nu_l^{(1)} & \dots & \nu_l^{(n)} & \nu_l^{(n+1)} \\ \xi_l^{(1)} & \dots & \xi_l^{(n)} & \xi_l^{(n+1)} \\ y_l^{(1)} & \dots & y_l^{(n)} & y_l^{(n+1)} \end{bmatrix}.$$

We further define as shorthands

$$X_l = \begin{bmatrix} x_l^{(1)} & \dots & x_l^{(n)} \end{bmatrix} \in \mathbb{R}^{2d\times n}, \; Y_l = \begin{bmatrix} y_l^{(1)} & \dots & y_l^{(n)} \end{bmatrix} \in \mathbb{R}^{1\times n}.$$

Then the block-wise structure of $Z_l$ can be succinctly expressed as:

$$Z_l = \begin{bmatrix} X_l & x_l^{(n+1)} \\ Y_l & y_l^{(n+1)} \end{bmatrix}.$$

For the input $Z_0$, we assume $\xi_0^{(n+1)} = 0, y_0^{(n+1)} = 0$ but all other entries of $Z_0$ are arbitrary. We recall our definition of $M$ in (2) and $\{P_0, Q_0\}$ in (7). In particular, we can express $Q_0$ in a more compact way as

$$M_1 \doteq \begin{bmatrix} -I_d & 0_{d\times d} \\ 0_{d\times d} & 0_{d\times d} \end{bmatrix} \in \mathbb{R}^{2d\times 2d}, \; B_0 \doteq \begin{bmatrix} C_0^\top & 0_{d\times d} \\ 0_{d\times d} & 0_{d\times d} \end{bmatrix} \in \mathbb{R}^{2d\times 2d},$$

$$A_0 \doteq B_0 M_1 = \begin{bmatrix} -C_0^\top & 0_{d\times d} \\ 0_{d\times d} & 0_{d\times d} \end{bmatrix} \in \mathbb{R}^{2d\times 2d},$$

$$Q_0 \doteq \begin{bmatrix} A_0 & 0_{2d\times 1} \\ 0_{1\times 2d} & 0 \end{bmatrix} \in \mathbb{R}^{(2d+1)\times(2d+1)}.$$

We will proceed with the following claims.

**Claim 1.** $X_1 \equiv X_0, x_1^{(n+1)} \equiv x_0^{(n+1)}$

Because we are considering the special case of $L = 1$ and because we utilize the same definition of $P_0$ as in Theorem 1, the argument proving Claim 1 in Theorem 1 holds here as well. As a result, we drop all the subscripts of $X_1$, as well as subscripts of $x_1^{(i)}$ for $i = 1, \dots, n+1$.

**Claim 2.**

$$Y_1 = Y_0 + \frac{1}{n} Y_0 X^\top A_0 X$$

$$y_1^{(n+1)} = y_0^{(n+1)} + \frac{1}{n} Y_0 X^\top A_0 x^{(n+1)}.$$

This claim is a special case of Claim 2 from the proof of Theorem 1 in Appendix A.1, where $L = 1$. Our block-wise construction of $Q_0$ matches that in the proof of Theorem 1. Although our $A_0$ here differs from the specific form of $A_0$ in the proof of Theorem 1, this specific form is not utilized in the proof of Claim 2. Therefore, the proof of Claim 2 in Appendix A.1 applies here, and we omit the steps to avoid redundancy.

**Claim 3.**

$$y_1^{(i)} = y_0^{(i)} + \left\langle M_1 x^{(i)}, \frac{1}{n} B_0^\top M_2 X Y_0^\top \right\rangle,$$

for $i = 1, \ldots, n+1$, where $M_2 = \begin{bmatrix} I_d & 0_{d \times d} \\ 0_{d \times d} & 0_{d \times d} \end{bmatrix}$.

This claim once again is the $L = 1$ case of Claim 3 from the proof of Theorem 1 in Appendix A.1. The specific form of $M_1$ is not utilized in the proof of Claim 3 from Appendix A.1, so it applies here.

We can then define $\psi_0 \doteq 0$ and,

$$\psi_1 \doteq \frac{1}{n} B_0^\top M_2 X Y_0^\top \in \mathbb{R}^{2d}. \tag{23}$$

Then we can write

$$y_1^{(i)} = y_0^{(i)} + \left\langle M_1 x^{(i)}, \psi_1 \right\rangle,$$

for $i = 1, \ldots, n+1$, which is the claim we made. In particular, since we assume $y_0^{(n+1)} = 0$, we have

$$y_1^{(n+1)} = \left\langle M_1 x^{(n+1)}, \psi_1 \right\rangle.$$

**Claim 4.** The bottom $d$ elements of $\psi_1$ are always 0, i.e., there exists $w_1 \in \mathbb{R}^d$ such that we can express $\psi_1$ as

$$\psi_1 = \begin{bmatrix} w_1 \\ 0_{d \times 1} \end{bmatrix}.$$

Since our $B_0$ here is identical to that in the proof of Theorem 1 in A.1, Claim 4 holds for the same reason. We therefore omit the proof details to avoid repetition.

Given all the claims above, we can then compute that

$$
\begin{aligned}
\left\langle \psi_1, M_1 x^{(n+1)} \right\rangle &= \frac{1}{n} \left\langle B_0^\top M_2 X Y_0^\top, M_1 x^{(n+1)} \right\rangle && \text{(By (23))} \\
&= \frac{1}{n} \sum_{i=1}^n \left\langle B_0^\top M_2 x^{(i)} y_0^{(i)}, M_1 x^{(n+1)} \right\rangle \\
&= \frac{1}{n} \sum_{i=1}^n \left\langle B_0^\top \begin{bmatrix} \nu^{(i)} \\ 0_{d \times 1} \end{bmatrix} \left( y_0^{(i)} \right), M_1 x^{(n+1)} \right\rangle \\
&= \frac{1}{n} \sum_{i=1}^n \left\langle \begin{bmatrix} C_0 \nu^{(i)} \\ 0_{d \times 1} \end{bmatrix} \left( y_0^{(i)} \right), M_1 x^{(n+1)} \right\rangle && \text{(By Claim 4)} \\
&= \frac{1}{n} \sum_{i=1}^n \left\langle \begin{bmatrix} C_0 \nu^{(i)} y_0^{(i)} \\ 0_{d \times 1} \end{bmatrix}, M_1 x^{(n+1)} \right\rangle
\end{aligned}
$$

This means

$$\left\langle w_1, \nu^{(n+1)} \right\rangle = \frac{1}{n} \sum_{i=1}^n \left\langle C_0 \nu^{(i)} y_0^{(i)}, \nu^{(n+1)} \right\rangle.$$

Since the choice of the query $\nu^{(n+1)}$ is arbitrary, we get

$$w_1 = \frac{1}{n}\sum_{i=1}^{n} C_0 y_0^{(i)} \nu^{(i)}.$$

In particular, when we construct $Z_0$ such that $\nu^{(i)} = \phi_{i-1}$ and $y_0^{(i)} = R_i$, we get

$$w_1 = \frac{1}{n}\sum_{i=1}^{n} C_0 R_i \phi_{i-1}$$

which is the update rule for a single step of TD(0) with $w_0 = 0$. We also have

$$y_1^{(n+1)} = \left\langle \psi_1, M_1 x^{(n+1)} \right\rangle = -\left\langle w_1, \phi^{(n+1)} \right\rangle.$$

This concludes our proof. $\qquad\square$

### A.3 PROOF OF THEOREM 2

**Preliminaries**  Before we present the proof, we first introduce notations convenient for our analysis. We decompose $P_0$ and $Q_0$ as

$$P_0 = \begin{bmatrix} P \in \mathbb{R}^{2d\times(2d+1)} \\ p \in \mathbb{R}^{1\times(2d+1)} \end{bmatrix}, Q_0 = \begin{bmatrix} Q_a \in \mathbb{R}^{d\times d} & Q_b \in \mathbb{R}^{d\times d} & q_c \in \mathbb{R}^{d\times 1} \\ Q'_a \in \mathbb{R}^{d\times d} & Q'_b \in \mathbb{R}^{d\times d} & q'_c \in \mathbb{R}^{d\times 1} \\ q_a \in \mathbb{R}^{1\times d} & q_b \in \mathbb{R}^{1\times d} & q''_c \in \mathbb{R} \end{bmatrix}.$$

One can readily check that $\mathrm{TF}_1$ is independent of $P, Q_b, Q'_b, q_b, q_c, q'_c, q''_c$. Thus, we can assume that these matrices are zero. Let $z^{(i)}$ be the $i$-th column of $Z_0$. Indeed, $\mathrm{TF}_1$ can be written as

$$\mathrm{TF}_1(Z_0, \{P_0, Q_0\}) = -Z_1[2d+1, n+1] \qquad\qquad \text{(By (4))}$$

$$= -\frac{1}{n}p^\top \left(\sum_{i=1}^{n} z^{(i)} z^{(i)\top}\right) Q_0 z^{(n+1)}$$

$$= -\frac{1}{n}\sum_{i=1}^{n} \left\langle p, z^{(i)} \right\rangle z^{(i)\top} Q_0 z^{(n+1)}$$

$$= -\frac{1}{n}\sum_{i=1}^{n} \left\langle p, z^{(i)} \right\rangle \left(\phi_{i-1}^\top Q_a \phi_{n+1} + \gamma \phi_i^\top Q'_a \phi_{n+1} + R_i \phi_{n+1}^\top q_a\right) \quad (24)$$

$$= -\frac{1}{n}\sum_{i=1}^{n} \left(\underbrace{\left\langle p_{[1:d]}, \phi_{i-1} \right\rangle + \gamma\left\langle p_{[d+1:2d]}, \phi_i \right\rangle + p_{[2d+1]} R_i}_{\alpha_i(Z_0, P_0)}\right)$$

$$\cdot \left(\underbrace{\phi_{i-1}^\top Q_a \phi_{n+1} + \gamma(\phi_i)^\top Q'_a \phi_{n+1} + R_i \phi_{n+1}^\top q_a}_{\beta_i(Z_0, Q_0)}\right).$$

We prepare the following gradient computations for future use:

$$\nabla_{p_{[1:d]}} \mathrm{TF}_1(Z_0, \{P_0, Q_0\}) = -\frac{1}{n}\sum_{i=1}^{n} \beta_i(Z_0, Q_0)\phi_{i-1}$$

$$\nabla_{p_{[d+1:2d]}} \mathrm{TF}_1(Z_0, \{P_0, Q_0\}) = -\frac{\gamma}{n}\sum_{i=1}^{n} \beta_i(Z_0, Q_0)\phi_i$$

$$\nabla_{Q_a} \mathrm{TF}_1(Z_0, \{P_0, Q_0\}) = -\frac{1}{n}\sum_{i=1}^{n} \alpha_i(Z_0, P_0)\phi_{i-1}\phi_{n+1}^\top \qquad (25)$$

$$\nabla_{Q'_a} \mathrm{TF}_1(Z_0, \{P_0, Q_0\}) = -\frac{\gamma}{n}\sum_{i=1}^{n} \alpha_i(Z_0, P_0)\phi_i\phi_{n+1}^\top$$

$$\nabla_{q_a} \mathrm{TF}_1(Z_0, \{P_0, Q_0\}) = -\frac{1}{n}\sum_{i=1}^{n} R_i\alpha_i(Z_0, P_0)\phi_{n+1}.$$

We will also reference the following two lemmas in our main proof.

**Lemma A.3.1.** *Let $\Lambda$ be a diagonal matrix whose diagonal elements are i.i.d Rademacher random variables [5] $\zeta_1, \ldots \zeta_d$. For any matrix $K \in \mathbb{R}^{d \times d}$, we have that $\mathbb{E}_\Lambda[\Lambda K \Lambda] = diag(K)$.*

*Proof.* First, we can write $\Lambda K \Lambda$ explicitly as

$$\Lambda K \Lambda = \begin{bmatrix} \zeta_1 & 0 & \ldots & 0 \\ 0 & \zeta_2 & \ldots & 0 \\ \vdots & \vdots & \ddots & \vdots \\ 0 & 0 & \ldots & \zeta_d \end{bmatrix} \begin{bmatrix} k_{11} & k_{12} & \ldots & k_{1d} \\ k_{21} & k_{22} & \ldots & k_{2d} \\ \vdots & \vdots & \ddots & \vdots \\ k_{d1} & k_{d2} & \ldots & k_{dd} \end{bmatrix} \begin{bmatrix} \zeta_1 & 0 & \ldots & 0 \\ 0 & \zeta_2 & \ldots & 0 \\ \vdots & \vdots & \ddots & \vdots \\ 0 & 0 & \ldots & \zeta_d \end{bmatrix}.$$

Using $(\Lambda K \Lambda)_{ij}$ to denote the element in the $i$-th row at column $j$ of $\Lambda K \Lambda$, from elementary matrix multiplication we have

$$(\Lambda K \Lambda)_{ij} = \zeta_i k_{ij} \zeta_j.$$

When $i \neq j$, $\mathbb{E}[\zeta_i \zeta_j] = \mathbb{E}[\zeta_i]\mathbb{E}[\zeta_j] = 0$ becasue $\zeta_i$ and $\zeta_j$ are independent. For $i = j$, $\mathbb{E}[\zeta_i \zeta_j] = \mathbb{E}[\zeta_i^2] = 1$. We can then compute the expectation

$$\mathbb{E}_\Lambda[(\Lambda K \Lambda)]_{ij} = \begin{cases} k_{ij} & i = j \\ 0 & i \neq j. \end{cases}$$

Consequently,

$$\mathbb{E}_\Lambda[\Lambda K \Lambda] = \mathrm{diag}(K).$$

$\square$

**Lemma A.3.2.** *Let $\Pi \in \mathbb{R}^{d \times d}$ be a random permutation matrix uniformly distributed over all $d \times d$ permutation matrices and $L \in \mathbb{R}^{d \times d}$ be a diagonal matrix. Then, it holds that*

$$\mathbb{E}_\Pi[\Pi L \Pi^\top] = \frac{1}{d} \operatorname{tr}(L) I_d.$$

*Proof.* By definition,

$$[\Pi L \Pi^\top]_{ij} = \sum_{k=1}^d \Pi_{ik} L_{kk} \Pi_{jk}.$$

We note that each row of $\Pi$ is a standard basis. Given the orthogonality of standard bases, we get

$$[\Pi L \Pi^\top]_{ij} = \begin{cases} 0 & i \neq j, \\ L_{q_i q_i} & i = j, \end{cases}$$

where $q_i$ is the unique index such that $\Pi_{iq_i} = 1$. If the distribution of $\Pi$ is uniform, then $[\Pi L \Pi^\top]_{ii}$ is equal to one of $L_{11}, \ldots, L_{dd}$ with the same probability. Thus, the expected value $[\Pi L \Pi^\top]_{ii}$ is $\frac{1}{d} \operatorname{tr}(L)$. $\square$

Now, we start with the proof of the theorem statement.

*Proof.* We recall the definition of the set $\Theta^*$ as

$$\Theta^* \doteq \cup_{\eta, c, c' \in \mathbb{R}} \left\{ P = \begin{bmatrix} 0_{2d \times 2d} & 0_{2d \times 1} \\ 0_{1 \times 2d} & \eta \end{bmatrix}, Q = \begin{bmatrix} cI_d & 0_{d \times d} & 0_{d \times 1} \\ c'I_d & 0_{d \times d} & 0_{d \times 1} \\ 0_{1 \times d} & 0_{1 \times d} & 0 \end{bmatrix} \right\}.$$

Suppose $\theta_k \in \Theta^*$, then by (24) and (25), we get

$$\mathrm{TF}_1(Z_0, \theta_k) = -\frac{\eta_k}{n} \sum_{i=1}^n R_i \big( c_k \phi_{i-1}^\top \phi_{n+1} + c'_k \gamma \phi_i^\top \phi_{n+1} \big) \tag{26}$$

---

[5] A Rademacher random variable takes values $1$ or $-1$, each with an equal probability of $0.5$.

$$\mathrm{TF}_1(Z_0', \theta_k) = -\frac{\eta_k}{n} \sum_{i=1}^{n} R_{i+1} \big(c_k \phi_i^\top \phi_{n+2} + c_k' \gamma \phi_{i+1}^\top \phi_{n+2}\big)$$

$$\nabla_{p_{[1:d]}} \mathrm{TF}_1(Z_0, \theta_k) = -\frac{1}{n} \sum_{i=1}^{n} \big(c_k \phi_{i-1}^\top \phi_{n+1} + c_k' \gamma \phi_i^\top \phi_{n+1}\big) \phi_{i-1}$$

$$\nabla_{p_{[d+1:2d]}} \mathrm{TF}_1(Z_0, \theta_k) = -\frac{\gamma}{n} \sum_{i=1}^{n} \big(c_k \phi_{i-1}^\top \phi_{n+1} + c_k' \gamma \phi_i^\top \phi_{n+1}\big) \phi_i$$

$$\nabla_{Q_a} \mathrm{TF}_1(Z_0, \theta_k) = -\frac{\eta_k}{n} \sum_{i=1}^{n} R_i \phi_{i-1} \phi_{n+1}^\top$$

$$\nabla_{Q_a'} \mathrm{TF}_1(Z_0, \theta_k) = -\frac{\gamma \eta_k}{n} \sum_{i=1}^{n} R_i \phi_i \phi_{n+1}^\top$$

$$\nabla_{q_a} \mathrm{TF}_1(Z_0, \theta_k) = -\frac{\eta_k}{n} \sum_{i=1}^{n} R_i^2 \phi_{n+1}$$

Recall the definition of $\Delta(\theta)$ in (10). With a slight abuse of notation, we define $\Delta(p_{[1:d]})$ to be the $p_{[1:d]}$ component of $\Delta(\theta)$, i.e.,

$$\Delta(p_{[1:d]}) \doteq \mathbb{E}\left[ (R + \gamma \mathrm{TF}_1(Z_0', \theta) - \mathrm{TF}_1(Z_0, \theta)) \frac{\partial \mathrm{TF}_1(Z_0, \theta)}{\partial p_{[1:d]}} \right].$$

Same goes for $\Delta(p_{[d+1:2d]}), \Delta(Q_a), \Delta(Q_a')$, and $\Delta(q_a)$.

We will prove that

(a) $\Delta(p_{[1:d]}) = \Delta(p_{[d+1:2d]}) = \Delta(q_a) = 0$ for $\Delta(\theta_k)$;

(b) $\Delta(Q_a) = \delta I_d$ and $\Delta(Q_a') = \delta' I_d$ for some $\delta, \delta' \in \mathbb{R}$ for $\Delta(\theta_k)$

using Assumptions 5.1 and 5.2. We can see that the combination of (a) and (b) are sufficient for proving the theorem. Recall that $Z_0$ and $Z_0'$ are sampled from $(p_0, p, r, \phi)$. We make the following claims to assist our proof of (a) and (b).

**Claim 1.** Let $\zeta$ be a Rademacher random variable. We denote $Z_\zeta$ and $Z_\zeta'$ as the prompts sampled from $(p_0, p, r, \zeta\phi)$. We then have $Z_0 \triangleq Z_\zeta$ and $Z_0' \triangleq Z_\zeta'$. To show this is true, we notice that for any realization of $\zeta$, denoted as $\bar\zeta \in \{1, -1\}$, we have

$$\begin{aligned}
\Pr(p_0, p, r, \phi) &= \Pr(p_0, p, r) \Pr(\phi) && \text{(Assumption 5.1)} \\
&= \Pr(p_0, p, r) \Pr(\bar\zeta I_d \phi) && \text{(Assumption 5.2)} \\
&= \Pr(p_0, p, r, \bar\zeta\phi). && \text{(Assumption 5.1)}
\end{aligned}$$

It then follows that

$$\begin{aligned}
\Pr(p_0, p, r, \phi) &= \Pr(p_0, p, r, \phi) \sum_{\bar\zeta \in \{1, -1\}} \Pr(\zeta = \bar\zeta) \\
&= \sum_{\bar\zeta \in \{1, -1\}} \Pr(p_0, p, r, \phi) \Pr(\zeta = \bar\zeta) \\
&= \sum_{\bar\zeta \in \{1, -1\}} \Pr(p_0, p, r, \bar\zeta\phi) \Pr(\zeta = \bar\zeta) \\
&= \Pr(p_0, p, r, \zeta\phi).
\end{aligned}$$

This implies Claim 1 holds.

**Claim 2.** Define $\Lambda$ as the diagonal matrix whose diagonal elements are i.i.d. Rademacher random variables $\zeta_1, \ldots, \zeta_d$. We denote $Z_\Lambda$ and $Z_\Lambda'$ as the prompts sampled from $(p_0, p, r, \Lambda\phi)$, where $\Lambda\phi$

means $[\Lambda\phi(s)]_{s\in\mathcal{S}}$. We then have $Z_0 \triangleq Z_\Lambda$ and $Z_0' \triangleq Z_\Lambda'$. The proof follows the same procedures as Claim 1.

**Claim 3.** Let $\Pi$ be a random permutation matrix uniformly distributed over all $d \times d$ permutation matrices. We denote $Z_\Pi$ and $Z_\Pi'$ as the prompts sampled from $(p_0, p, r, \Pi\phi)$, where $\Pi\phi$ means $[\Pi\phi(s)]_{s\in\mathcal{S}}$. We then have $Z_0 \triangleq Z_\Pi$ and $Z_0' \triangleq Z_\Pi'$. The proof follows the same procedures as Claim 1.

**Proof of (a) using Claim 1**  It is easy to check by (26) that

$$
\begin{aligned}
\mathrm{TF}_1(Z_\zeta, \theta_k) &= -\frac{\eta_k}{n} \sum_{i=1}^n R_i\big(c_k\zeta^2\phi_{i-1}^\top\phi_{n+1} + c_k'\gamma\zeta^2\phi_i^\top\phi_{n+1}\big) \\
&= \underbrace{\zeta^2}_{=1} \mathrm{TF}_1(Z_0, \theta_k) \\
&= \mathrm{TF}_1(Z_0, \theta_k).
\end{aligned}
\tag{27}
$$

Similarly, one can check that $\mathrm{TF}_1(Z_\zeta', \theta_k) = \mathrm{TF}_1(Z_0', \theta_k)$.

Furthermore,

$$
\begin{aligned}
\nabla_{p_{[1:d]}}\mathrm{TF}_1(Z_\zeta, \theta_k) &= -\frac{1}{n} \sum_{i=1}^n \left( c_k \underbrace{\zeta^2}_{=1} \phi_{i-1}^\top\phi_{n+1} + c_k'\gamma \underbrace{\zeta^2}_{=1} \phi_i^\top\phi_{n+1} \right) \zeta\phi_{i-1} \\
&= -\frac{\zeta}{n} \sum_{i=1}^n \big(c_k\phi_{i-1}^\top\phi_{n+1} + c_k'\gamma\phi_i^\top\phi_{n+1}\big)\phi_{i-1} \\
&= \zeta\nabla_{p_{[1:d]}}\mathrm{TF}_1(Z_0, \theta_k).
\end{aligned}
\tag{28}
$$

Then, from (10), we get

$$
\begin{aligned}
&\Delta(p_{[1:d]}) \\
&= \mathbb{E}\big[\big(R_{n+2} + \gamma\mathrm{TF}_1(Z_0', \theta_k) - \mathrm{TF}_1(Z_0, \theta_k)\big)\nabla_{p_{[1:d]}}\mathrm{TF}_1(Z_0, \theta_k)\big] \\
&= \mathbb{E}\big[\big(R_{n+2} + \gamma\mathrm{TF}_1(Z_0', \theta_k) - \mathrm{TF}_1(Z_\zeta, \theta_k)\big)\nabla_{p_{[1:d]}}\mathrm{TF}_1(Z_\zeta, \theta_k)\big] && \text{(By Claim 1)} \\
&= \mathbb{E}_\zeta\big[\mathbb{E}\big[\big(R_{n+2} + \gamma\mathrm{TF}_1(Z_\zeta', \theta_k) - \mathrm{TF}_1(Z_\zeta, \theta_k)\big)\nabla_{p_{[1:d]}}\mathrm{TF}_1(Z_\zeta, \theta_k) \mid \zeta\big]\big] \\
&= \mathbb{E}_\zeta\big[\mathbb{E}\big[\big(R_{n+2} + \gamma\mathrm{TF}_1(Z_0', \theta_k) - \mathrm{TF}_1(Z_0, \theta_k)\big)\zeta\nabla_{p_{[1:d]}}\mathrm{TF}_1(Z_0, \theta_k) \mid \zeta\big]\big] && \text{(By (27), (28))} \\
&= \mathbb{E}_\zeta\big[\zeta\mathbb{E}\big[\big(R_{n+2} + \gamma\mathrm{TF}_1(Z_0', \theta_k) - \mathrm{TF}_1(Z_0, \theta_k)\big)\nabla_{p_{[1:d]}}\mathrm{TF}_1(Z_0, \theta_k) \mid \zeta\big]\big] \\
&= \mathbb{E}_\zeta\big[\zeta\mathbb{E}\big[\big(R_{n+2} + \gamma\mathrm{TF}_1(Z_0', \theta_k) - \mathrm{TF}_1(Z_0, \theta_k)\big)\nabla_{p_{[1:d]}}\mathrm{TF}_1(Z_0, \theta_k)\big]\big] \\
&= \mathbb{E}_\zeta[\zeta]\mathbb{E}\big[\big(R_{n+2} + \gamma\mathrm{TF}_1(Z_0', \theta_k) - \mathrm{TF}_1(Z_0, \theta_k)\big)\nabla_{p_{[1:d]}}\mathrm{TF}_1(Z_0, \theta_k)\big] \\
&= 0.
\end{aligned}
$$

The proof is analogous for $\Delta(p_{[d+1:2d]}) = 0$, and $\Delta(q_a) = 0$.

**Proof of (b) using Claims 2 and 3**  We first show that $\Delta(Q_a)$ is a diagonal matrix. Similar to (a), we have

$$
\begin{aligned}
\mathrm{TF}_1(Z_\Lambda, \theta_k) &= -\frac{1}{n} \sum_{i=1}^n \eta_k R_i\left( c_k\phi_{i-1}^\top \underbrace{\Lambda^2}_{=I} \phi_{n+1} + c_k'\gamma\phi_i^\top \underbrace{\Lambda^2}_{=I} \phi_{n+1} \right) \\
&= \mathrm{TF}_1(Z_0, \theta_k).
\end{aligned}
\tag{29}
$$

Similarly, we get $\mathrm{TF}_1(Z_\Lambda', \theta_k) = \mathrm{TF}_1(Z_0', \theta_k)$. Additionally, we have

$$
\nabla_{Q_a}\mathrm{TF}_1(Z_\Lambda, \theta_k) = -\frac{1}{n} \sum_{i=1}^n \eta_k R_i \Lambda\phi_{i-1}\phi_{n+1}^\top\Lambda^\top = \Lambda\nabla_{Q_a}\mathrm{TF}_1(Z_0, \theta_k)\Lambda.
\tag{30}
$$

By (10) again, we get

$$
\Delta(Q_a)
$$

$$
\begin{aligned}
&=\mathbb{E}[(R_{n+2} + \gamma \mathrm{TF}_1(Z'_0, \theta_k) - \mathrm{TF}_1(Z_0, \theta_k)) \nabla_{Q_a} \mathrm{TF}_1(Z_0, \theta_k)] \\
&=\mathbb{E}[(R_{n+2} + \gamma \mathrm{TF}_1(Z'_\Lambda, \theta_k) - \mathrm{TF}_1(Z_\Lambda, \theta_k)) \nabla_{Q_a} \mathrm{TF}_1(Z_\Lambda, \theta_k)] && \text{(By Claim 2)} \\
&=\mathbb{E}_\Lambda[\mathbb{E}[(R_{n+2} + \gamma \mathrm{TF}_1(Z'_\Lambda, \theta_k) - \mathrm{TF}_1(Z_\Lambda, \theta_k)) \nabla_{Q_a} \mathrm{TF}_1(Z_\Lambda, \theta_k) \mid \Lambda]] \\
&=\mathbb{E}_\Lambda[\mathbb{E}[(R_{n+2} + \gamma \mathrm{TF}_1(Z'_0, \theta_k) - \mathrm{TF}_1(Z_0, \theta_k)) \Lambda \nabla_{Q_a} \mathrm{TF}_1(Z_0, \theta_k) \Lambda \mid \Lambda]] && \text{(By (29), (30))} \\
&=\mathbb{E}_\Lambda[\Lambda \mathbb{E}[(R_{n+2} + \gamma \mathrm{TF}_1(Z'_0, \theta_k) - \mathrm{TF}_1(Z_0, \theta_k)) \nabla_{Q_a} \mathrm{TF}_1(Z_0, \theta_k) \mid \Lambda] \Lambda] \\
&=\mathbb{E}_\Lambda[\Lambda \mathbb{E}[(R_{n+2} + \gamma \mathrm{TF}_1(Z'_0, \theta_k) - \mathrm{TF}_1(Z_0, \theta_k)) \nabla_{Q_a} \mathrm{TF}_1(Z_0, \theta_k)] \Lambda] \\
&=\mathrm{diag}(\mathbb{E}[(R_{n+2} + \gamma \mathrm{TF}_1(Z'_0, \theta_k) - \mathrm{TF}_1(Z_0, \theta_k)) \nabla_{Q_a} \mathrm{TF}_1(Z_0, \theta_k)]) && \text{(By Lemma A.3.1)} \\
&=\mathrm{diag}(\Delta(Q_a)).
\end{aligned}
$$

The last equation holds if and only if $\Delta(Q_a)$ is diagonal. We have proven this claim.

Now, we prove that $\Delta(Q_a) = \delta I_d$ for some $\delta \in \mathbb{R}$ using Claim 3 and Lemma A.3.2. Let $\Pi$ be a random permutation matrix uniformly distributed over all permutation matrices. Recall the definition of $Z_\Pi$ and $Z'_\Pi$ in Claim 3. We have

$$
\mathrm{TF}_1(Z_\Pi, \theta_k) = -\frac{1}{n} \sum_{i=1}^n \eta_k R_i \left( c_k \phi_{i-1}^\top \underbrace{\Pi^\top \Pi}_{=I} \phi_{n+1} + c'_k \gamma \phi_i^\top \underbrace{\Pi^\top \Pi}_{=I} \phi_{n+1} \right) = \mathrm{TF}_1(Z_0, \theta_k). \tag{31}
$$

Analogously, we get $\mathrm{TF}_1(Z'_\Pi, \theta_k) = \mathrm{TF}_1(Z'_0, \theta_k)$. Furthermore, we have

$$
\nabla_{Q_a} \mathrm{TF}_1(Z_\Pi, \theta_k) = -\frac{1}{n} \sum_{i=1}^n \eta_k R_i \Pi \phi_{i-1} \phi_{n+1}^\top \Pi^\top = \Pi \nabla_{Q_a} \mathrm{TF}_1(Z_0, \theta_k) \Pi^\top. \tag{32}
$$

By (10), we are ready to show that

$$
\begin{aligned}
&\Delta(Q_a) \\
&=\mathbb{E}[(R_{n+2} + \gamma \mathrm{TF}_1(Z'_0, \theta_k) - \mathrm{TF}_1(Z_0, \theta_k)) \nabla_{Q_a} \mathrm{TF}_1(Z_0, \theta_k)] \\
&=\mathbb{E}[(R_{n+2} + \gamma \mathrm{TF}_1(Z'_\Pi, \theta_k) - \mathrm{TF}_1(Z_\Pi, \theta_k)) \nabla_{Q_a} \mathrm{TF}_1(Z_\Pi, \theta_k)] && \text{(By Claim 3)} \\
&=\mathbb{E}_\Pi[\mathbb{E}[(R_{n+2} + \gamma \mathrm{TF}_1(Z'_\Pi, \theta_k) - \mathrm{TF}_1(Z_\Pi, \theta_k)) \nabla_{Q_a} \mathrm{TF}_1(Z_\Pi, \theta_k) \mid \Pi]] \\
&=\mathbb{E}_\Pi\left[\mathbb{E}[(R_{n+2} + \gamma \mathrm{TF}_1(Z'_0, \theta_k) - \mathrm{TF}_1(Z_0, \theta_k)) \Pi \nabla_{Q_a} \mathrm{TF}_1(Z_0, \theta_k) \Pi^\top \mid \Pi]\right] && \text{(By (31), (32))} \\
&=\mathbb{E}_\Pi\left[\Pi \mathbb{E}[(R_{n+2} + \gamma \mathrm{TF}_1(Z'_0, \theta_k) - \mathrm{TF}_1(Z_0, \theta_k)) \nabla_{Q_a} \mathrm{TF}_1(Z_0, \theta_k) \mid \Pi] \Pi^\top\right] \\
&=\mathbb{E}_\Pi\left[\Pi \mathbb{E}[(R_{n+2} + \gamma \mathrm{TF}_1(Z'_0, \theta_k) - \mathrm{TF}_1(Z_0, \theta_k)) \nabla_{Q_a} \mathrm{TF}_1(Z_0, \theta_k)] \Pi^\top\right] \\
&=\mathbb{E}_\Pi\left[\Pi \mathrm{diag}(\Delta(Q_a)) \Pi^\top\right] \\
&=\frac{1}{d} \mathrm{tr}(\Delta(Q_a)) I_d && \text{(By Lemma A.3.2)} \\
&=\delta I_d.
\end{aligned}
$$

The proof is analogous for $\Delta(Q'_a) = \delta' I_d$ for some $\delta' \in \mathbb{R}$.

Suppose that $\Delta(p_{[2d+1]}) = \rho \in \mathbb{R}$, we now can conclude that

$$
\Delta(\theta_k) = \left\{ \Delta(P_0) = \begin{bmatrix} 0_{2d \times 2d} & 0_{2d \times 1} \\ 0_{1 \times 2d} & \rho \end{bmatrix}, \Delta(Q_0) = \begin{bmatrix} \delta I_d & 0_{d \times d} & 0_{d \times 1} \\ \delta' I_d & 0_{d \times d} & 0_{d \times 1} \\ 0_{1 \times d} & 0_{1 \times d} & 0 \end{bmatrix} \right\}.
$$

Therefore, according to (10), we get

$$
\begin{aligned}
&\theta_{k+1} \\
&=\theta_k + \alpha_k \Delta(\theta_k) \\
&=\left\{ \begin{bmatrix} 0_{2d \times 2d} & 0_{2d \times 1} \\ 0_{1 \times 2d} & \eta_k + \alpha_k \rho \end{bmatrix}, \begin{bmatrix} c_k + \alpha_k \delta I_d & 0_{d \times d} & 0_{d \times 1} \\ c'_k + \alpha_k \delta' I_d & 0_{d \times d} & 0_{d \times 1} \\ 0_{1 \times d} & 0_{1 \times d} & 0 \end{bmatrix} \right\} \in \Theta_*.
\end{aligned}
$$

$\square$

A.4 PROOF OF COROLLARY 2

*Proof.* We recall from (3) that the embedding evolves according to

$$Z_{l+1} = Z_l + \frac{1}{n} P_l Z_l M (Z_l^\top Q_l Z_l).$$

We again refer to the elements in $Z_l$ as $\left\{ (x_l^{(i)}, y_l^{(i)}) \right\}_{i=1,\ldots,n+1}$ in the following way

$$Z_l = \begin{bmatrix} x_l^{(1)} & \cdots & x_l^{(n)} & x_l^{(n+1)} \\ y_l^{(1)} & \cdots & y_l^{(n)} & y_l^{(n+1)} \end{bmatrix},$$

where we recall that $Z_l \in \mathbb{R}^{(2d+1)\times(n+1)}, x_l^{(i)} \in \mathbb{R}^{2d}, y_l^{(i)} \in \mathbb{R}$. Sometimes, it is more convenient to refer to the first half and second half of $x_l^{(i)}$ separately, by, e.g., $\nu_l^{(i)} \in \mathbb{R}^d, \xi_l^{(i)} \in \mathbb{R}^d$, i.e., $x_l^{(i)} = \begin{bmatrix} \nu_l^{(i)} \\ \xi_l^{(i)} \end{bmatrix}$. Then, we have

$$Z_l = \begin{bmatrix} \nu_l^{(1)} & \cdots & \nu_l^{(n)} & \nu_l^{(n+1)} \\ \xi_l^{(1)} & \cdots & \xi_l^{(n)} & \xi_l^{(n+1)} \\ y_l^{(1)} & \cdots & y_l^{(n)} & y_l^{(n+1)} \end{bmatrix}.$$

We utilize the shorthands

$$X_l = \begin{bmatrix} x_l^{(1)} & \cdots & x_l^{(n)} \end{bmatrix} \in \mathbb{R}^{2d \times n},$$
$$Y_l = \begin{bmatrix} y_l^{(1)} & \cdots & y_l^{(n)} \end{bmatrix} \in \mathbb{R}^{1 \times n}.$$

Then we have

$$Z_l = \begin{bmatrix} X_l & x_l^{(n+1)} \\ Y_l & y_l^{(n+1)} \end{bmatrix}.$$

For the input $Z_0$, we assume $\xi_0^{(n+1)} = 0, y_0^{(n+1)} = 0$ but all other entries of $Z_0$ are arbitrary. We recall our definition of $M$ in (2) and $\{P_l^{\mathrm{RG}}, Q_l^{\mathrm{RG}}\}$ in (12). In particular, we can express $Q_l^{\mathrm{RG}}$ in a more compact way as

$$M_1 \doteq \begin{bmatrix} -I_d & I_d \\ 0_{d\times d} & 0_{d\times d} \end{bmatrix} \in \mathbb{R}^{2d\times 2d},$$
$$M_2 \doteq -M_1$$
$$B_l \doteq \begin{bmatrix} C_l^\top & 0_{d\times d} \\ 0_{d\times d} & 0_{d\times d} \end{bmatrix} \in \mathbb{R}^{2d\times 2d},$$
$$A_l \doteq M_2^\top B_l M_1 = \begin{bmatrix} -C_l^\top & C_l^\top \\ C_l^\top & -C_l^\top \end{bmatrix} \in \mathbb{R}^{2d\times 2d},$$
$$Q_l^{\mathrm{RG}} \doteq \begin{bmatrix} A_l & 0_{2d\times 1} \\ 0_{1\times 2d} & 0 \end{bmatrix} \in \mathbb{R}^{(2d+1)\times(2d+1)}.$$

We then verify the following claims.

**Claim 1.** $X_l \equiv X_0, x_l^{(n+1)} \equiv x_0^{(n+1)}, \forall l$.

We note that $P_l^{\mathrm{RG}}$ is the key reason Claim 1 holds and is the same as the TD(0) case. Referring to A.1, we omit the proof of Claim 1 here.

**Claim 2.**

$$Y_{l+1} = Y_l + \frac{1}{n} Y_l X^\top A_l X$$
$$y_{l+1}^{(n+1)} = y_l^{(n+1)} + \frac{1}{n} Y_l X^\top A_l x^{(n+1)}.$$

Since the only difference between the true residual gradient and TD(0) configurations is the internal structure of $A_l$, we argue that it's irrelevant to Claim 2. We therefore again refer the readers to A.1 for a detailed proof.

**Claim 3.**

$$y_{l+1}^{(i)} = y_0^{(i)} + \left\langle M_1 x^{(i)}, \frac{1}{n} \sum_{j=0}^{l} B_j^\top M_2 X Y_j^\top \right\rangle,$$

for $i = 1, \ldots, n+1$.

By Claim 2, we can unroll $Y_{l+1}$ as

$$Y_{l+1} = Y_l + \frac{1}{n} Y_l X^\top A_l X$$

$$Y_l = Y_{l-1} + \frac{1}{n} Y_{l-1} X^\top A_{l-1} X$$

$$\vdots$$

$$Y_1 = Y_0 + \frac{1}{n} Y_0 X^\top A_0 X.$$

We can then compactly express $Y_{l+1}$ as

$$Y_{l+1} = Y_0 + \frac{1}{n} \sum_{j=0}^{l} Y_j X^\top A_j X.$$

Recall that we define $A_j = M_2^\top B_j M_1$. Then, we can rewrite $Y_{l+1}$ as

$$Y_{l+1} = Y_0 + \frac{1}{n} \sum_{j=0}^{l} Y_j X^\top M_2^\top B_j M_1 X.$$

With the identical procedure, we can easily rewrite $y_{l+1}^{(n+1)}$ as

$$y_{l+1}^{(n+1)} = y_0^{(n+1)} + \frac{1}{n} \sum_{j=0}^{l} Y_j X^\top M_2^\top B_j M_1 x^{(n+1)}.$$

In light of this, we define $\psi_0 \doteq 0$ and for $l = 0, \ldots$

$$\psi_{l+1} \doteq \frac{1}{n} \sum_{j=0}^{l} B_j^\top M_2 X Y_j^\top \in \mathbb{R}^{2d}$$

$$= \psi_l + \frac{1}{n} B_l^\top M_2 X Y_l^\top \tag{33}$$

Then we can write

$$y_{l+1}^{(i)} = y_0^{(i)} + \left\langle M_1 x^{(i)}, \psi_{l+1} \right\rangle, \tag{34}$$

for $i = 1, \ldots, n+1$, which is the claim we made. In particular, since we assume $y_0^{(n+1)} = 0$, we have

$$y_{l+1}^{(n+1)} = \left\langle M_1 x^{(n+1)}, \psi_{l+1} \right\rangle.$$

**Claim 4.** The bottom $d$ elements of $\psi_l$ are always 0, i.e., there exists a sequence $\{w_l \in \mathbb{R}^d\}$ such that we can express $\psi_l$ as

$$\psi_l = \begin{bmatrix} w_l \\ 0_{d \times 1} \end{bmatrix}.$$

for all $l = 0, 1, \ldots, L$.

Since $B_l$ is the key reason Claim 4 holds and is identical to the TD(0) case, we refer the reader to A.1 for detailed proof.

Given all the claims above, we can then compute that

$$
\begin{aligned}
&\left\langle \psi_{l+1}, M_1 x^{(n+1)} \right\rangle \\
&= \left\langle \psi_l, M_1 x^{(n+1)} \right\rangle + \frac{1}{n} \left\langle B_l^\top M_2 X Y_l^\top, M_1 x^{(n+1)} \right\rangle && \text{(By (33))} \\
&= \left\langle \psi_l, M_1 x^{(n+1)} \right\rangle + \frac{1}{n} \sum_{i=1}^n \left\langle B_l^\top M_2 x^{(i)} y_l^{(i)}, M_1 x^{(n+1)} \right\rangle \\
&= \left\langle \psi_l, M_1 x^{(n+1)} \right\rangle + \frac{1}{n} \sum_{i=1}^n \left\langle B_l^\top M_2 x^{(i)} \left( \left\langle \psi_l, M_1 x^{(i)} \right\rangle + y_0^{(i)} \right), M_1 x^{(n+1)} \right\rangle && \text{(By (34))} \\
&= \left\langle \psi_l, M_1 x^{(n+1)} \right\rangle + \frac{1}{n} \sum_{i=1}^n \left\langle B_l^\top \begin{bmatrix} \nu^{(i)} - \xi^{(i)} \\ 0_{d\times 1} \end{bmatrix} \left( \left\langle \psi_l, \begin{bmatrix} -\nu^{(i)} + \xi^{(i)} \\ 0_{d\times 1} \end{bmatrix} \right\rangle + y_0^{(i)} \right), M_1 x^{(n+1)} \right\rangle \\
&= \left\langle \psi_l, M_1 x^{(n+1)} \right\rangle + \frac{1}{n} \sum_{i=1}^n \left\langle \begin{bmatrix} C_l\left( \nu^{(i)} - \xi^{(i)} \right) \\ 0_{d\times 1} \end{bmatrix} \left( y_0^{(i)} + w_l^\top \xi^{(i)} - w_l^\top \nu^{(i)} \right), M_1 x^{(n+1)} \right\rangle \\
& && \text{(By Claim 4)} \\
&= \left\langle \psi_l, M_1 x^{(n+1)} \right\rangle + \frac{1}{n} \sum_{i=1}^n \left\langle \begin{bmatrix} C_l\left( \nu^{(i)} - \xi^{(i)} \right)\left( y_0^{(i)} + w_l^\top \xi^{(i)} - w_l^\top \nu^{(i)} \right) \\ 0_{d\times 1} \end{bmatrix}, M_1 x^{(n+1)} \right\rangle
\end{aligned}
$$

This means

$$
\left\langle w_{l+1}, \nu^{(n+1)} \right\rangle = \left\langle w_l, \nu^{(n+1)} \right\rangle + \frac{1}{n} \sum_{i=1}^n \left\langle C_l\left( \nu^{(i)} - \xi^{(i)} \right)\left( y_0^{(i)} + w_l^\top \xi^{(i)} - w_l^\top \nu^{(i)} \right), \nu^{(n+1)} \right\rangle.
$$

Since the choice of the query $\nu^{(n+1)}$ is arbitrary, we get

$$
w_{l+1} = w_l + \frac{1}{n} \sum_{i=1}^n C_l \left( y_0^{(i)} + w_l^\top \xi^{(i)} - w_l^\top \nu^{(i)} \right)\left( \nu^{(i)} - \xi^{(i)} \right).
$$

In particular, when we construct $Z_0$ such that $\nu^{(i)} = \phi_{i-1}$, $\xi^{(i)} = \gamma \phi_i$ and $y_0^{(i)} = R_i$, we get

$$
w_{l+1} = w_l + \frac{1}{n} \sum_{i=1}^n C_l \left( R_i + \gamma w_l^\top \phi_i - w_l^\top \phi_{i-1} \right)\left( \phi_{i-1} - \gamma \phi_i \right)
$$

which is the update rule for pre-conditioned residual gradient learning. We also have

$$
y_l^{(n+1)} = \left\langle \psi_l, M_1 x^{(n+1)} \right\rangle = -\left\langle w_l, \phi^{(n+1)} \right\rangle.
$$

This concludes our proof. $\qquad\square$

## A.5 PROOF OF COROLLARY 3

*Proof.* The proof presented here closely mirrors the methodology and notation established in the proof of Theorem 1 from Appendix A.1. We begin by recalling the embedding evolution from (3) as,

$$
Z_{l+1} = Z_l + \frac{1}{n} P_l Z_l M^{\text{TD}(\lambda)} (Z_l^\top Q_l Z_l).
$$

where we have substituted the original mask defined in (2) with the TD($\lambda$) mask in (14). We once again refer to the elements in $Z_l$ as $\left\{ (x_l^{(i)}, y_l^{(i)}) \right\}_{i=1,\ldots,n+1}$ in the following way

$$Z_l = \begin{bmatrix} x_l^{(1)} & \cdots & x_l^{(n)} & x_l^{(n+1)} \\ y_l^{(1)} & \cdots & y_l^{(n)} & y_l^{(n+1)} \end{bmatrix},$$

where we recall that $Z_l \in \mathbb{R}^{(2d+1) \times (n+1)}, x_l^{(i)} \in \mathbb{R}^{2d}, y_l^{(i)} \in \mathbb{R}$. We utilize, $\nu_l^{(i)} \in \mathbb{R}^d, \xi_l^{(i)} \in \mathbb{R}^d$, to refer to the first half and second half of $x_l^{(i)}$ i.e., $x_l^{(i)} = \begin{bmatrix} \nu_l^{(i)} \\ \xi_l^{(i)} \end{bmatrix}$.

Then we have

$$Z_l = \begin{bmatrix} \nu_l^{(1)} & \cdots & \nu_l^{(n)} & \nu_l^{(n+1)} \\ \xi_l^{(1)} & \cdots & \xi_l^{(n)} & \xi_l^{(n+1)} \\ y_l^{(1)} & \cdots & y_l^{(n)} & y_l^{(n+1)} \end{bmatrix}.$$

We further define as shorthands,

$$X_l = \begin{bmatrix} x_l^{(1)} & \cdots & x_l^{(n)} \end{bmatrix} \in \mathbb{R}^{2d \times n},$$
$$Y_l = \begin{bmatrix} y_l^{(1)} & \cdots & y_l^{(n)} \end{bmatrix} \in \mathbb{R}^{1 \times n}.$$

Then the blockwise structure of $Z_l$ can be succinctly expressed as:

$$Z_l = \begin{bmatrix} X_l & x_l^{(n+1)} \\ Y_l & y_l^{(n+1)} \end{bmatrix}.$$

We proceed to the formal arguments by paralleling those in Theorem 1. As in the theorem, we assume that certain initial conditions, such as $\xi_0^{(n+1)} = 0$ and $y_0^{(n+1)} = 0$, hold, but other entries of $Z_0$ are arbitrary. We recall our definition of $M^{\text{TD}(\lambda)}$ in (14) and $\left\{ P_l^{\text{TD}}, Q_l^{\text{TD}} \right\}_{l=0,\ldots,L-1}$ in (7). In particular, we can express $Q_l^{\text{TD}}$ in a more compact way as

$$M_1 \doteq \begin{bmatrix} -I_d & I_d \\ 0_{d \times d} & 0_{d \times d} \end{bmatrix} \in \mathbb{R}^{2d \times 2d},$$

$$B_l \doteq \begin{bmatrix} C_l^\top & 0_{d \times d} \\ 0_{d \times d} & 0_{d \times d} \end{bmatrix} \in \mathbb{R}^{2d \times 2d},$$

$$A_l \doteq B_l M_1 = \begin{bmatrix} -C_l^\top & C_l^\top \\ 0_{d \times d} & 0_{d \times d} \end{bmatrix} \in \mathbb{R}^{2d \times 2d},$$

$$Q_l^{\text{TD}} \doteq \begin{bmatrix} A_l & 0_{2d \times 1} \\ 0_{1 \times 2d} & 0 \end{bmatrix} \in \mathbb{R}^{(2d+1) \times (2d+1)},$$

We now proceed with the following claims.

In subsequent steps, it sometimes is useful to refer to the matrix $M^{\text{TD}(\lambda)} Z^\top$ in block form. Therefore, we will define $H^\top \in \mathbb{R}^{(n \times 2d)}$ as the first $n$ rows of $M_{\text{TD}(\lambda)} Z^\top$ except for the last column, which we define as $Y_l^{(\lambda)} \in \mathbb{R}^n$.

$$M^{\text{TD}(\lambda)} Z_l^\top = \begin{bmatrix} H^\top & Y_l^{(\lambda)} \\ 0_{1 \times 2d} & 0 \end{bmatrix} \in \mathbb{R}^{(n+1) \times (2d+1)}$$

Let $h^{(i)}$ denote $i$-th column of $H$.

We proceed with the following claims.

**Claim 1.** $X_l \equiv X_0, x_l^{(n+1)} \equiv x_0^{(n+1)}, \forall l$.

Because we utilize the same definition of $P_l^{\text{TD}}$ as in Theorem 1, the argument proving Claim 1 in Theorem 1 holds here as well. As a result, we drop all the subscripts of $X_l$, as well as subscripts of $x_l^{(i)}$ for $i = 1, \ldots, n+1$.

**Claim 2.** Let $H \in \mathbb{R}^{(2d \times n)}$, where the $i$-th column of $H$ is,

$$h^{(i)} = \sum_{k=1}^{i} \lambda^{i-k} x^{(i)} \in \mathbb{R}^{2d}.$$

Then we can write the updates for $Y_{l+1}$, and $y_{l+1}^{(n+1)}$ as,

$$Y_{l+1} = Y_l + \frac{1}{n} Y_l H^\top A_l X,$$

$$y_{l+1}^{(n+1)} = y_l^{(n+1)} + \frac{1}{n} Y_l H^\top A_l x^{(n+1)}.$$

We will show this by factoring the embedding evolution into the product of $P_l^{\text{TD}} Z_l$ and $M^{\text{TD}(\lambda)} Z_l^\top$, and $Q_l^{\text{TD}} Z_l$. Firstly, we have

$$P_l^{\text{TD}} Z_l = \begin{bmatrix} 0_{2d \times n} & 0_{2d \times 1} \\ Y_l & y_l^{(n+1)} \end{bmatrix}.$$

Next we analyze $M^{\text{TD}(\lambda)} Z_l^\top$. From basic matrix algebra we have,

$$M^{\text{TD}(\lambda)} Z^\top = \begin{bmatrix} 1 & 0 & 0 & 0 & \cdots & 0 & 0 \\ \lambda & 1 & 0 & 0 & \cdots & 0 & 0 \\ \lambda^2 & \lambda & 1 & 0 & \cdots & 0 & 0 \\ \lambda^3 & \lambda^2 & \lambda & 1 & \cdots & 0 & 0 \\ \vdots & \vdots & \vdots & \vdots & \ddots & \vdots & \vdots \\ \lambda^{n-1} & \lambda^{n-2} & \lambda^{n-3} & \lambda^{n-4} & \cdots & 1 & 0 \\ 0 & 0 & 0 & 0 & \cdots & 0 & 0 \end{bmatrix} \begin{bmatrix} x^{(1)\top} & y^{(1)} \\ x^{(2)\top} & y^{(2)} \\ x^{(3)\top} & y^{(3)} \\ \vdots & \vdots \\ x^{(n)\top} & y^{(n)} \\ x^{(n+1)\top} & 0 \end{bmatrix}$$

$$= \begin{bmatrix} x^{(1)\top} & y_l^{(1)} \\ x^{(2)\top} + \lambda x^{(1)\top} & y_l^{(2)} + \lambda y_l^{(2)} \\ \vdots & \vdots \\ \sum_{i=1}^{n} \lambda^{n-i} x_i^\top & \sum_{i=1}^{n} \lambda^{n-i} y_l^{(i)} \\ 0_{1 \times 2d} & 0 \end{bmatrix},$$

$$= \begin{bmatrix} h^{(1)\top} & y_l^{(1)} \\ h^{(2)\top} & y_l^{(2)} + \lambda y_l^{(1)} \\ \vdots & \vdots \\ h^{(n)\top} & \sum_{i=1}^{n} \lambda^{n-i} y_l^{(n)} \\ 0_{1 \times 2d} & 0 \end{bmatrix}$$

$$= \begin{bmatrix} H^\top & K_l^{(\lambda)} \\ 0_{1 \times 2d} & 0 \end{bmatrix},$$

where $K_l^{(\lambda)} \in \mathbb{R}^d$ is introduced for notation simplicity.

Then, we analyze $M^{\text{TD}(\lambda)} Z_l^\top Q_l^{\text{TD}} Z_l$. Applying the block matrix notations, we get

$$\left( M^{\text{TD}(\lambda)} Z_l^\top \right) Q_l^{\text{TD}} Z_l = \begin{bmatrix} H^\top & K_l^{(\lambda)} \\ 0_{1 \times 2d} & 0 \end{bmatrix} \begin{bmatrix} A_l & 0_{2d \times 1} \\ 0_{1 \times 2d} & 0 \end{bmatrix} \begin{bmatrix} X & x^{(n+1)} \\ Y_l & y_l^{(n+1)} \end{bmatrix}$$

$$= \begin{bmatrix} H^\top A_l & 0_{n \times 1} \\ 0_{1 \times 2d} & 0 \end{bmatrix} \begin{bmatrix} X & x^{(n+1)} \\ Y_l & y_l^{(n+1)} \end{bmatrix}$$

$$= \begin{bmatrix} H^\top A_l X & H^\top A_l x^{(n+1)} \\ 0_{1 \times 2d} & 0 \end{bmatrix}.$$

Combining the two, we get

$$P_l^{\text{TD}} Z_l \big(M^{\text{TD}(\lambda)} Z_l^\top Q_l^{\text{TD}} Z_l\big) = \begin{bmatrix} 0_{2d\times n} & 0_{2d\times 1} \\ Y_l & y_l^{(n+1)} \end{bmatrix} \begin{bmatrix} H^\top A_l X & H^\top A_l x^{(n+1)} \\ 0_{1\times 2d} & 0 \end{bmatrix}$$

$$= \begin{bmatrix} 0_{2d\times n} & 0_{2d\times 1} \\ Y_l H^\top A_l X & Y_l H^\top A_l x^{(n+1)} \end{bmatrix}.$$

Hence, according to our update rule in (3), we get

$$Y_{l+1} = Y_l + \frac{1}{n} Y_l H^\top A_l X$$

$$y_{l+1}^{(n+1)} = y_l^{(n+1)} + \frac{1}{n} Y_l H^\top A_l x^{(n+1)}.$$

**Claim 3.**

$$y_{l+1}^{(i)} = y_0^{(i)} + \left\langle M_1 x^{(i)}, \frac{1}{n} \sum_{i=0}^{l} B_i^\top M_2 X Y_i^\top \right\rangle,$$

for $i = 1, \dots, n+1$, where $M_2 = \begin{bmatrix} I_d & 0_{d\times d} \\ 0_{d\times d} & 0_{d\times d} \end{bmatrix}$.

Following Claim 2, we can unroll the recursive definition of $Y_{l+1}$ and express it compactly as,

$$Y_{l+1} = Y_0 + \frac{1}{n} \sum_{i=0}^{l} Y_i H^\top A_i X.$$

Recall that we define $A_i = B_i M_1$. Then, we can rewrite $Y_{l+1}$ as

$$Y_{l+1} = Y_0 + \frac{1}{n} \sum_{i=0}^{l} Y_i H^\top M_2 B_i M_1 X.$$

The introduction of $M_2$ here does not break the equivalence because $B_i = M_2 B_i$. However, it will help make our proof steps easier to comprehend later.

With the identical recursive unrolling procedure, we can rewrite $y_{l+1}^{(n+1)}$ as

$$y_{l+1}^{(n+1)} = y_0^{(n+1)} + \frac{1}{n} \sum_{i=0}^{l} Y_i H^\top M_2 B_i M_1 x^{(n+1)}.$$

In light of this, we define $\psi_0 \doteq 0$ and for $l = 0, \dots$

$$\psi_{l+1} \doteq \frac{1}{n} \sum_{i=0}^{l} B_i^\top M_2 H Y_i^\top \in \mathbb{R}^{2d}. \tag{35}$$

Then we can write

$$y_{l+1}^{(i)} = y_0^{(i)} + \left\langle M_1 x^{(i)}, \psi_{l+1} \right\rangle, \tag{36}$$

for $i = 1, \dots, n+1$, which is the claim we made. In particular, since we assume $y_0^{(n+1)} = 0$, we have

$$y_{l+1}^{(n+1)} = \left\langle M_1 x^{(n+1)}, \psi_{l+1} \right\rangle.$$

**Claim 4.** The bottom $d$ elements of $\psi_l$ are always 0, i.e., there exists a sequence $\{w_l \in \mathbb{R}^d\}$ such that we can express $\psi_l$ as

$$\psi_l = \begin{bmatrix} w_l \\ 0_{d\times 1} \end{bmatrix}.$$

for all $l = 0, 1, \ldots, L$.

Because we utilize the same definition of $B_l$ as in Theorem 1 when defining $\psi_{l+1}$, the argument proving Claim 4 in Theorem 1 holds here as well. We omit the steps to avoid redundancy.

Given all the claims above, we can then compute that

$$
\begin{aligned}
&\left\langle \psi_{l+1}, M_1 x^{(n+1)} \right\rangle \\
=&\left\langle \psi_l, M_1 x^{(n+1)} \right\rangle + \frac{1}{n} \left\langle B_l^\top M_2 H Y_l^\top, M_1 x^{(n+1)} \right\rangle && \text{(By (35))} \\
=&\left\langle \psi_l, M_1 x^{(n+1)} \right\rangle + \frac{1}{n} \sum_{i=1}^n \left\langle B_l^\top M_2 h^{(i)} y_l^{(i)}, M_1 x^{(n+1)} \right\rangle \\
=&\left\langle \psi_l, M_1 x^{(n+1)} \right\rangle + \frac{1}{n} \sum_{i=1}^n \left\langle B_l^\top M_2 h^{(i)} \left( \left\langle \psi_l, M_1 x^{(i)} \right\rangle + y_0^{(i)} \right), M_1 x^{(n+1)} \right\rangle && \text{(By (36))} \\
=&\left\langle \psi_l, M_1 x^{(n+1)} \right\rangle + \frac{1}{n} \sum_{i=1}^n \left\langle B_l^\top \left[ \begin{matrix} \left( \sum_{k=1}^i \lambda^{i-k} \nu^{(i)} \right) \\ 0_{d \times 1} \end{matrix} \right] \left( \left\langle \psi_l, \left[ \begin{matrix} -\nu^{(i)} + \xi^{(i)} \\ 0_{d \times 1} \end{matrix} \right] \right\rangle + y_0^{(i)} \right), M_1 x^{(n+1)} \right\rangle \\
=&\left\langle \psi_l, M_1 x^{(n+1)} \right\rangle + \frac{1}{n} \sum_{i=1}^n \left\langle \left[ \begin{matrix} C_l \left( \sum_{k=1}^i \lambda^{i-k} \nu^{(i)} \right) \\ 0_{d \times 1} \end{matrix} \right] \left( y_0^{(i)} + w_l^\top \xi^{(i)} - w_l^\top \nu^{(i)} \right), M_1 x^{(n+1)} \right\rangle \\
&&& \text{(By Claim 4)} \\
=&\left\langle \psi_l, M_1 x^{(n+1)} \right\rangle + \frac{1}{n} \sum_{i=1}^n \left\langle \left[ \begin{matrix} C_l \left( y_0^{(i)} + w_l^\top \xi^{(i)} - w_l^\top \nu^{(i)} \right) \left( \sum_{k=1}^i \lambda^{i-k} \nu^{(i)} \right) \\ 0_{d \times 1} \end{matrix} \right], M_1 x^{(n+1)} \right\rangle
\end{aligned}
$$

This means

$$
\left\langle w_{l+1}, \nu^{(n+1)} \right\rangle = \left\langle w_l, \nu^{(n+1)} \right\rangle + \frac{1}{n} \sum_{i=1}^n \left\langle C_l \left( y_0^{(i)} + w_l^\top \xi^{(i)} - w_l^\top \nu^{(i)} \right) \left( \sum_{k=1}^i \lambda^{i-k} \nu^{(i)} \right), \nu^{(n+1)} \right\rangle .
$$

Since the choice of the query $\nu^{(n+1)}$ is arbitrary, we get

$$
w_{l+1} = w_l + \frac{1}{n} \sum_{i=1}^n C_l \left( y_0^{(i)} + w_l^\top \xi^{(i)} - w_l^\top \nu^{(i)} \right) \left( \sum_{k=1}^i \lambda^{i-k} \nu^{(i)} \right).
$$

In particular, when we construct $Z_0$ such that $\nu^{(i)} = \phi_{i-1}$, $\xi^{(i)} = \gamma \phi_i$ and $y_0^{(i)} = R_i$, we get

$$
w_{l+1} = w_l + \frac{1}{n} \sum_{i=1}^n C_l \left( R_i + \gamma w_l^\top \phi_i - w_l^\top \phi_{i-1} \right) e_{i-1}
$$

where

$$
e_i = \sum_{k=1}^i \lambda^{i-k} \phi_k. \in \mathbb{R}^d
$$

which is the update rule for pre-conditioned TD($\lambda$). We also have

$$
y_l^{(n+1)} = \left\langle \psi_l, M_1 x^{(n+1)} \right\rangle = - \left\langle w_l, \phi^{(n+1)} \right\rangle.
$$

This concludes our proof. $\qquad \square$

### A.6 Proof of Theorem 3

*Proof.* We recall from (18) that the embedding evolves according to

$$Z_{l+1} = Z_l + \frac{1}{n}\text{TwoHead}(Z_l; P_l^{\overline{\text{TD}},(1)}, Q_l^{\overline{\text{TD}}}, M^{\overline{\text{TD}},(1)}, P_l^{\overline{\text{TD}},(2)}, Q_l^{\overline{\text{TD}}}, M^{\overline{\text{TD}},(2)}, W_l)$$

$$= Z_l + \frac{1}{n}W_l \begin{bmatrix} \text{LinAttn}(Z_l; P_l^{\overline{\text{TD}},(1)}, Q_l^{\overline{\text{TD}}}, M^{\overline{\text{TD}},(1)}) \\ \text{LinAttn}(Z_l; P_l^{\overline{\text{TD}},(2)}, Q_l^{\overline{\text{TD}}}, M^{\overline{\text{TD}},(2)}) \end{bmatrix}$$

In this configuration, we refer to the elements in $Z_l$ as $\left\{ (x_l^{(i)}, y_l^{(i)}, h_l^{(i)}) \right\}_{i=1,\ldots,n+1}$ in the following way,

$$Z_l = \begin{bmatrix} x_l^{(1)} & \ldots & x_l^{(n)} & x_l^{(n+1)} \\ y_l^{(1)} & \ldots & y_l^{(n)} & y_l^{(n+1)} \\ h_l^{(1)} & \ldots & h_l^{(n)} & h_l^{(n+1)} \end{bmatrix},$$

where we recall that $Z_l \in \mathbb{R}^{(2d+2)\times(n+1)}$, $x_l^{(i)} \in \mathbb{R}^{2d}$, $y_l^{(i)} \in \mathbb{R}$ and $h_l^{(i)} \in \mathbb{R}$.

Sometimes, it is more convenient to refer to the first half and second half of $x_l^{(i)}$ separately, by, e.g., $\nu_l^{(i)} \in \mathbb{R}^d, \xi_l^{(i)} \in \mathbb{R}^d$, i.e., $x_l^{(i)} = \begin{bmatrix} \nu_l^{(i)} \\ \xi_l^{(i)} \end{bmatrix}$. Then we have

$$Z_l = \begin{bmatrix} \nu_l^{(1)} & \ldots & \nu_l^{(n)} & \nu_l^{(n+1)} \\ \xi_l^{(1)} & \ldots & \xi_l^{(n)} & \xi_l^{(n+1)} \\ y_l^{(1)} & \ldots & y_l^{(n)} & y_l^{(n+1)} \\ h_l^{(1)} & \ldots & h_l^{(n)} & h_l^{(n+1)} \end{bmatrix}.$$

We further define as shorthands

$$X_l \doteq \begin{bmatrix} x_l^{(1)} & \ldots & x_l^{(n)} \end{bmatrix} \in \mathbb{R}^{2d\times n},$$

$$Y_l \doteq \begin{bmatrix} y_l^{(1)} & \ldots & y_l^{(n)} \end{bmatrix} \in \mathbb{R}^{1\times n},$$

$$H_l \doteq \begin{bmatrix} h_l^{(1)} & \ldots & h_l^{(n)} \end{bmatrix} \in \mathbb{R}^{1\times n}.$$

Then we can express $Z_l$ as

$$Z_l = \begin{bmatrix} X_l & x_l^{(n+1)} \\ Y_l & y_l^{(n+1)} \\ H_l & h_l^{(n+1)} \end{bmatrix}.$$

For the input $Z_0$, we assume $\xi_0^{(n+1)} = 0$ and $h_0^{(i)} = 0$ for $i = 1, \ldots, n+1$. All other entries of $Z_0$ are arbitrary. We recall our definition of $M^{\overline{\text{TD}},(1)}, M^{\overline{\text{TD}},(2)}$ in (17), $\left\{ P_l^{\overline{\text{TD}},(1)}, P_l^{\overline{\text{TD}},(2)}, Q_l^{\overline{\text{TD}}}, W_l \right\}$ in (15) and (16). We again express $Q_l^{\overline{\text{TD}}}$ as

$$M_1 \doteq \begin{bmatrix} -I_d & I_d \\ 0_{d\times d} & 0_{d\times d} \end{bmatrix} \in \mathbb{R}^{2d\times 2d},$$

$$B_l \doteq \begin{bmatrix} C_l^\top & 0_{d\times d} \\ 0_{d\times d} & 0_{d\times d} \end{bmatrix} \in \mathbb{R}^{2d\times 2d},$$

$$A_l \doteq B_l M_1 = \begin{bmatrix} -C_l^\top & C_l^\top \\ 0_{d\times d} & 0_{d\times d} \end{bmatrix} \in \mathbb{R}^{2d\times 2d},$$

$$Q_l^{\overline{\text{TD}}} \doteq \begin{bmatrix} A_l & 0_{2d\times 2} \\ 0_{2\times 2d} & 0_{2\times 2} \end{bmatrix} \in \mathbb{R}^{(2d+2)\times(2d+2)}.$$

We now proceed with the following claims that assist in proving our main theorem.

**Claim 1.** $X_l \equiv X_0, x_l^{(n+1)} \equiv x_0^{(n+1)}, Y_l \equiv Y_0, y_l^{(n+1)} = y_0^{(n+1)}, \forall l.$

We define

$$V_l^{(1)} \doteq P_l^{\overline{\text{TD}},(1)} Z_l M^{\overline{\text{TD}},(1)} \left( Z_l^\top Q_l^{\overline{\text{TD}}} Z_l \right) \in \mathbb{R}^{(2d+2) \times (n+1)}$$

$$V_l^{(2)} \doteq P_l^{\overline{\text{TD}},(2)} Z_l M^{\overline{\text{TD}},(2)} \left( Z_l^\top Q_l^{\overline{\text{TD}}} Z_l \right) \in \mathbb{R}^{(2d+2) \times (n+1)}.$$

Then the evolution of the embedding can be written as

$$Z_{l+1} = Z_l + \frac{1}{n} W_l \begin{bmatrix} V_l^{(1)} \\ V_l^{(2)} \end{bmatrix}.$$

By simple matrix arithmetic, we realize $W_l$ is merely summing up the $(2d+1)$-th row of $V_l^{(1)}$ and the $(2d+2)$-th row of $V_l^{(2)}$ and putting the result on its bottom row. Thus, we have

$$W_l \begin{bmatrix} V_l^{(1)} \\ V_l^{(2)} \end{bmatrix} = \begin{bmatrix} 0_{(2d+1) \times (n+1)} \\ V_l^{(1)}(2d+1) + V_l^{(2)}(2d+2) \end{bmatrix} \in \mathbb{R}^{(2d+2) \times (n+1)},$$

where $V_l^{(1)}(2d+1)$ and $V_l^{(2)}(2d+2)$ respectively indicate the $(2d+1)$-th row of $V_l^{(1)}$ and the $(2d+2)$-th row of $V_l^{(2)}$. It clearly holds according to the update rule that

$$Z_{l+1}(1:2d+1) = Z_l(1:2d+1)$$
$$\implies X_{l+1} = X_l;$$
$$x_{l+1}^{(n+1)} = x_l^{(n+1)};$$
$$Y_{l+1} = Y_l;$$
$$y_{l+1}^{(n+1)} = y_l^{(n+1)}.$$

Then, we can easily arrive at our claim by a simple induction. In light of this, we drop the subscripts of $X_l, x_l^{(i)}, Y_l$ and $y_l^{(i)}$ for all $i = 1, \ldots, n+1$ and write $Z_l$ as

$$Z_l = \begin{bmatrix} X & x^{(n+1)} \\ Y & y^{(n+1)} \\ H_l & h_l^{(n+1)} \end{bmatrix}.$$

**Claim 2.**

$$H_{l+1} = H_l + \frac{1}{n}(H_l + Y - \bar{Y}) X^\top A_l X$$

$$h_{l+1}^{(n+1)} = h_l^{(n+1)} + \frac{1}{n}(H_l + Y - \bar{Y}) X^\top A_l x^{(n+1)},$$

where $\bar{y}^{(i)} \doteq \sum_{k=1}^i \frac{y^{(k)}}{i}$ and $\bar{Y} \doteq [\bar{y}^{(1)}, \bar{y}^{(2)}, \ldots, \bar{y}^{(n)}] \in \mathbb{R}^{1 \times n}$.

We show how this claim holds by investigating the function of each attention head in our formulation. The first attention head, corresponding to $V_l^{(1)}$ in claim 1, has the form

$$P_l^{\overline{\text{TD}},(1)} Z_l M^{\overline{\text{TD}},(1)} \left( Z_l^\top Q_l^{\overline{\text{TD}}} Z_l \right).$$

We first analyze $P_l^{\overline{\text{TD}},(1)} Z_l M^{\overline{\text{TD}},(1)}$. It should be clear that $P_l^{\overline{\text{TD}},(1)} Z_l$ selects out the $(2d+1)$-th row of $Z_l$ and gives us

$$P_l^{\overline{\text{TD}},(1)} = \begin{bmatrix} 0_{2d \times n} & 0_{2d \times 1} \\ Y & y^{(n+1)} \\ 0_{1 \times n} & 0 \end{bmatrix}.$$

The matrix $M^{\overline{\text{TD}},(1)}$ is essentially computing $Y - \bar{Y}$ and filtering out the $(n+1)$-th entry when applied to $P_l^{\overline{\text{TD}},(1)} Z_l$. We break down the steps here:

$$P_l^{\overline{\text{TD}},(1)} Z_l M^{\overline{\text{TD}},(1)}$$

$$= P_l^{\overline{\text{TD}},(1)} Z_l \big(I_{n+1} - U_{n+1}\text{diag}\big(\begin{bmatrix} 1 & \frac{1}{2} & \cdots & \frac{1}{n} \end{bmatrix}\big)\big) M^{\overline{\text{TD}},(2)}$$

$$= P_l^{\overline{\text{TD}},(1)} Z_l M^{\overline{\text{TD}},(2)} - P_l^{\overline{\text{TD}},(1)} Z_l U_{n+1}\text{diag}\big(\begin{bmatrix} 1 & \frac{1}{2} & \cdots & \frac{1}{n} \end{bmatrix}\big) M^{\overline{\text{TD}},(2)}$$

$$= \begin{bmatrix} 0_{2d\times n} & 0_{2d\times 1} \\ Y & 0 \\ 0_{1\times n} & 0 \end{bmatrix} - \begin{bmatrix} 0_{2d\times 1} & 0_{2d\times 1} & \cdots & 0_{2d\times 1} & 0_{2d\times 1} \\ y^{(1)} & \frac{1}{2}\big(y^{(1)} + y^{(2)}\big) & \cdots & \frac{1}{n}\sum_{i=1}^{n} y^{(i)} & \frac{1}{n+1}\sum_{i=1}^{n+1} y^{(i)} \\ 0 & 0 & \cdots & 0 & 0 \end{bmatrix} M^{\overline{\text{TD}},(2)}$$

$$= \begin{bmatrix} 0_{2d\times n} & 0_{2d\times 1} \\ Y & 0 \\ 0_{1\times n} & 0 \end{bmatrix} - \begin{bmatrix} 0_{2d\times n} & 0_{2d\times 1} \\ \bar{Y} & 0 \\ 0_{1\times n} & 0 \end{bmatrix}$$

$$= \begin{bmatrix} 0_{2d\times n} & 0_{2d\times 1} \\ Y - \bar{Y} & 0 \\ 0_{1\times n} & 0 \end{bmatrix}.$$

We then analyze the remaining product $Z_l^\top Q_l^{\overline{\text{TD}}} Z_l$.

$$Z_l^\top Q_l^{\overline{\text{TD}}} Z_l$$

$$= \begin{bmatrix} X^\top & Y^\top & H_l^\top \\ x^{(n+1)\top} & y^{(n+1)\top} & h_l^{(n+1)\top} \end{bmatrix} \begin{bmatrix} A_l & 0_{2d\times 1} & 0_{2d\times 1} \\ 0_{1\times 2d} & 0 & 0 \\ 0_{1\times 2d} & 0 & 0 \end{bmatrix} \begin{bmatrix} X & x^{(n+1)} \\ Y & y^{(n+1)} \\ H_l & h_l^{(n+1)} \end{bmatrix}$$

$$= \begin{bmatrix} X^\top A_l & 0_{n\times 1} & 0_{n\times 1} \\ x^{(n+1)\top} A_l & 0 & 0 \end{bmatrix} \begin{bmatrix} X & x^{(n+1)} \\ Y & y^{(n+1)} \\ H_l & h_l^{(n+1)} \end{bmatrix}$$

$$= \begin{bmatrix} X^\top A_l X & X^\top A_l x^{(n+1)} \\ x^{(n+1)\top} A_l X & x^{(n+1)\top} A_l x^{(n+1)} \end{bmatrix}.$$

Putting them together, we get

$$P_l^{\overline{\text{TD}},(1)} Z_l M^{\overline{\text{TD}},(1)} \big(Z_l^\top Q_l^{\overline{\text{TD}}} Z_l\big) = \begin{bmatrix} 0_{2d\times n} & 0_{2d\times 1} \\ Y - \bar{Y} & 0 \\ 0_{1\times n} & 0 \end{bmatrix} \begin{bmatrix} X^\top A_l X & X^\top A_l x^{(n+1)} \\ x^{(n+1)\top} A_l X & x^{(n+1)\top} A_l x^{(n+1)} \end{bmatrix}$$

$$= \begin{bmatrix} 0_{2d\times n} & 0_{2d\times 1} \\ \big(Y - \bar{Y}\big) X^\top A_l X & \big(Y - \bar{Y}\big) X^\top A_l x^{(n+1)} \\ 0_{1\times n} & 0 \end{bmatrix}.$$

The second attention head, corresponding to $V_l^{(2)}$ in claim 1, has the form

$$P_l^{\overline{\text{TD}},(2)} Z_l M^{\overline{\text{TD}},(2)} \big(Z_l^\top Q_l^{\overline{\text{TD}}} Z_l\big).$$

It's obvious that $P_l^{\overline{\text{TD}},(2)}$ selects out the $(2d+2)$-th row of $Z_l$ as

$$P_l^{\overline{\text{TD}},(2)} Z_l = \begin{bmatrix} 0_{(2d+1)\times n} & 0_{(2d+1)\times 1} \\ H_l & h_l^{(n+1)} \end{bmatrix}.$$

Applying the mask $M^{\overline{\text{TD}},(2)}$, we get

$$P_l^{\overline{\text{TD}},(2)} Z_l M^{\overline{\text{TD}},(2)} = \begin{bmatrix} 0_{(2d+1)\times n} & 0_{(2d+1)\times 1} \\ H_l & 0 \end{bmatrix}.$$

The product $Z_l^\top Q_l^{\overline{\text{TD}}} Z_l$ is identical to the first attention head. Hence, we see the computation of the second attention head gives us

$$P_l^{\overline{\text{TD}},(2)} Z_l M^{\overline{\text{TD}},(2)} \big(Z_l^\top Q_l^{\overline{\text{TD}}} Z_l\big)$$

$$= \begin{bmatrix} 0_{(2d+1)\times n} & 0_{(2d+1)\times 1} \\ H_l & 0 \end{bmatrix} \begin{bmatrix} X^\top A_l X & X^\top A_l x^{(n+1)} \\ x^{(n+1)\top} A_l X & x^{(n+1)\top} A_l x^{(n+1)} \end{bmatrix}$$

$$= \begin{bmatrix} 0_{(2d+1)\times n} & 0_{(2d+1)\times 1} \\ H_l X^\top A_l X & H_l X^\top A_l x^{(n+1)} \end{bmatrix}.$$

Lastly, the matrix $W_l$ combines the output from the two heads and gives us

$$W_l \begin{bmatrix} P_l^{\overline{\text{TD}},(1)} Z_l M^{\overline{\text{TD}},(1)} \left( Z_l^\top Q_l^{\overline{\text{TD}}} Z_l \right) \\ P_l^{\overline{\text{TD}},(2)} Z_l M^{\overline{\text{TD}},(2)} \left( Z_l^\top Q_l^{\overline{\text{TD}}} Z_l \right) \end{bmatrix} = \begin{bmatrix} 0_{(2d+1)\times n} & 0_{(2d+1)\times 1} \\ (H_l + Y - \bar{Y}) X^\top A_l X & (H_l + Y - \bar{Y}) X^\top A_l x^{(n+1)} \end{bmatrix}.$$

Hence, we obtain the update rule for $H_l$ and $h_l^{(n+1)}$ as

$$H_{l+1} = H_l + \frac{1}{n}(H_l + Y - \bar{Y})X^\top A_l X$$

$$h_{l+1}^{(n+1)} = h_l^{(n+1)} + \frac{1}{n}(H_l + Y - \bar{Y})X^\top A_l x^{(n+1)}$$

and claim 2 has been verified.

**Claim 3.**

$$h_{l+1}^{(i)} = \left\langle M_1 x^{(i)}, \frac{1}{n}\sum_{j=0}^{l} B_i^\top M_2 X (H_j + Y - \bar{Y})^\top \right\rangle,$$

for $i = 1, \ldots, n+1$, where $M_2 = \begin{bmatrix} I_d & 0_{d\times d} \\ 0_{d\times d} & 0_{d\times d} \end{bmatrix}$.

Following claim 2, we unroll $H_{l+1}$ as

$$H_{l+1} = H_l + \frac{1}{n}(H_l + Y - \bar{Y})X^\top A_l X$$

$$H_l = H_{l-1} + \frac{1}{n}(H_{l-1} + Y - \bar{Y})X^\top A_{l-1}X$$

$$\vdots$$

$$H_1 = H_0 + \frac{1}{n}(H_0 + Y - \bar{Y})X^\top A_0 X.$$

We therefore can express $H_{l+1}$ as

$$H_{l+1} = H_0 + \frac{1}{n}\sum_{j=0}^{l}(H_j + Y - \bar{Y})X^\top A_j X.$$

Recall that we have defined $A_j \doteq B_j M_1$ and assumed $H_0 = 0$. Then, we have

$$H_{l+1} = \frac{1}{n}\sum_{j=0}^{l}(H_j + Y - \bar{Y})X^\top M_2 B_j M_1 X.$$

Note that the introduction of $M_2$ here does not break the equivalence because $B_j = M_2 B_j$. We include it in our expression for the convenience of the main proof later.

With the identical procedure, we can easily rewrite $h_{l+1}^{(n+1)}$ as

$$h_{l+1}^{(n+1)} = \frac{1}{n}\sum_{j=0}^{l}(H_j + Y - \bar{Y})X^\top M_2 B_j M_1 x^{(n+1)}.$$

In light of this, we define $\psi_0 \doteq 0$, and for $l = 0, \ldots$

$$\psi_{l+1} = \frac{1}{n}\sum_{j=0}^{l} B_j^\top M_2 X (H_j + Y - \bar{Y})^\top \in \mathbb{R}^{2d}.$$

We then can write

$$h_{l+1}^{(i)} = \left\langle M_1 x^{(i)}, \psi_{l+1} \right\rangle \tag{37}$$

for $i = 1, \ldots, n+1$, which is the claim we made.

**Claim 4.** The bottom $d$ elements of $\psi_l$ are always 0, i.e., there exists a sequence $\{w_l \in \mathbb{R}^d\}$ such that we can express $\psi_l$ as

$$\psi_l = \begin{bmatrix} w_l \\ 0_{d \times 1} \end{bmatrix}.$$

for all $l = 0, 1, \ldots, L$.

Since our $B_j$ here is identical to the proof of Theorem 1 in A.1 for $j = 0, 1, \ldots$, Claim 4 holds for the same reason. We therefore omit the proof details to avoid repetition.

Given all the claims above, we proceed to prove our main theorem.

$$
\begin{aligned}
& \left\langle \psi_{l+1}, M_1 x^{(n+1)} \right\rangle \\
=& \left\langle \psi_l, M_1 x^{(n+1)} \right\rangle + \frac{1}{n} \left\langle B_l^\top M_2 X (H_l + Y - \bar{Y})^\top, M_1 x^{(n+1)} \right\rangle \\
=& \left\langle \psi_l, M_1 x^{(n+1)} \right\rangle + \frac{1}{n} \sum_{i=1}^n \left\langle B_l^\top M_2 x^{(i)} (h_l^{(i)} + y^{(i)} - \bar{y}^{(i)}), M_1 x^{(n+1)} \right\rangle \\
=& \left\langle \psi_l, M_1 x^{(n+1)} \right\rangle + \frac{1}{n} \sum_{i=1}^n \left\langle B_l^\top M_2 x^{(i)} \left( \left\langle \psi_l, M_1 x^{(i)} \right\rangle + y^{(i)} - \bar{y}^{(i)} \right), M_1 x^{(n+1)} \right\rangle \quad \text{(By (37))} \\
=& \left\langle \psi_l, M_1 x^{(n+1)} \right\rangle + \frac{1}{n} \sum_{i=1}^n \left\langle B_l^\top \begin{bmatrix} \nu^{(i)} \\ 0_{d \times 1} \end{bmatrix} \left( \left\langle \psi_l, \begin{bmatrix} -\nu^{(i)} + \xi^{(i)} \\ 0_{d \times 1} \end{bmatrix} \right\rangle + y^{(i)} - \bar{y}^{(i)} \right), M_1 x^{(n+1)} \right\rangle \\
=& \left\langle \psi_l, M_1 x^{(n+1)} \right\rangle + \frac{1}{n} \sum_{i=1}^n \left\langle \begin{bmatrix} C_l \nu^{(i)} \\ 0_{d \times 1} \end{bmatrix} \left( y^{(i)} - \bar{y}^{(i)} + w_l^\top \xi^{(i)} - w_l^\top \nu^{(i)} \right), M_1 x^{(n+1)} \right\rangle \\
& \hspace{10cm} \text{(By Claim 4)} \\
=& \left\langle \psi_l, M_1 x^{(n+1)} \right\rangle + \frac{1}{n} \sum_{i=1}^n \left\langle \begin{bmatrix} C_l \nu^{(i)} \left( y^{(i)} - \bar{y}^{(i)} + w_l^\top \xi^{(i)} - w_l^\top \nu^{(i)} \right) \\ 0_{d \times 1} \end{bmatrix}, M_1 x^{(n+1)} \right\rangle
\end{aligned}
$$

This means

$$\left\langle w_{l+1}, \nu^{(n+1)} \right\rangle = \left\langle w_l, \nu^{(n+1)} \right\rangle + \frac{1}{n} \sum_{i=1}^n \left\langle C_l \nu^{(i)} \left( y^{(i)} - \bar{y}^{(i)} + w_l^\top \xi^{(i)} - w_l^\top \nu^{(i)} \right), \nu^{(n+1)} \right\rangle.$$

Since the choice of the query $\nu^{(n+1)}$ is arbitrary, we get

$$w_{l+1} = w_l + \frac{1}{n} \sum_{i=1}^n C_l \left( y^{(i)} - \bar{y}^{(i)} + w_l^\top \xi^{(i)} - w_l^\top \nu^{(i)} \right) \nu^{(i)}.$$

In particular, when we construct $Z_0$ such that $\nu^{(i)} = \phi_{i-1}, \xi^{(i)} = \phi_i$ and $y^{(i)} = R_i$, we get

$$w_{l+1} = w_l + \frac{1}{n} \sum_{i=1}^n C_l \left( R_i - \bar{r}_i + w_l^\top \phi_i - w_l^\top \phi_{i-1} \right) \phi_{i-1}$$

which is the update rule for pre-conditioned average reward TD learning. We also have

$$h_l^{(n+1)} = \left\langle \psi_l, M_1 x^{(n+1)} \right\rangle = -\left\langle w_l, \phi^{(n+1)} \right\rangle.$$

This concludes our proof. $\qquad \square$

## B    EXPERIMENTAL DETAILS OF FIGURE 1

We generate Figure 1 with 300 randomly generated policy evaluation tasks. Each task consists of a randomly generated Markov Decision Process (MDP), a randomly generated policy, and a randomly generated feature function (See Section 3 for detailed definition). The number of states of the MDP ranges from 5 to 10, while the features are always in $\mathbb{R}^5$. The reward is also randomly generated, but we make sure the true value function is representable (cf. Algorithm 3). This treatment ensures that the minimal possible MSVE for each task is always 0. The discount factor is always $\gamma = 0.9$.

## C    BOYAN'S CHAIN EVALUATION TASK GENERATION

To generate the evaluation tasks used to meta-train our transformer in Algorithm 1, we utilize Boyan's chain, detailed in Figure 3. Notably, we make some minor adjustments to the original Boyan's chain in Boyan (1999) to make it an infinite horizon chain.

Recall that an evaluation task is defined by the tuple $(p_0, p, r, \phi)$. We consider Boyan's chain MRPs with $m$ states. To construct $p_0$, we first sample a $m$-dimensional random vector uniformly in $[0, 1]^m$ and then normalize it to a probability distribution. To construct $p$, we keep the structure of Boyan's chain but randomize the transition probabilities. In particular, the transition function $p$ can be regarded as a random matrix taking value in $\mathbb{R}^{m \times m}$. To simplify the presentation, we use both $p(s, s')$ and $p(s'|s)$ to denote the probability of transitioning to $s'$ from $s$. In particular, for $i = 1, \ldots, m - 2$, we set $p(i, i + 1) = \epsilon$ and $p(i, i + 2) = 1 - \epsilon$, with $\epsilon$ sampled uniformly from $(0, 1)$. For the last two states, we have $p(m|m - 1) = 1$ and $p(\cdot|m)$ is a random distribution over all states. Each element of the vector $r \in \mathbb{R}^m$ and the matrix $\phi \in \mathbb{R}^{d \times m}$ are sampled i.i.d. from a uniform distribution over $[-1, 1]$. The overall task generation process is summarized in Algorithm 2. Almost surely, no task will be generated twice. In our experiments in the main text, we use Boyan Chain MRPs, which consist of $m = 10$ states, each with feature dimension $d = 4$.

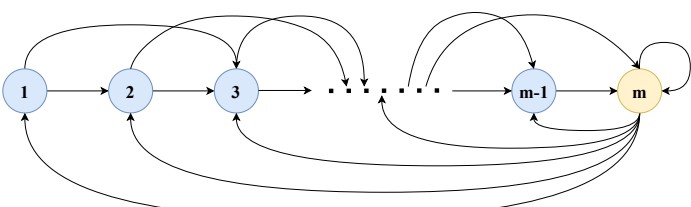

Figure 3: Boyan's Chain of $m$ States

**Representable Value Function.** With the above sampling procedure, there is no guarantee that the true value function $v$ is always representable by the features. In other words, there is no guarantee that there exists a $w \in \mathbb{R}^d$ satisfying $v(s) = \langle w, \phi(s) \rangle$ for all $s \in \mathcal{S}$. Most of our experiments use this setup. It is, however, also beneficial sometimes to work with evaluation tasks where the true value function is guaranteed to be representable. Algorithm 3 achieves this by randomly generating a $w_*$ first and compute $v(s) \doteq \langle w_*, \phi(s) \rangle$. The reward is then analytically computed as $r \doteq (I_m - \gamma p)v$. We recall that in the above, we regard $p$ as a matrix in $\mathbb{R}^{m \times m}$.

---

**Algorithm 2:** Boyan Chain MRP and Feature Generation (Non-Representable)

---

1: **Input:** state space size $m = |\mathcal{S}|$, feature dimension $d$
2: **for** $s \in \mathcal{S}$ **do**
3:      $\phi(s) \sim \text{Uniform}\left[(-1, 1)^d\right]$   `// feature`
4: **end for**
5: $p_0 \sim \text{Uniform}\left[(0, 1)^m\right]$   `// initial distribution`
6: $p_0 \leftarrow p_0 / \sum_s p_0(s)$
7: $r \sim \text{Uniform}\left[(-1, 1)^m\right]$   `// reward function`
8: $p \leftarrow 0_{m \times m}$   `// transition function`
9: **for** $i = 1, \ldots, m-2$ **do**
10:      $\epsilon \sim \text{Uniform}\left[(0, 1)\right]$
11:      $p(i, i+1) \leftarrow \epsilon$
12:      $p(i, i+2) \leftarrow 1 - \epsilon$
13: **end for**
14: $p(m-1, m) \leftarrow 1$
15: $z \leftarrow \text{Uniform}\left[(0, 1)^m\right]$
16: $z \leftarrow z / \sum_s z(s)$
17: $p(m, 1:m) \leftarrow z$
18: **Output:** MRP $(p_0, p, r)$ and feature map $\phi$

---

---

**Algorithm 3:** Boyan Chain MRP and Feature Generation (Representable)

---

1: **Input:** state space size $m = |\mathcal{S}|$, feature dimension $d$, discount factor $\gamma$
2: $w^* \sim \text{Uniform}\left[(-1, 1)^d\right]$   `// ground-truth weight`
3: **for** $s \in \mathcal{S}$ **do**
4:      $\phi(s) \sim \text{Uniform}\left[(-1, 1)^d\right]$   `// feature`
5:      $v(s) \leftarrow \langle w^*, \phi(s) \rangle$   `// ground-truth value function`
6: **end for**
7: $p_0 \sim \text{Uniform}\left[(0, 1)^m\right]$   `// initial distribution`
8: $p_0 \leftarrow p_0 / \sum_s p_0(s)$
9: $p \leftarrow 0_{m \times m}$   `// transition function`
10: **for** $i = 1, \ldots, m-2$ **do**
11:      $\epsilon \sim \text{Uniform}\left[(0, 1)\right]$
12:      $p(i, i+1) \leftarrow \epsilon$
13:      $p(i, i+2) \leftarrow 1 - \epsilon$
14: **end for**
15: $p(m-1, m) \leftarrow 1$
16: $z \leftarrow \text{Uniform}\left[(0, 1)^m\right]$
17: $z \leftarrow z / \sum_s z(s)$
18: $p(m, 1:m) \leftarrow z$
19: $r \leftarrow (I_m - \gamma p)v$   `// reward function`
20: **Output:** MRP $(p_0, p, r)$ and feature map $\phi$

---

# D   ADDITIONAL EXPERIMENTS WITH LINEAR TRANSFORMERS

## D.1   EXPERIMENT SETUP

We use Algorithm 2 as $d_{\text{task}}$ for the experiments in the main text with Boyan's chain of 10 states. In particular, we consider a context of length $n = 30$, feature dimension $d = 4$, and utilize a discount factor $\gamma = 0.9$. In Section 5, we consider a 3-layer transformer ($L = 3$), but additional analyses on the sensitivity to the number of transformer layers ($L$) and results from a larger scale experiment with $d = 8, n = 60$, and $|\mathcal{S}| = 20$ are presented in D.2. We also explore non-autoregressive (i.e., "sequential") layer configurations in D.3.

When training our transformer, we utilize an Adam optimizer (Kingma and Ba, 2015) with an initial learning rate of $\alpha = 0.001$ and weight decay rate of $1 \times 10^{-6}$. $P_0$ and $Q_0$ are randomly

initialized using Xavier initialization with a gain of $0.1$. We trained our transformer on $k = 4000$ different evaluation tasks. For each task, we generated a trajectory of length $\tau = 347$, resulting in $\tau - n - 2 = 320$ transformer parameter updates.

Since the models in these experiments are small ($\sim 10$ KB), we did not use any GPU during our experiments. We trained our transformers on a standard Intel i9-12900-HK CPU, and training each transformer took $\sim 20$ minutes.

For implementation[6], we used NumPy (Harris et al., 2020) to process the data and construct Boyan's chain, PyTorch (Ansel et al., 2024) to define and train our models, and Matplotlib (Hunter, 2007) plus SciencePlots (Garrett, 2021) to generate our figures.

### D.1.1   TRAINED TRANSFORMER ELEMENT-WISE CONVERGENCE METRICS

To visualize the parameters of the linear transformer trained by Algorithm 1, we report element-wise metrics. For $P_0$, we report the value of its bottom-right entry, which, as noted in (7), should approach one if the transformer is learning to implement TD. The other entries of $P_0$ should remain close to zero. Additionally, we report the average absolute value of the elements of $P_0$, excluding the bottom-right entry, to check if these elements stay near zero during training.

For $Q_0$, we recall from (7) that if the transformer learned to implement normal batch TD, the upper-left $d \times d$ block of the matrix should converge to some $-I_d$, while the upper-right $d \times d$ block (excluding the last column) should converge to $I_d$. To visualize this, we report the trace of the upper-left $d \times d$ block and the trace of the upper-right $d \times d$ block (excluding the last column). The rest of the elements of $Q_0$ should remain close to 0, and to verify this, we report the average absolute value of the entries of $Q_0$, excluding the entries that were utilized in computing the traces.

Since, $P_0$ and $Q_0$ are in the same product in (1) we sometimes observe during training that $P_0$ converges to $-P_0^{\mathrm{TD}}$ and $Q_0$ converges to $-Q_0^{\mathrm{TD}}$ simultaneously. When visualizing the matrices, we negate both $P_0$ and $Q_0$ when this occurs.

It's also worth noting that in Theorem 1 we prove a $L$-layer transformer parameterized as in (7) with $C_0 = I_d$ implements $L$ steps of batch TD exactly with a fixed update rate of one. However, the transformer trained using Algorithm 1 could learn to perform TD with an arbitrary learning rate ($\alpha$ in (5)). Therefore, even if the final trained $P_0$ and $Q_0$ differ from their constructions in (7) by some scaling factor, the resulting algorithm implemented by the trained transformer will still be implementing TD. In light of this, we rescale $P_0$ and $Q_0$ before visualization. In particular, we divide $P_0$ and $Q_0$ by the maximum of the absolute values of their entries, respectively, such that they both stay in the range $[-1, 1]$ after rescaling.

### D.1.2   TRAINED TRANSFORMER AND BATCH TD COMPARISON METRICS

To compare the transformers with batch TD we report several metrics following Von Oswald et al. (2023a); Akyürek et al. (2023). Given a context $C \in \mathbb{R}^{(2d+1) \times n}$ and a query $\phi \in \mathbb{R}^d$, we construct the prompt as

$$Z^{(\phi, C)} \doteq \begin{bmatrix} C & \begin{bmatrix} \phi \\ 0_{d \times 1} \\ 0 \end{bmatrix} \end{bmatrix}.$$

We will suppress the context $C$ in subscript when it does not confuse. We use $Z^{(s)} \doteq Z^{(\phi(s))}$ as shorthand. We use $d_p$ to denote the stationary distribution of the MRP with transition function $p$ and assume the context $C$ is constructed based on trajectories sampled from this MRP. Then, we can define $v_\theta \in \mathbb{R}^{|\mathcal{S}|}$, where $v_\theta(s) \doteq \mathrm{TF}_L(Z_0^{(s)}; \theta)$ for each $s \in \mathcal{S}$. Notably, $v_\theta$ is then the value function estimation induced by the transformer parameterized by $\theta \doteq \{(P_l, Q_l)\}$ given the context $C$. In the rest of the appendix, we will use $\theta_{\mathrm{TF}}$ as the learned parameter from Algorithm 1. As a result, $v_{\mathrm{TF}} \doteq v_{\theta_{\mathrm{TF}}}$ denotes the learned value function.

We define $\theta_{\mathrm{TD}} \doteq \left\{ (P_l^{\mathrm{TD}}, Q_l^{\mathrm{TD}}) \right\}_{l=0,\dots,L-1}$ with $C_l = \alpha I$ (see (7)) and

$$v_{\mathrm{TD}}(s) \doteq \mathrm{TF}_L(Z_0^{(s)}; \theta_{\mathrm{TD}}).$$

---
[6]The code will be made publicly available upon publication.

In light of Theorem 1, $v_{\text{TD}}$ is then the value function estimation obtained by running the batch TD algorithm (8) on the context $C$ for $L$ iterations, using a constant learning rate $\alpha$.

We would like to compare the two functions $v_{\text{TF}}$ and $v_{\text{TD}}$ to future examine the behavior of the learned transformers. However, $v_{\text{TD}}$ is not well-defined yet because it still has a free parameter $\alpha$, the learning rate. Von Oswald et al. (2023a) resolve a similar issue in the in-context regression setting via using a line search to find the (empirically) optimal $\alpha$. Inspired by Von Oswald et al. (2023a), we also aim to find the empirically optimal $\alpha$ for $v_{\text{TD}}$. We recall that $v_{\text{TD}}$ is essentially the transformer $\text{TF}_L(Z_0^{(s)}; \theta_{\text{TD}})$ with only 1 single free parameter $\alpha$. We then train this transformer with Algorithm 1. We observe that $\alpha$ quickly converges and use the converged $\alpha$ to complete the definition of $v_{\text{TD}}$. We are now ready to present different metrics to compare $v_{\text{TF}}$ and $v_{\text{TD}}$. We recall that both are dependent on the context $C$.

**Value Difference (VD).** First, for a given context $C$, we compute the Value Difference (VD) to measure the difference between the value function approximated by the trained transformer and the value function learned by batch TD, weighted by the stationary distribution. To this end, we define,

$$\text{VD}(v_{\text{TF}}, v_{\text{TD}}) \doteq \|v_{\text{TF}} - v_{\text{TD}}\|_{d_p}^2,$$

We recall that $d_p \in \mathbb{R}^{|\mathcal{S}|}$ is the stationary distribution of the MRP, and the weighted $\ell_2$ norm is defined as $\|v\|_d \doteq \sqrt{\sum_s v(s)^2 d(s)}$.

**Implicit Weight Similarity (IWS).** We recall that $v_{\text{TD}}$ is a linear function, i.e., $v_{\text{TD}}(s) = \langle w_L, \phi(s) \rangle$ with $w_L$ defined in Theorem 1. We refer to this $w_L$ as $w_{\text{TD}}$ for clarity. The learned value function $v_{\text{TF}}$ is, however, not linear even with a linear transformer. Following Akyürek et al. (2023), we compute the best linear approximation of $v_{\text{TF}}$. In particular, given a context $C$, we define

$$w_{\text{TF}} \doteq \arg\min_w \|\Phi w - v_{\text{TF}}\|_{d_p}.$$

Here $\Phi \in \mathbb{R}^{|\mathcal{S}| \times d}$ is the feature matrix, each of which is $\phi(s)^\top$. Such a $w_{\text{TF}}$ is referred to as implicit weight in Akyürek et al. (2023). Following Akyürek et al. (2023), we define

$$\text{IWS}(v_{\text{TF}}, v_{\text{TD}}) \doteq d_{\cos}(w_{\text{TF}}, w_{\text{TD}})$$

to measure the similarity between $w_{\text{TF}}$ and $w_{\text{TD}}$. Here $d_{\cos}(\cdot, \cdot)$ computes the cos similarity between two vectors.

**Sensitivity Similarity (SS).** Recall that $v_{\text{TF}}(s) = \text{TF}_L(Z_0^{(s)}; \theta_{\text{TF}})$ and $v_{\text{TD}}(s) = \text{TF}_L(Z_0^{(s)}; \theta_{\text{TD}})$. In other words, given a context $C$, both $v_{\text{TF}}(s)$ and $v_{\text{TD}}(s)$ are functions of $\phi(s)$. Following Von Oswald et al. (2023a), we then measure the sensitivity of $v_{\text{TF}}(s)$ and $v_{\text{TD}}(s)$ w.r.t. $\phi(s)$. This similarity is easily captured by gradients. In particular, we define

$$\text{SS}(v_{\text{TF}}, v_{\text{TD}}) \doteq \sum_s d_p(s) d_{\cos} \left( \nabla_\phi \text{TF}_L(Z_0^{(\phi)}; \theta_{\text{TF}}) \Big|_{\phi=\phi(s)}, \nabla_\phi \text{TF}_L(Z_0^{(\phi)}; \theta_{\text{TD}}) \Big|_{\phi=\phi(s)} \right).$$

Notably, it trivially holds that

$$w_{\text{TD}} = \nabla_\phi \text{TF}_L(Z_0^{(\phi)}; \theta_{\text{TD}}) \Big|_{\phi=\phi(s)}.$$

We note that the element-wise convergence of learned transformer parameters (e.g., Figure 2a) is the most definite evidence for the emergence of in-context TD. The three metrics defined in this section are only auxiliary when linear attention is concerned. That being said, **the three metrics are important when nonlinear attention is concerned**.

## D.2    Autoregressive Linear Transformers with $L = 1, 2, 3, 4$ Layers

In this section, we present the experimental results for autoregressive linear transformers with different numbers of layers. In Figure 4, we present the element-wise convergence metrics for autoregressive transformers with $L = 1, 2, 4$ layers. The plot with $L = 3$ is in Figure 2 in the main text. We can see that for the $L = 1$ case, $P_0$ and $Q_0$ converge to the construction in Corollary 1, which, as proved, implements TD(0) in the single layer case. For the $L = 2, 4$ cases, we see that $P_0$ and $Q_0$ converge to

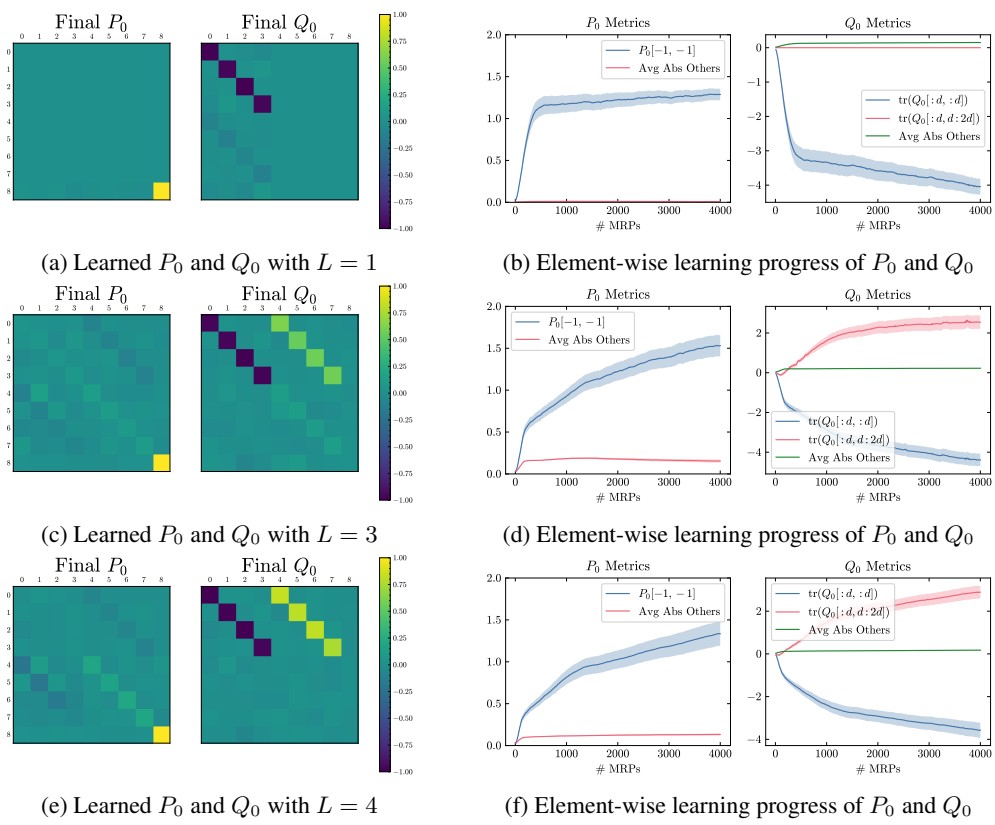

Figure 4: Visualization of the learned **autoregressive** transformers and the learning progress. Averaged across 30 seeds and the shaded region denotes the standard errors. See Appendix D.1.1 for details about normalization of $P_0$ and $Q_0$ before visualization.

the construction in Theorem 1. We also observe that as the number of transformer layers $L$ increases, the learned parameters are more aligned with the construction of $P_0^{\text{TD}}$ and $Q_0^{\text{TD}}$ with $C_0 = I$.

We also present the comparison of the learned transformer with batch TD according to the metrics described in Appendix D.1.2. In Figure 5, we present the value difference, implicit weight similarity, and sensitivity similarity. In Figures 5a − 5d, we present the results for different transformer layer numbers $L = 1, 2, 3, 4$. In Figure 5e, we present the metrics for a 3-layer transformer, but we increase the feature dimension to $d = 8$ and also the context length to $n = 60$.

In all instances, we see a strong similarity between the trained linear transformers and batch TD. We see that the cosine similarities of the sensitivities are near one, as are the implicit weight similarities. Additionally, the value difference approaches zero during training. This further demonstrates that the autoregressive linear transformers trained according to Algorithm 1 learn to implement TD(0).

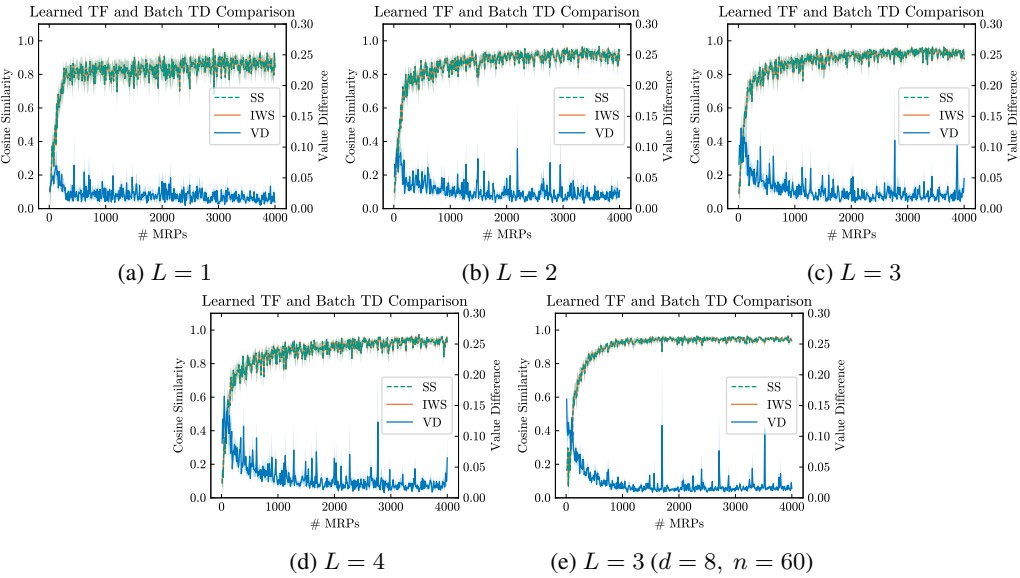

Figure 5: Value difference (VD), implicit weight similarity (IWS), and sensitivity similarity (SS) between the learned **autoregressive** transformers and batch TD with different layers. All curves are averaged over 30 seeds and the shaded regions are the standard errors.

## D.3 SEQUENTIAL TRANSFORMERS WITH $L = 2, 3, 4$ LAYERS

So far, we have been using linear transformers with one parametric attention layer applied repeatedly for $L$ steps to implement an $L$-layer transformer. Another natural architecture in contrast with the autoregressive transformer is a sequential transformer with $L$ distinct attention layers, where the embedding passes over each layer exactly once during one pass of forward propagation.

In this section, we repeat the same experiments we conduct on the autoregressive transformer with sequential transformers with $L = 2, 3, 4$ as their architectures coincide when $L = 1$. We compare the sequential transformers with batch TD(0) and report the three metrics in Figure 6. We observe that the implicit weight similarity and the sensitivity similarity grow drastically to near 1, and the value difference drops considerably after a few hundred MRPs for all three layer numbers. It suggests that sequential transformers trained via Algorithm 1 are functionally close to batch TD.

Figure 7 shows the visualization of the converged $\{P_l, Q_l\}_{l=0,1,2}$ of a 3-layer sequential linear transformer and their element-wise convergence. Sequential transformers exhibit very special patterns in their learned weights. We see that the input layer converges to a pattern very close to our configuration in Theorem (1). However, the deeper the layer, we observe the more the diagonal of $Q_l[1:d, d+1:2d]$ fades. The $P$ matrices, on the other hand, follow our configuration closely, especially for the final layer. We speculate this pattern emerges because sequential transformers have

more parametric attention layers and thus can assign a slightly different role to each layer but together implement batch TD(0) as suggested by the black-box functional comparison in Figure 6.

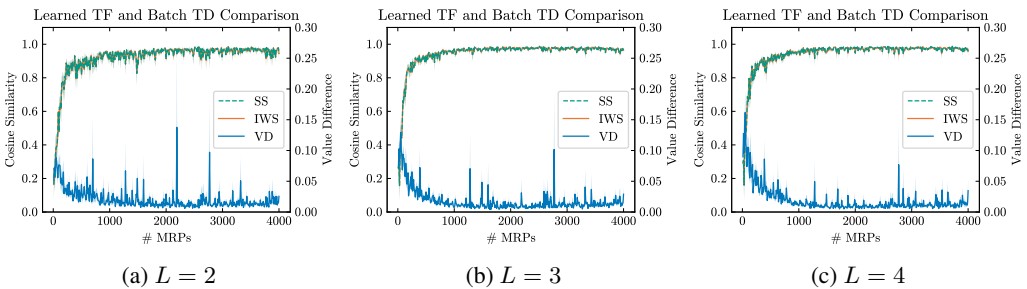

(a) $L = 2$        (b) $L = 3$        (c) $L = 4$

Figure 6: Value difference (VD), implicit weight similarity (IWS), and sensitivity similarity (SS) between the learned **sequential** transformers and batch TD with different layers. All curves are averaged over 30 seeds, and the shaded regions are the standard errors.

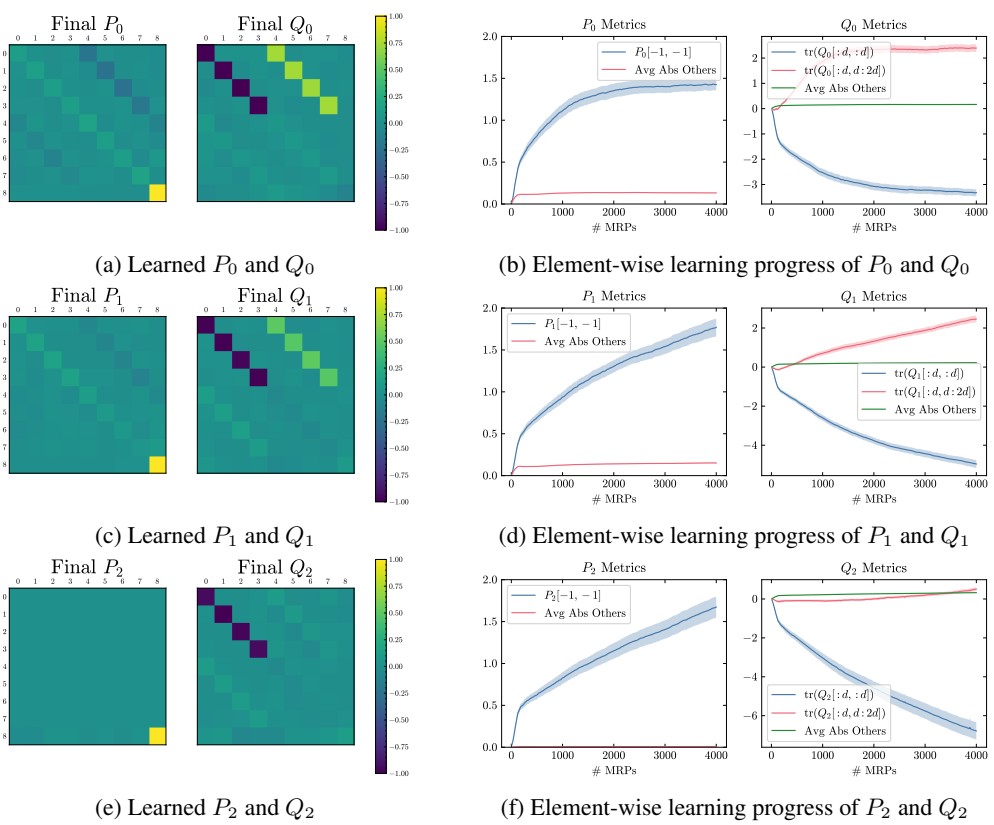

(a) Learned $P_0$ and $Q_0$      (b) Element-wise learning progress of $P_0$ and $Q_0$

(c) Learned $P_1$ and $Q_1$      (d) Element-wise learning progress of $P_1$ and $Q_1$

(e) Learned $P_2$ and $Q_2$      (f) Element-wise learning progress of $P_2$ and $Q_2$

Figure 7: Visualization of the learned $L = 3$ **sequential** transformers and the learning progress. Averaged across 30 seeds and the shaded region denotes the standard errors. See Appendix D.1.1 for details about normalization of $P_0$ and $Q_0$ before visualization.

# E  NONLINEAR ATTENTION

Until now, we have focused on only linear attention. In this section, we empirically investigate original transformers with the softmax function. Given a matrix $Z$, we recall that self-attention computes its embedding as

$$\text{Attn}(Z; P, Q) = PZM\text{softmax}\big(Z^\top QZ\big).$$

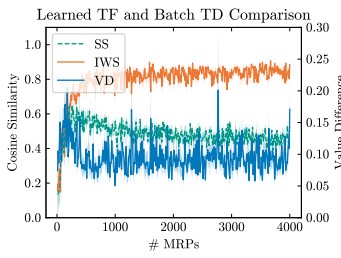
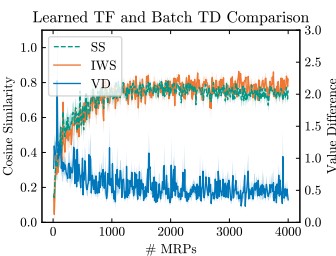

(a) General Value Function       (b) Representable Value Function

Figure 8: Value difference (VD), implicit weight similarity (IWS), and sensitivity similarity (SS) between the learned softmax transformers and linear batch TD. All curves are averaged over 30 seeds, and the shaded regions are the standard errors.

Let $Z_l \in \mathbb{R}^{(2d+1)\times(n+1)}$ denote the input to the $l$-th layer, the output of an $L$-layer transformer with parameters $\{(P_l, Q_l)\}_{l=0,\dots,L-1}$ is then computed as

$$Z_{l+1} = Z_l + \frac{1}{n}\text{Attn}(Z_l; P_l, Q_l) = Z_l + \frac{1}{n}PZM\text{softmax}\big(Z^\top QZ\big).$$

Analogous to the linear transformer, we define

$$\widetilde{\text{TF}}_L\Big(Z_0; \{P_l, Q_l\}_{l=0,1\dots,L-1}\Big) \doteq -Z_L[2d+1, n+1].$$

As a shorthand, we use $\widetilde{\text{TF}}_L(Z_0)$ to denote the output of the softmax transformers given prompt $Z_0$. We use the same training procedure (Algorithm 1) to train the softmax transformers. In particular, we consider a 3-layer autoregressive softmax transformer.

Notably, the three metrics in Appendix D.1.2 apply to softmax transformers as well. We still compare the learned softmax transformer with the linear batch TD in (8). In other words, the $v_{\text{TD}}$ related quantities are the same, and we only recompute $v_{\text{TF}}$ related quantities in Appendix D.1.2. As shown in Figure 8a, the value difference remains small, and the implicit weight similarity increases. This suggests that the learned softmax transformer behaves similarly to linear batch TD. The sensitivity similarity, however, drops. This is expected. The learned softmax transformer $\widetilde{\text{TF}}_L$ is unlikely to be a linear function w.r.t. to the query while $v_{\text{TD}}$ is linear w.r.t. the query. So their gradients w.r.t. the query are unlikely to match. To further investigate this hypothesis, we additionally consider evaluation tasks where the true value function is guaranteed to be representable (Algorithm 3) and is thus a linear function w.r.t. the state feature. This provides more incentives for the learned softmax transformer to behave like a linear function. As shown in Figure 8b, the sensitivity similarity now increases.

# F   EXPERIMENTS WITH CARTPOLE ENVIRONMENT

In this section, we present additional experimental results demonstrating that in-context TD emerges after large-scale pretraining using Algorithm 1 where $d_{\text{task}}$ is derived from the CartPole environment (Brockman et al., 2016).

## F.1   CARTPOLE EVALUATION TASK GENERATION

Recall that in the main text, as well as Appendix D and E, the transformers are pre-trained with tasks drawn from $d_{\text{task}}$ based on Boyan's Chain (See Appendix C). Here, we extend the analysis by introducing $d_{\text{task}}$ based on the CartPole environment. Figure 9 provides an introduction to the CartPole environment.

Recall that an evaluation task is defined by the tuple $(p_0, p, r, \phi)$. In the canonical CartPole environment, the states are a vector $s \in \mathbb{R}^4$ where the entries are the current position of the cart, the velocity of the cart, the angle of the pole, and the angular velocity of the pole. In our experiments, the initial state distribution $p_0$ and environment transition dynamics $p(s'|s, a)$ are given by the standard CartPole equations (e.g. see OpenAI CartPole Github). These transition dynamics, which we denote

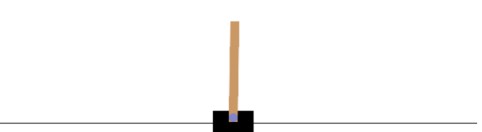

Figure 9: The OpenAI Gym CartPole environment (Brockman et al., 2016) is a classic RL control task where the goal is to balance a pole on a cart by applying forces to move the cart left or right. The state consists of the cart's position and velocity and the pole's angle and angular velocity. The episode ends if the cart moves out of bounds or the pole falls beyond a threshold angle.

as $p_{\text{CartPole}}(s'|s, a)$, implicitly depend on the physical parameters $\Psi \doteq (m_{\text{cart}}, m_{\text{pole}}, g, l_{\text{pole}}, \tau, f)$ representing the mass of the cart and pole, gravitational constant, length of the pole, frame rate, and the force magnitude. We abuse the notation of $p_{\text{CartPole}}(s'|s, a; \Psi)$ to highlight the transition dependency on $\Psi$. The joint distribution over these parameters, denoted by $\Delta_\Psi$, defines the the possible CartPole environments. In our experiments, we sampled $m_{\text{cart}}, m_{\text{pole}}, l_{\text{pole}} \sim \text{Uniform}[0.5, 1.5]$, $g \sim \text{Uniform}[7, 12], \tau \sim \text{Uniform}[0.01, 0.05], f \sim \text{Uniform}[5, 15]$.

Then, the state transition function $p(s'|s)$ which characterizes an MRP is defined using $p_{\text{CartPole}}(s'|s, a)$, and a fixed random policy $\pi_\epsilon(a|s)$ parameterized by $\epsilon \sim \text{Uniform}[(0, 1)]$. Under $\pi_\epsilon(a|s)$, the probability of moving the cart to the right is $\epsilon$ and the probability of moving the cart to the left is $1 - \epsilon$. This means that $p(s'|s) = \sum_{a \in \{0,1\}} p(s'|s, a)\pi_\epsilon(a|s)$ where 0 means going left and 1 means going right. The environment is extended to an infinite horizon. When the pole falls, or the cart moves out of bounds, the state is reset by sampling a new initial state from $p_0$.

Rather than using the standard CartPole observations and reward structure of +1 per time step until failure, we provide a diverse set of reward functions and features by sampling $r$ and $\phi$ randomly. In CartPole, the state $s$ is continuous, resulting in an infinite state space $\mathcal{S}$. To address this, we use tile coding (Sutton and Barto, 2018) with a random projection to generate a feature function $\phi : \mathcal{S} \to \mathbb{R}^d$ for $s \in \mathcal{S}$. Tile coding with random projection maps $s$ to a feature vector sampled from $\text{Uniform}[(-1, 1)^d]$. Similarly, for the reward function $r : \mathcal{S} \to \mathbb{R}$, $s$ is mapped to a reward value, also sampled from $\text{Uniform}[-1, 1]$. The joint distribution over random features and reward functions is denoted $\Delta_{\phi, r}(d)$. For each CartPole MRP, we sample from $\Delta_{\phi, r}$ to obtain the feature and reward functions $\phi$ and $r$. This approach, detailed in Algorithm 4, enables the transformer to encounter a variety of tasks during pre-training.

---

**Algorithm 4:** CartPole MRP and Feature Generation

---

1: **Input:** feature dimension $d$, action space $\mathcal{A} = \{0, 1\}$, joint distribution over CartPole parameters $\Delta_\Psi$, joint distribution over features and rewards $\Delta_{\phi, r}$
2: $\Psi \sim \Delta_\Psi$  // sample CartPole parameter
3: $p_0 \leftarrow \text{Uniform}[(-0.05, 0.05)^4]$  // CartPole initial distribution
4: $\phi, r \leftarrow \Delta_{\phi, r}(d)$  // sample features and rewards
5: $\epsilon \sim \text{Uniform}[(0, 1)]$  // sample random policy parameter
6: $p(s'|s) \leftarrow \sum_{a \in \mathcal{A}} \pi_\epsilon(a|s)p_{\text{CartPole}}(s'|s, a; \Psi)$  // CartPole state transition
7: **Output:** MRP $(p_0, p, r)$ and feature map $\phi$

---

### F.2 EXPERIMENTAL RESULTS OF PRE-TRAINING WITH CARTPOLE

In our experiments in Figure 10, we pre-train a 3-layer autoregressive transformer using Algorithm 1, where the task distribution $d_{\text{task}}$ is generated using CartPole MRPs (see Algorithm 4) with a feature vector of dimension $d = 4$. We used a significantly larger context window length $n = 250$. Despite the increased complexity of the transition dynamics in the CartPole environment compared to Boyan's

chain environment used in Figure 2, our results demonstrate that $P_0$ and $Q_0$ still converge to the construction in Theorem 1 (up to some noise), which we proved exactly implements TD(0).

It is worth noting that our theoretical results (Theorem 2), which prove that the weights implementing TD are in the invariant set of the updates in Algorithm 1, do not depend on any specific properties of the environment $p$. Thus, it is unsurprising that TD(0) emerges naturally even after pre-training on environments with complicated dynamics.

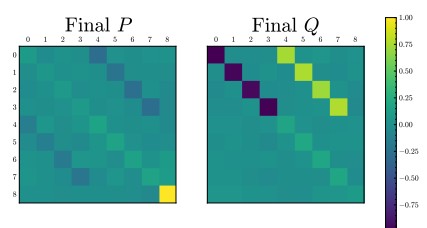
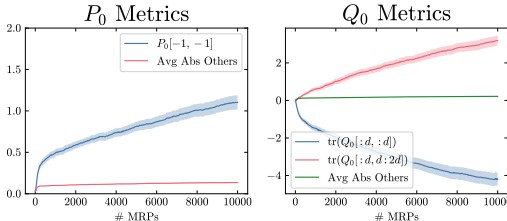

(a) Learned $P_0$ and $Q_0$ after 10000 MRPs

(b) Element-wise learning progress of $P_0$ and $Q_0$

Figure 10: Visualization of the learned transformers and the learning progress after pretraining with the CartPole environment for 10,000 MRPs. Both (a) and (b) are averaged across 30 seeds and the shaded regions in (b) denote the standard errors.

## G    INVESTIGATION OF IN-CONTEXT TD WITH RNN

We have focused primarily on the transformer's capability to implement TD in context. Before transformers, the canonical architecture to tackle sequence modelling problems is the recurrent neural network (RNN) (Elman, 1990; Bengio et al., 2017). Thus, it's worth investigating the algorithmic capacity of RNN in implementing TD in its forward pass. In particular, we try to answer the following two questions in this section:

1. Can RNN implement TD in context?
2. Does in-context TD emerge in RNN via multi-task pre-training?

A canonical deep RNN with $L$ layers is parameterized by $\left\{ W_{ax}^{(l)}, W_{aa}^{(l)}, b_a^{(l)} \right\}_{l=0,\dots,L-1}$. Let $m$ denote the dimension of the raw input tokens and $h$ denote that of the hidden states, respectively. Then, we have $W_{ax}^{(0)} \in \mathbb{R}^{h \times m}$, $W_{ax}^{(l)} \in \mathbb{R}^{h \times h}$ for $l = 1, \dots, L-1$, and $W_{aa}^{(l)} \in \mathbb{R}^{h \times h}$, $b_a^{(l)} \in \mathbb{R}^h$ for $l = 0, \dots, L-1$. Let $x_t^{(l)}$ denote the input token and $a_t^{(l)}$ denote the hidden state for layer $l$ at time step $t$. Unlike transformers that process the whole sequence at once, an RNN processes one token after another sequentially by updating the hidden states. The hidden state evolves according to

$$a_{t+1}^{(l)} = f\left( W_{ax}^{(l)} x_t^{(l)} + W_{aa}^{(l)} a_t^{(l)} + b_a^{(l)} \right)$$

where $f$ is an activation function. In addition, we have $x_t^{(l)} = a_t^{(l-1)}$ for all $t$ and $l = 1, \dots, L-1$. In other words, the input to the next depth is the hidden state from the previous depth except for the first layer. The initial hidden states $a_0^{(l)}, l = 0, \dots, L-1$ are selected arbitrarily. Popular options include zero initialization and random normal initialization.

When we apply RNN to policy evaluation, we are interested in predicting a scalar value at the end, also known as many-to-one prediction. Suppose the input sequence has $n$ tokens one typically passes $a_n^{L-1}$, the final hidden state at the last recurrent layer, through a fully connected output layer $W_o \in \mathbb{R}^{1 \times h}$, such that

$$\hat{v} = W_o a_n^{L-1}.$$

### G.1    THEORETICAL ANALYSIS OF LINEAR RNN

We first investigate Question 1 via a theoretical analysis of RNN in the context of TD. Due to the intractable difficulty of nonlinear activations present in deep neural network analysis, we resort to

analyzing a single-layer linear RNN, i.e., $L = 1$ and $f$ is the identity mapping. Hence, we will drop the superscript indicating the layer index and $f$ in our notation to simplify the presentation. We shall also remove the bias term $b_a$ because it is a constant independent of the context. Under this formulation, the hidden state evolves according to

$$a_{t+1} = W_{ax}x_t + W_{aa}a_t.$$

If we initialize $a_0 = 0$, we then have

$$a_0 = 0$$
$$a_1 = W_{aa}a_0 + W_{ax}x_0 = W_{ax}x_0$$
$$a_2 = W_{aa}a_1 + W_{ax}x_1 = W_{ax}x_1 + W_{aa}W_{ax}x_0$$
$$a_3 = W_{aa}a_2 + W_{ax}x_2 = W_{ax}x_2 + W_{aa}W_{ax}x_1 + W_{aa}^2 W_{ax}x_0$$
$$\vdots$$

Assuming a sequence of $n$ tokens, the final hidden state $a_n$ is

$$a_n = \sum_{t=0}^{n-1} W_{aa}^{n-t-1} W_{ax}x_t.$$

Applying a linear output layer $W_o \in \mathbb{R}^{1 \times h}$ to the hidden state for value prediction, we then get

$$\hat{v} = W_o a_n = \sum_{t=0}^{n-1} W_o W_{aa}^{n-t-1} W_{ax}x_t = \sum_{t=0}^{n-1} w_t^\top x_t, \tag{38}$$

where $w_t \doteq \left(W_o W_{aa}^{n-t-1} W_{ax}\right)^\top \in \mathbb{R}^h$ is a vector. (38) demonstrates that the predicted value is the sum of the inner product between each token and some vector for linear RNN. Recall that each context token $x_t$ for in-context TD is defined as

$$x_t \doteq \begin{bmatrix} \phi_t \\ \gamma\phi_t' \\ R_t \end{bmatrix},$$

corresponding to column $t$ of the prompt $Z$. Hence, we can write

$$\hat{v} = \sum_{t=0}^{n-1} w_t^\top \begin{bmatrix} \phi_t \\ \gamma\phi_t' \\ R_t \end{bmatrix}.$$

Under this representation, it is impossible to construct the TD error, not to mention applying the semi-gradient term. Therefore, it is safe for us to claim that linear RNN is incapable of implementing TD in its forward pass. This result is easily extendable to the multi-layer case since it is only performing linear combinations of the tokens, thus reducible to the format of (38). One important insight gained by comparing the forward pass of an RNN and a transformer under linear activation is that one at least needs $x_t^\top Q x_t$ where $Q$ is a square matrix to have any hope to compute the TD error, which is necessary for TD. Therefore, we speculate that a deep RNN equipped with a common nonlinear activation such as $\tanh$ and ReLU is also unable to implement TD in context. We will leave the investigation to Question 2. For now, we can confidently give a negative answer to Question 1 concerning linear RNNs.

## G.2 MULTI-TASK TD WITH DEEP RNN

We answer Question 2 via an empirical study with a deep RNN. We employ a 3-layer RNN with a hidden state dimension of $h = 4$ and $\tanh$ as the activation function and train it via multi-task TD (Algorithm 1) on 4,000 randomly generated Boyan's chain MRPs with a feature dimension of $d = 4$. Since we cannot apply a mask $M$ like in the transformer to distinguish the query from the context, we instead append a binary flag to each token for the same purpose. Suppose there are $n$ context columns, the prompt $Z$ has the form

$$Z = \begin{bmatrix} \phi_1 & \phi_2 & \cdots & \phi_n & \phi_{n+1} \\ \gamma\phi_1' & \gamma\phi_2' & \cdots & \gamma\phi_n' & 0 \\ R_1 & R_2 & \cdots & R_n & 0 \\ 0 & 0 & \cdots & 0 & 1 \end{bmatrix} \in \mathbb{R}^{(2d+2)\times(n+1)}.$$

The forward pass of the deep RNN processes the tokens sequentially in the prompt to update the hidden states. The final hidden state of the last layer of the RNN is fed into a fully connected layer to output a scalar value prediction. Figure 11 shows the learning curve of the RNN throughout the multi-task TD training. The MSVE decreases for the first 1,000 MRPs and stays low for the remainder of the training. Thus, some learning occurs during the training of RNN. However, it is unclear whether it is implementing in-context TD. To clarify, we use the last checkpoint of the model and repeat the same experiment used to generate Figure 1. We gradually increase the context length and verify if the MSVE drops as observed in the transformers. We run the experiment on the Loop environment used to generate Figure 1 and the Boyan's chain environment used for training for 500 instances each to produce Figure 12. The MSVE increases with context length in both environments for the trained RNN, exhibiting a trend opposite to the transformer. Furthermore, the standard errors are much higher than in Figure 1 despite having more runs. Therefore, the prediction does not improve with more context data for the RNN, indicating the absence of any in-context policy evaluation algorithms. Consequently, the answer to Question 2 is again negative.

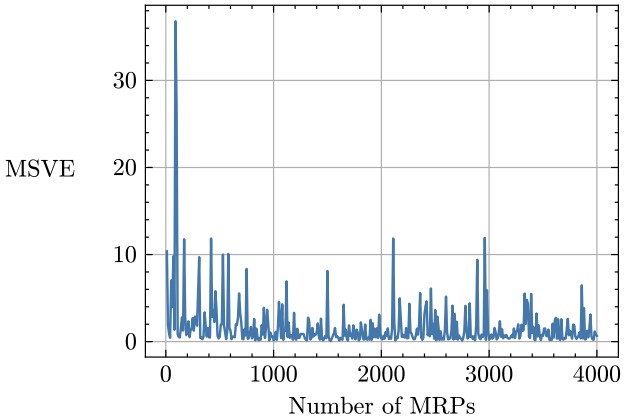

Figure 11: RNN MSVE against the number of MRPs in multi-task TD training.

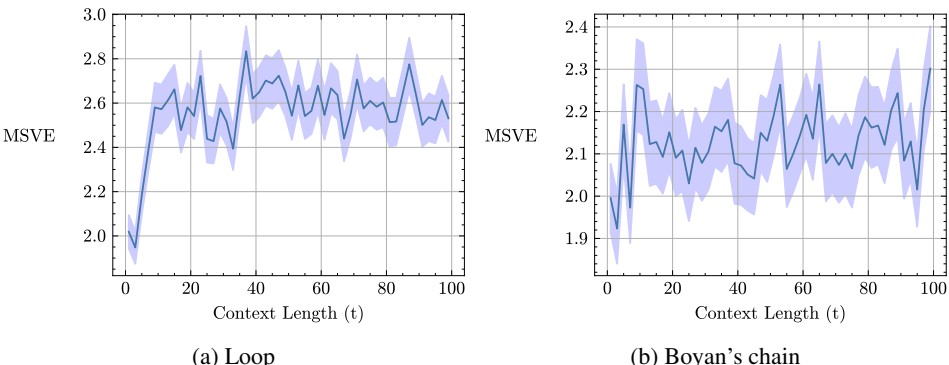

| (a) Loop | (b) Boyan's chain |
|---|---|

Figure 12: MSVE vs. context length with the trained RNN. The shaded regions are the standard errors.

## H    NUMERICAL VERIFICATION OF PROOFS

We provide numerical verification for our proofs by construction (Theorem 1, Corollary 2, Corollary 3, and Theorem 3) as a sanity check. In particular, we plot $\log \left| -\langle \phi_n, w_l \rangle - y_l^{n+1} \right|$ against the number of layers $l$. For example, for Theorem 1, we first randomly generate $Z_0$ and $\{C_l\}$. Then $y_l^{(n+1)}$ is computed by unrolling the transformer layer by layer following (3) while $w_l$ is computed iteration by iteration following (8). We use double-precision floats and run for 30 seeds, each with a new prompt.

As shown in Figure 13, even after 40 layers/iterations, the difference is still in the order of $10^{-10}$. It is not strictly 0 because of numerical errors. It sometimes increases because of the accumulation of numerical errors.

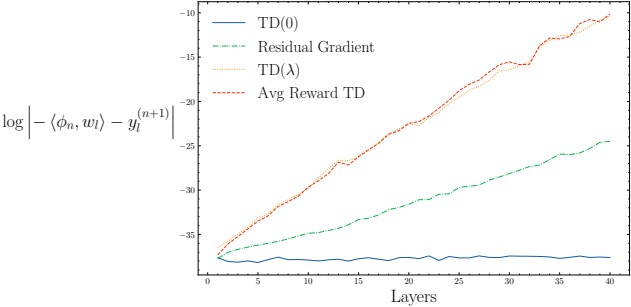

Figure 13: Differences between transformer output and batch TD output. Curves are averaged over 30 random seeds with the (invisible) shaded region showing the standard errors.

