# OpenReview forum: "Transformers Can Learn Temporal Difference Methods for In-Context Reinforcement Learning"
_ICLR.cc/2025/Conference — ICLR 2025 Poster_

### Official Review · Reviewer_zyPc · 2024-10-31

**Soundness:** 3
**Presentation:** 2
**Contribution:** 3
**Rating:** 6
**Confidence:** 3

**Summary:**

This paper provides theoretical analysis on the equivalence between a forward pass and in-context TD in a simple setting - linear single-layer transformers.

**Strengths:**

It is a new try and an interesting direction to investigate the equivalence between transformers and in-context TD.

**Weaknesses:**

1. Many of the proofs rely on strict initial conditions, like specific embeddings and matrix structures. The validity of these proofs depends heavily on these controlled setups, which may not be feasible or sustainable in more complex, real-world settings. As a result, some claims in the paper appear intuitive and unsubstantiated.
2. While the paper claims the emergence of in-context TD learning through multi-task TD pretraining, the empirical analysis is constrained to quite simple tasks.

**Questions:**

1. In Corollary 3, the authors show that the transformer can implement TD(λ) updates with a specific mask matrix, $M_{TD(\lambda)}$. The matrix structure requires strict alignment of eligibility traces, which might be difficult to achieve in real tasks where temporal dependencies and eligibility traces shift dynamically.

2. Lines 523-524 in the Conclusion are unclear. Are you trying to emphasize that policy iteration or optimization forms the key foundation?

3. In Lines 531-532, the claim seems unsupported; working well in a small-scale setting doesn’t necessarily imply the same for larger, more complex tasks.

---

> ### Author Response · Authors · 2024-11-21
> **Response to reviewer zyPc**
>
> > Weaknesses:
> > 1. Many of the proofs rely on strict initial conditions, like specific embeddings and matrix structures. The validity of these proofs depends heavily on these controlled setups, which may not be feasible or sustainable in more complex, real-world settings. As a result, some claims in the paper appear intuitive and unsubstantiated.
>
> We would like to clarify that we did not use any "special initial conditions" in our work. The matrices and embeddings we provide in Theorems do not represent initial conditions. Instead, we are saying that that if the network parameters converge to the specified values, the transformer will exhibit the desired property, namely exactly implementing some known RL algorithm (TD(0) in Theorem 1, TD($\lambda$) in Theorem 3, and average reward TD in Theorem 4) in the forward pass. This is the essence of a proof by construction: rather than merely asserting the existence of parameters with certain properties, we provide a concrete construction of those parameters. In our experiments, the transformers are initialized randomly using standard initialization methods, and the specific parameters naturally emerge during training.
>
> Regarding the claim that "some claims in the paper appear intuitive and unsubstantiated", Reviewer 77PR confirms that "the paper argues with proofs and empirical evidences for its results and is able to remove any possible doubt of their veracity." Reviewer 5HXw also appreciates our rigorous mathematical proofs: "The proposed theorems and corollaries in the paper are meticulously detailed in the appendix, where each proof is presented with clarity and rigor." Yet, we will gladly elaborate on any **specific results** that reviewer found unsubstantiated.
>
>
> > 2. While the paper claims the emergence of in-context TD learning through multi-task TD pretraining, the empirical analysis is constrained to quite simple tasks.
>
> First, we want to emphasize that the simplicity of the original pretraining environment strengthens our findings. Isn’t it interesting that a randomly initialized transformer, trained on simple tasks, can learn to implement TD(0)—a provably capable RL algorithm for policy evaluation?
>
> Nevertheless, in Appendix F, we also added new empirical analysis using pretraining tasks from CartPole—a more complex and widely-used benchmark in RL. In these large-scale experiments, we pretrain transformers on policy evaluation problems with a context length of n=250. Consistent with our findings on simpler tasks, the transformers successfully learn to implement in-context TD learning.
>
> We also want to highlight that our theoretical results (Theorem 2) do not rely on specific properties of the environment $P$. Theorem 2 proves that the weights implementing TD remain invariant under the updates in Algorithm 1, which explains why TD(0) emerges naturally, even in environments with more complex dynamics.

---

> ### Author Response · Authors · 2024-11-21
> **Response to reviewer zyPc (part 2)**
>
> > Questions:
> > 1. In Corollary 3, the authors show that the transformer can implement TD(λ) updates with a specific mask matrix, $M_{TD(\lambda)}$. The matrix structure requires strict alignment of eligibility traces, which might be difficult to achieve in real tasks where temporal dependencies and eligibility traces shift dynamically.
>
> We want to emphasize that we do not claim that this $TD(\lambda)$ weight construction emerges from any particular pre-training methods. Our result in Corollary 3 only states that transformers are expressive enough to exactly implement $TD(\lambda)$ within the forward pass, and makes no mention of how these $TD(\lambda)$ weights may actually be learned. This is an interesting avenue for future work.
>
> > 2. Lines 523-524 in the Conclusion are unclear. Are you trying to emphasize that policy iteration or optimization forms the key foundation?
>
> We find the question itself somewhat unclear and kindly ask the reviewer to elaborate. It is not evident what the reviewer means by "key foundation" or whether they are asking about policy iteration, policy evaluation, or optimization as foundational concepts in RL. To clarify our position, the mentioned lines in the conclusion emphasize that our work is a step toward understanding the phenomenon of in context RL (e.g. [1,2,3]). Specifically, we study policy evaluation, which is a fundamental component of RL. Policy evaluation serves as the backbone of policy iteration and optimization, making it essential for understanding and advancing RL methods [1]. Our work demonstrates that transformers can learn to solve policy evaluation tasks in-context (without gradient updates) by learning to implement TD methods in the forward pass.
>
> > 3. In Lines 531-532, the claim seems unsupported; working well in a small-scale setting doesn’t necessarily imply the same for larger, more complex tasks.
>
> We believe this comment reflects a misunderstanding of the goals of our work. While it is true that success in small-scale settings does not guarantee effectiveness in larger, more complex tasks, our primary contribution is to establish a formal equivalence between the behavior of our learned transformer and the well-known linear TD(0) algorithm. Unlike previous approaches, which treat the transformer as a black box, our work provides a "white-box" analysis by explicitly demonstrating how the transformer implements TD(0) within its forward pass. TD(0) has been a foundational algorithm in reinforcement learning since the 1980s and has been extensively validated across both small and large-scale tasks over decades of research [4]. Given this equivalence, further demonstrations of its effectiveness in larger-scale settings would be redundant and outside the scope of our work.
>
> We have revised the paper to clarify that by "effective in larger tasks", we mean it can achieve the same performance as TD. This should be evident because we proved that the learned transformer is mathematically equivalent to TD.
>
> The primary goal of our work is not to propose a new state-of-the-art RL algorithm for large, complex tasks. TD(0) is clearly not a state of the art algorithm. Rather, we aim to show that transformers can learn to implement TD(0) in-context, providing insight into the capacity of transformers to perform policy evaluation without the need for gradient updates.

---

> ### Author Response · Authors · 2024-11-21
> **References for response to Reviewer zyPc**
>
> **References**
> [1] Laskin, M., Wang, L., Oh, J., Parisotto, E., Spencer, S., Steigerwald, R., Strouse, D.J., Hansen, S.S., Filos, A., Brooks, E., Gazeau, M., Sahni, H., Singh, S., & Mnih, V. (2023). In-context Reinforcement Learning with Algorithm Distillation. *ICLR* 2023.\
> [2] Lee, J., Xie, A., Pacchiano, A., Chandak, Y., Finn, C., Nachum, O., and Brunskill, E. (2023). Supervised pretraining can learn in-context reinforcement learning. *NeurIPS* 2023.\
> [3] Raparthy, S. C., Hambro, E., Kirk, R., Henaff, M., & Raileanu, R. (2024). Learning to Solve New Sequential Decision-Making Tasks with In-Context Learning. *NeurIPS* 2024 FMDM Workshop\
> [4] Sutton, R., & Barto, A. (2018). Reinforcement learning: An introduction. MIT press.

---

> ### Author Response · Authors · 2024-11-21
> **Response to reviewer zyPc (part 3)**
>
> We want to thank the reviewer for their prompt reply. And we fully agree with the reviewer that it is better to be prudent to avoid any possible overstatement. We double checked the entire paper, it seems to us that the conclusion is the only part that risks overstatement. Correspondingly, in the new version, we have entirely rewritten that paragraph to explicitly acknowledge all the limitations raised by the reviewer. We are happy to address more if the reviewer finds any other sentences that risk overstatements and we hope this new "Conclusion" section can clear the reviewer's concern.

---

> ### Author Response · Authors · 2024-11-25
>
> We sincerely thank the reviewer for their valuable feedback and prompt engagement in the discussion. In response to the reviewer’s critique regarding the overstatement in our conclusion, we have revised the section to present more precise claims about our contributions and explicitly acknowledge the limitations of our work. Additionally, we conducted further experiments on the more complex CartPole environment and performed studies on in-context TD with the RNN architecture. We hope these efforts address the reviewer’s concerns and would greatly appreciate their opinion on the updated manuscript.

---

### Official Review · Reviewer_5HXw · 2024-11-03

**Soundness:** 3
**Presentation:** 3
**Contribution:** 3
**Rating:** 6
**Confidence:** 3

**Summary:**

The authors performed a mathematical analysis of the transformer architecture commonly used in the field of in-context reinforcement learning (ICRL). They proved that when a transformer with a linear attention structure is trained to estimate value functions in a multi-task environment, the forward pass within the transformer becomes equivalent to temporal-difference (TD) learning in reinforcement learning. Even when out-of-distribution task contexts are provided as input to the transformer, this TD learning effect is induced internally, enabling it to estimate the correct value function. The authors experimentally demonstrate this in a simple Boyan's chain environment.

**Strengths:**

1. The authors mathematically analyze that defining a Transformer as a value function estimator and training it in a multi-task environment makes the attention mechanism within the Transformer equivalent to TD(0) learning. The proof process appears to be concrete, and the resulting application seems straightforward. Particularly, while previous reinforcement learning studies using Transformers have merely mentioned its high performance, this paper analyzes why the Transformer structure works so effectively. In this regard, this paper seems to present a notable novelty.

2. The authors mathematically demonstrated that the internal operations of the transformer can be extended not only to TD(0) learning but also to various RL algorithms, such as TD($\lambda$), average reward TD, and residual gradient. This suggests that these algorithms can be applied in diverse ways to approximate a generalized value function through a transformer.

3. The proposed theorems and corollaries in the paper are meticulously detailed in the appendix, where each proof is presented with clarity and rigor. This comprehensive approach allows readers to follow the logical reasoning behind each statement. This thoroughness underpins the robustness of the paper and enhances its reproducibility, which will be beneficial for future researchers.

**Weaknesses:**

1. The experimental setup and evaluation metrics may not effectively demonstrate the algorithm's impact. The authors defined the value function estimation error for new states in out-of-distribution (OOD) tasks as the evaluation metric in Figure 1. However, since the Boyan's chain environment is relatively simple, it's possible that generalization effects could also be achieved using standard architectures like MLPs and RNNs, resulting in graphs similar to those in Figure 1. This raises the question of whether the results in Figure 1 reflect the effects of the transformer's in-context TD learning or if they could be similarly obtained with conventional networks. To address this, it seems the authors should consider conducting comparative experiments.

2. The comparison experiments seem insufficient. If there were experiments comparing value estimation between the authors' proposed value-estimating Transformer algorithm and existing reinforcement learning algorithms that utilize Transformers (other than behavior cloning methods), it would support the claim that the proposed algorithm is more effective. Although this paper focuses more on mathematical analysis, it seems to not have enough experiments on other algorithms or tasks.

3. The paper appears to be written in a way that makes it difficult to read. It seems to borrow equations and results from previous studies, but the notation and equations are challenging to follow at first glance. For instance, it is hard to understand how equations like Equation (4) were derived. A brief explanation of the existing equations would be helpful.

**Questions:**

1. The authors explain that in-context TD learning occurs when the task distribution is sufficiently challenging. I wonder why such a phenomenon occurs. They trained the transformer across thousands of Markov Reward Processes. Also, I wonder if the same TD learning effect would emerge internally within the transformer if it were trained in a single-task setup rather than a multi-task one.

2. In most prior research, transformers are directly applied to perform control tasks. However, in this paper, the transformer is used to estimate the value function. While a generalized approach to value function estimation is indeed important in reinforcement learning, this study does not directly address the control problem in RL. I would be interested to hear the authors' perspective on how their approach could be applied to control tasks in future research.

3. When a transformer estimates the value function, its internal operations align with TD learning. Most transformers, however, serve as policies rather than value function estimators. It would be interesting to know the authors’ perspective on whether, if the transformer were to act as a policy instead of estimating a value function, the internal operations would still resemble TD learning, or if another reinforcement learning algorithm would take place internally.

---

> ### Author Response · Authors · 2024-11-21
> **Response to Reviewer 5HXw**
>
> > Weaknesses:
> > 1. The experimental setup and evaluation metrics may not effectively demonstrate the algorithm's impact.  ... since the Boyan's chain environment is relatively simple, it's possible that generalization effects could also be achieved using standard architectures like MLPs and RNNs, ... This raises the question of whether the results in Figure 1 reflect the effects of the transformer's in-context TD learning or if they could be similarly obtained with conventional networks. To address this, it seems the authors should consider conducting comparative experiments.
>
> Transformers have become the architecture of choice in the ICRL community (e.g. [2,3,4]), supported by evidence that MLP's and RNNs fail to achieve the same level generalization. While a recent study [1] shows that MLPs can solve regression tasks with a fixed input size, a single MLP cannot solve out-of-distribution regression tasks with different input sizes. This limitation makes MLPs unsuitable for autoregressive decision-making in online environments, where the observation history grows dynamically—a key requirement for in-context reinforcement learning (ICRL) and for generating results like those in Figure 1.
>
> As for RNN's, based on the reviewer's feedback, we conducted additional experiments and found that RNNs failed to generalize to OOD tasks compared to transformers (Appendix G.1). Theoretical analysis (Appendix G.2) also shows that RNNs lack the expressivity to implement TD in context. This aligns with the study in Appendix K of [2] where they observe that LSTMs are less effective at performing Algorithm Distillation (a prominent ICRL method) than transformers. Even if another architecture, such as an MLP or RNN, could generalize better under specific pretraining, this would not diminish our key result: transformers learn to implement a known RL algorithm (TD) in-context, enabling generalization across tasks and sequence lengths.
>
> >2. The comparison experiments seem insufficient. If there were experiments comparing value estimation between the authors' proposed value-estimating Transformer algorithm and existing reinforcement learning algorithms that utilize Transformers (other than behavior cloning methods), it would support the claim that the proposed algorithm is more effective. Although this paper focuses more on mathematical analysis, it seems to not have enough experiments on other algorithms or tasks.
>
> We want to clarify that our primary goal is **not** to claim our pre-trained transformer achieves state-of-the-art performance on the policy evaluation task presented in Figure 1. In fact, a central claim of our work is that our learned transformer recovers the linear TD(0) algorithm, which is isn't a state-of-the-art RL algorithm. Instead, Figure 1 illustrates a well-documented phenomenon: transformers can solve out-of-distribution (OOD) RL tasks in-context (without gradient updates). This phenomenon, known as ICRL, has been observed in prior work (e.g., [2,3,4,6]). Unlike prior work, however, our contribution is the first to explain **how** transformers achieve ICRL, showing that they can and do learn to implement Temporal Difference (TD) learning in their forward pass.
>
> To address the concern that our work lacks sufficient experiments on other tasks, we added Appendix F, where we pre-train transformers on policy evaluation problems generated by the CartPole task—a more complex environment commonly used to benchmark RL algorithms. We conduct large-scale experiments with a context length of $n = 250$, observing once again that the transformers learn to implement in-context TD.
>
> It is worth noting that our theoretical results (Theorem 2), which prove that the weights implementing TD are in the invariant set of the updates in Algorithm 1, do not depend on any specific properties of the environment $P$. Thus, it is unsurprising that TD(0) emerges naturally even after pre-training on environments with complicated dynamics.
>
> >3. The paper appears to be written in a way that makes it difficult to read. It seems to borrow equations and results from previous studies, but the notation and equations are challenging to follow at first glance. For instance, it is hard to understand how equations like Equation (4) were derived. A brief explanation of the existing equations would be helpful.
>
> To address your concern, Eq. 4 is not a "derived" result, either in our work or in prior studies [7,8]. Instead, it establishes the notation convention we adopt to define the output of an $L$-layer transformer. Specifically, linear attention produces a matrix, but for policy evaluation, we require a scalar output. Following prior works, we define the bottom-right element of the output matrix as this scalar. To clarify, we have added an explanation of Eq. 4 to the main paper (written in turquoise).
>
> If there are any other specific equations you found unclear, we are happy to elaborate and include further clarifications in the main text.

---

> ### Author Response · Authors · 2024-11-21
> **Response to Reviewer 5HXw (part 2)**
>
> >Questions:
> > 1. The authors explain that in-context TD learning occurs when the task distribution is sufficiently challenging. I wonder why such a phenomenon occurs. They trained the transformer across thousands of Markov Reward Processes. Also, I wonder if the same TD learning effect would emerge internally within the transformer if it were trained in a single-task setup rather than a multi-task one.
>
> We thank the reviewer for raising this interesting and important question. As the reviewer correctly notes, we found that in-context TD learning emerges only when the transformer is trained across a diverse range of environments. When restricted to a single-task setup, the transformer can "cheat" by learning heuristics specific to that environment rather than a generalizable value estimation algorithm.
>
> By exposing the transformer to multiple tasks, we force its weights to generalize, leading to the emergence of a robust algorithm like TD. This aligns with previous studies (including in-context supervised learning studies) which demonstrate that task diversity during training plays a crucial role in encouraging the development of generalizable in-context learning capabilities [9,2,3].
>
> > 2. In most prior research, transformers are directly applied to perform control tasks. However, in this paper, the transformer is used to estimate the value function. While a generalized approach to value function estimation is indeed important in reinforcement learning, this study does not directly address the control problem in RL. I would be interested to hear the authors' perspective on how their approach could be applied to control tasks in future research.
>
> Several works [2,3,6] explore transformers' ability to perform control tasks in-context, but the reasons for their success remain a black box. Our study takes the first step toward "white-boxing" in-context RL by investigating how transformers perform policy evaluation. While we do not directly improve in-context RL for control, understanding in-context policy evaluationserves as a stepping stone toward uncovering the mechanisms behind in-context control.
>
> Many prominent control algorithms, such as Q-learning and SARSA, are based on TD methods. Our results show that transformers can learn to implement TD(0) for policy evaluation in-context, suggesting they might also learn TD-based control algorithms like Q-learning and SARSA in-context. For example, Q-learning extends TD(0) by requiring a max operation over the action space in its update rule for the state-action values. While this max operation adds soem complexity, our understanding of how transformers implement TD(0) provides a foundation for exploring whether and how they might implement Q-learning in the forward pass in future work.
>
> In addition to just TD based control algorithms, we also envision our work to help understand whether transformers can "plan" in context. For example, if we establish that transformers can implement Q-learning in the forward pass, it is also possible that transformers might be able to implement something like Dyna in deterministic environments to integrate learning and model-based planning.
>
>
> > 3. When a transformer estimates the value function, its internal operations align with TD learning. Most transformers, however, serve as policies rather than value function estimators. It would be interesting to know the authors’ perspective on whether, if the transformer were to act as a policy instead of estimating a value function, the internal operations would still resemble TD learning, or if another reinforcement learning algorithm would take place internally.
>
> This is another great question. It’s important to clarify that the TD(0) algorithm we proved transformers can implement is inherently a value estimation algorithm and cannot directly function as a policy. To derive a policy, we would need a different TD-based algorithm like Q-learning, which computes Q-values over state-action pairs and selects actions using a greedy or $epsilon$-greedy strategy. Therefore, if a transformer were trained to act as a policy rather than estimate a value function, we believe its weights might learn to implement a control algorithm, potentially involving elements of planning or exploration beyond basic TD learning.
>
> Interestingly, prior works like [2] have shown that with supervised pretraining on control tasks, transformers can effectively behavior clone RL algorithms like DQN and PPO. While this demonstrates that transformers can replicate the behavior of DQN, it doesn’t directly reveal whether the internal mechanism for action selection actually mirrors Deep Q-learning or involves a different and possibly unknown RL algorithm that *behaves* similarly. We agree with the reviewer that this is an interesting avenue for future research.

---

> ### Author Response · Authors · 2024-11-21
> **References for response to reviewer 5XHw**
>
> **References**\
> [1] Tong, WL., Pehlevan C., (2024). MLPs Learn In-Context on Regression and Classification Tasks. *arxiv preprint*.\
> [2] Laskin, M., Wang, L., Oh, J., Parisotto, E., Spencer, S., Steigerwald, R., Strouse, D.J., Hansen, S.S., Filos, A., Brooks, E., Gazeau, M., Sahni, H., Singh, S., & Mnih, V. (2023). In-context Reinforcement Learning with Algorithm Distillation. *ICLR* 2023.\
> [3] Lee, J., Xie, A., Pacchiano, A., Chandak, Y., Finn, C., Nachum, O., and Brunskill, E. (2023). Supervised pretraining can learn in-context reinforcement learning. *NeurIPS* 2023.\
> [4] Raparthy, S. C., Hambro, E., Kirk, R., Henaff, M., & Raileanu, R. (2024). Learning to Solve New Sequential Decision-Making Tasks with In-Context Learning. *NeurIPS* 2024 FMDM Workshop\
> [5] Xu, M., Shen, Y., Zhang, S., Lu, Y., Zhao, D., Tenenbaum, J., and Gan, C. (2022). Prompting decision Transformer for Fewshot Policy Generalization. *ICML* 2022.\
> [6] Sinii, V., Nikulin, A., Kurenkov, V., Zisman, I., Kolesnikov, S. (2024). In-Context Reinforcement Learning for Variable Action Spaces. *ICML* 2024.\
> [7] von Oswald, J., Niklasson, E., Randazzo, E., Sacramento, J., Mordvintsev, A., Zhmoginov, A., Vladymyrov, M. (2022). Transformers learn in-context by gradient descent. *CoRR* 2022.\
> [8] Ahn, K., Cheng, X., Daneshmand, H., & Sra, S. (2023). Transformers learn to implement preconditioned gradient descent for in-context learning. *NeurIPS* 2023.\
> [9] Chan, S., Santoro, A., Lampinen, A., Wang, J., Singh, A., Richemond, P., McClelland, J., & Hill, F. (2022) Data Distributional Properties Drive Emergent In-Context Learning in Transformers. *Neurips* 2022.

---

> ### Author Response · Authors · 2024-11-25
>
> We sincerely thank the reviewer again for their detailed and comprehensive feedback. In response to the reviewer’s suggestions, we have conducted additional experiments on in-context TD using the CartPole environment and carried out further theoretical and empirical studies on in-context TD with RNNs. These new results have been added to the Appendix. Additionally, we have provided explanations and justifications for our notations. We would greatly appreciate the reviewer’s updated opinion on the revised manuscript.

---

### Official Review · Reviewer_77PR · 2024-11-08

**Soundness:** 4
**Presentation:** 2
**Contribution:** 2
**Rating:** 8
**Confidence:** 2

**Summary:**

The paper shows that linear transformers can execute TD(0) over their context both by constructing the matrices that would do so, but also by showing a pre-training method that gives rise to such properties. Finally they show similar constructions for other Policy Evaluation algorithms like TD-lambda, Average-Reward TD etc. Besides some empirical evidences the paper presents many proofs of its statements.

**Strengths:**

The paper argues with proofs and empirical evidences for its results and is able to remove any possible doubt of their veracity. Furthermore it's a novel exploration of the capabilities of linear transformers and further helps the understanding of In-Context Reinforcement Learning.

**Weaknesses:**

The title of the paper is rather strong, not only is the paper only about Linear Attention, as opposed to the commonly used Softmax attention, but also they prove that transformers with a specific kind of training implement TD-Learning, so I believe "Transformers Can Learn Temporal Difference Methods for In-Context Reinforcement Learning" would be more appropriate.  Furthermore one must wonder how relevant is studying Reinforcement Learning implemented by linear attention over such short horizons such as 40 steps, as most RL problems involve orders of magnitude longer trajectories, whereas attention tends to become computationally expensive and lose precision as the context length grows.

**Questions:**

1. Would it be possible for the authors to change the paper's title to something that better reflects what is shown by the paper?
2. Could the authors better justify the study of In-Context RL, both for Linear Attention-based transformers and otherwise?

---

> ### Author Response · Authors · 2024-11-21
> **Response to Reviewer 77PR**
>
> >Weaknesses: The title of the paper is rather strong, not only is the paper only about Linear Attention, as opposed to the commonly used softmax attention, but also they prove that transformers with a specific kind of training implement TD-Learning, so I believe "Transformers Can Learn Temporal Difference Methods for In-Context Reinforcement Learning" would be more appropriate.
>
> We appreciate the feedback and have updated the title of the paper to "Transformers Can Learn Temporal Difference Methods for In-Context Reinforcement Learning" as proposed.
>
> We would like to emphasize that our work demonstrates that, under the specific pretraining scheme outlined in Algorithm 1, transformers with linear attention do indeed learn to implement TD(0). We acknowledge and explicitly state in the paper that our results, in its current form, are not generalizable to all possible pretraining schemes (e.g., supervised pretraining) and that our theoretical findings are specific to linear attention. That being said, we agree the revised title better reflects the scope and contributions of our work.
>
> >Furthermore one must wonder how relevant is studying Reinforcement Learning implemented by linear attention over such short horizons such as 40 steps, as most RL problems involve orders of magnitude longer trajectories, whereas attention tends to become computationally expensive and lose precision as the context length grows.
>
>
> We recognize the computational challenges associated with transformers, particularly their $O(n^2)$ complexity with respect to context length $n$. Despite this, a growing body of research highlights their feasibility and effectiveness for in-context reinforcement learning (ICRL), even in large-scale RL problems. For example, studies such as [1,2] provide empirical evidence that transformers can effectively manage longer trajectories of  approximately 5000 environment steps and demonstrate improved performance as the context length increases. These studies show that transformers can successfully solve tasks over 50 episodes (each episode is 100 steps), where only the last n=200 steps are provided as context. Remarkably, this is achieved without gradient updates during inference. [3] conducts even larger experiments demonstrating the ICRL ability of transformers with ~310M parameters with context lengths of $n=2048$ tokens. Such findings validate the capability of transformers for ICRL in scenarios requiring significantly longer horizons. To further demonstrate this phenomenon in our work, we added some additional experiments in Appendix F where the pre-training tasks were constructed from the CartPole environment, which required us to study much longer trajectories with a context length of $n=250$.
>
> The goal of our work is not to replicate or "outperform" these large-scale results, but to provide theoretical insights and experiments that deepen our understanding of the ICRL phenomenon observed in them. While our experiments focus on shorter horizons, they serve as a foundation for understanding how transformers implement TD(0) and for exploring the theoretical underpinnings of ICRL.
>
> Additionally, we note the emergence of related work investigating the ICRL capabilities of less computationally expensive models, such as structured state-space models (SSMs) and Mamba, which have $O(n)$ complexity. We agree this is an exciting avenue for future research and will include a discussion of this limitation and potential future directions in the camera-ready version of our paper.

---

> ### Author Response · Authors · 2024-11-21
> **References for Response to Reviewer 77PR**
>
> **References**
> [1] Laskin, M., Wang, L., Oh, J., Parisotto, E., Spencer, S., Steigerwald, R., Strouse, D.J., Hansen, S.S., Filos, A., Brooks, E., Gazeau, M., Sahni, H., Singh, S., & Mnih, V. (2023). In-context Reinforcement Learning with Algorithm Distillation. *ICLR* 2023.\
> [2] Lee, J., Xie, A., Pacchiano, A., Chandak, Y., Finn, C., Nachum, O., and Brunskill, E. (2023). Supervised pretraining can learn in-context reinforcement learning. *NeurIPS* 2023.\
> [3] Raparthy, S. C., Hambro, E., Kirk, R., Henaff, M., & Raileanu, R. (2024). Learning to Solve New Sequential Decision-Making Tasks with In-Context Learning. *NeurIPS* 2023 FMDM Workshop\
> [4]Mahankali, A. V., Hashimoto, T., & Ma, T. (2024). One Step of Gradient Descent is Provably the Optimal In-Context Learner with One Layer of Linear Self-Attention. *ICLR* 2024.\
> [5] Li, Y., Rawat, A. S., & Oymak, S. (2024). Fine-grained Analysis of In-Context Linear Estimation. *ICML* 2024 HiLD Workshop\
> [6] Ahn, K., Cheng, X., Daneshmand, H., & Sra, S. (2023). Transformers learn to implement preconditioned gradient descent for in-context learning. *NeurIPS* 2023.
> [7] Sander, M. E., Giryes, R., Suzuki, T., Blondel, M., & Peyré, G. (2024). How do Transformers perform In-Context Autoregressive Learning?. arXiv preprint arXiv:2402.05787.

---

> ### Author Response · Authors · 2024-11-21
> **Response to Reviewer 77PR (part 2)**
>
> > Could the authors better justify the study of In-Context RL, both for Linear Attention-based transformers and otherwise?
>
> We have justified the study of in-context RL in our response to the raised weaknesses.
> While we acknowledge that our analysis using linear attention can be limiting, we wish to argue that theories on neural networks are notoriously challenging to prove, not to mention the novelty of our construction. Sometimes, theories on a single-layer neural network require a lengthy paper to explain. Examples include [4] and [5]. We make up for this shortcoming with an empirical study using softmax activations in Appendix E to provide evidence for the emergence of in-context TD in canonical transformers. In addition, linear attention is a common variant for simplifying analysis and computation. Please see our footnote on page 2 for an extensive list of references. We also wish to disclose that we drew our initial inspiration from the broader in-context learning community, where linear attention is a common assumption in theoretical analysis. Some recent works using linear attention to analyze in-context linear regression include [6] and [7]. To our knowledge, our work is the first to apply this idea to provide a concrete mathematical understanding of the mechanisms of ICRL with transformers.

---

> ### Author Response · Authors · 2024-11-25
>
> We sincerely thank the reviewer again for their constructive feedback. As per the reviewer's suggestion, we have updated the title of our paper to "Transformers **Can** Learn Temporal Difference Methods for In-Context Reinforcement Learning" and provided additional justifications for in-context RL and linear attention in our earlier responses. We would love to learn the reviewer's updated opinion on the current version of our manuscript.

---

> > ### Comment · Reviewer_77PR · 2024-11-25
> >
> > Thank you for changing the title, and I have been convinced of the theoretical, and up to some point practical, relevance of this line of research

---

> ### Comment · Reviewer_77PR · 2024-12-02
>
> After further reflection I believe the paper is proving what it sets out to prove, and that the results is indeed interesting, at least as a way to better understand the capabilities of transformer-like architectures and attention, and maybe even imply some interesting practical results. As such I think the paper is on a good level for Neurips and I have accordingly increased my score to 8.

---

### Meta-Review · Area_Chair_y51H · 2024-12-19

**Metareview:**

This paper presents a "proof by construction" arguing that transformers with linear attention can implement temporal difference (TD) learning during the forward pass (in-context TD). The authors provide theoretical and empirical evidence for this phenomenon.

The theoretical insights provided by this work are both interesting and novel, and can lay the groundwork for future research in this space. The clarity of proofs in the appendix are also appreciated.

There were some concerns with the claims made by the authors, as well as the extent of the empirical evaluations.

However, after the changes made during the discussion phase, I believe this is a paper worthy of an ICLR acceptance and which will be of interest to many in the community.

**Additional Comments On Reviewer Discussion:**

There were a number of concerns raised with regards to overclaiming (including the original title). The authors have successfully addressed these during the rebuttal.

Some of the concerns regarding empirical evaluations were also mostly addressed.

---

### Decision · Program_Chairs · 2025-01-22

Accept (Poster)